# Convergence Analysis of Tsetlin Machines under Noise-Free and Noisy Training Conditions: From 2 Bits to $k$ Bits

**Xuan Zhang**[1,2], **Lei Jiao**[1]*, **Ole-Christoffer Granmo**[1]
[1]The Centre for Artificial Intelligence Research, University of Agder, Grimstad, Norway
[2]NORCE Research AS (NORCE), Grimstad, Norway
`xuzh@norceresearch.no; {lei.jiao;ole.granmo}@uia.no`

## Abstract

The Tsetlin Machine (TM) is an innovative machine learning algorithm grounded in propositional logic, achieving state-of-the-art performance across a variety of pattern recognition tasks. Prior theoretical work has established convergence results for the 1-bit operator under both noisy and noise-free conditions, and for the 2-bit XOR operator under noise-free conditions. This paper first extends the analysis to the 2-bit AND and OR operators. We show that the TM converges almost surely to the correct 2-bit AND and OR operators under the noise-free training condition, and we identify a distinctive property of the 2-bit OR operator, where a single clause can jointly represent two sub-patterns, in contrast to the XOR operator. We further investigate noisy training scenarios, demonstrating that mislabelled samples prevent exact convergence but still permit efficient learning, whereas irrelevant variables do not prevent almost-sure convergence. Building on the 2-bit analysis, we then generalize the results to the $k$-bit setting ($k > 2$), providing a unified theoretical treatment applicable to general scenarios. Together, these findings provide a robust and comprehensive theoretical foundation for analyzing TM convergence.

## 1 Introduction

The Tsetlin Machine (TM) is a classification algorithm. A TM (Granmo, 2018) organizes clauses, each associated with a team of Tsetlin Automata (TAs) (Tsetlin, 1961), to collaboratively capture distinct sub-patterns[1] for a certain class. A TA, which is the core learning entity of TM, is a kind of learning automata (Zhang et al., 2020; Yazidi et al., 2019; Omslandseter et al., 2024) that tackles the multi-armed bandit problem, learning the optimal action through the interaction with its environment which gives rewards and penalties. In a TM, all TAs play in a game orchestrated by the TM's feedback tables. Each TA takes care of one literal of input, which takes boolean values of either 0 or 1. Literals are basically features of the input data. A TA decides to "Include" or "Exclude" the literal, i.e., to consider or not to consider the feature in the final classification. A clause is a conjunction of all included literals, representing a sub-pattern of a certain class. Once distinct sub-patterns are learned by a number of clauses, the overall pattern recognition task is completed by a voting scheme from the clauses.

The TM and its variants (Granmo et al., 2019; Abeyrathna et al., 2021; Darshana Abeyrathna et al., 2020; Sharma et al., 2023) have been applied to diverse tasks, including word sense disambiguation (Yadav et al., 2021c), aspect-based sentiment analysis (Yadav et al., 2021b), novelty detection (Bhattarai et al., 2021), interpretable text classification (Yadav et al., 2021a; Yadav et al., 2022), federated learning (Qi et al., 2025), signal classification (Jeeru et al., 2025a;b), and contextual bandits (Seraj et al., 2022), where they often match or surpass state-of-the-art methods. As a symbolic model, the TM offers transparent learning and inference (Granmo et al., 2025; Bhattarai et al., 2024; Abeyrathna et al., 2023; Rafiev et al., 2022). Its reliance solely on logical operations also makes

---

*Corresponding author.
[1]The concept of sub-pattern will be found in the example given in Section 2.

it hardware-friendly and energy-efficient (Maheshwari et al., 2023; Rahman et al., 2022; Kishore et al., 2023; Tunheim et al., 2025a;b). Appendix L provides a more detailed account of real-world application examples.

The TM is proven to almost surely convergence to the Identity/NOT operator with 1-bit input in (Zhang et al., 2022), where the role of the hyperparameter $s$ is also revealed. In (Jiao et al., 2023), TM's convergence to the XOR operator with 2-bit input was proven, highlighting the functionality of the hyperparameter $T$. In this paper, we first analyze the 2-bit AND and OR operators using noise-free training samples. We then examine the convergence properties of AND, OR, and XOR under noisy conditions, including scenarios with incorrect labels and irrelevant inputs. Finally, we extend these results to the general $k$-bit case.

This paper differs from previous studies in several key aspects. While (Zhang et al., 2022) used stationary distribution analysis of discrete-time Markov chains (DTMC), the current study focuses on absorbing states. For XOR (Jiao et al., 2023), where sub-patterns are bit-wise exclusive, TM learns and converges to sub-patterns individually. In contrast, the OR operator's sub-patterns share features (e.g., $[x_1 = 1, x_2 = 1]$ and $[x_1 = 1, x_2 = 0]$ share $x_1 = 1$), allowing joint representation. We show that TM can effectively learn and represent these shared features, making the convergence process distinct. Additionally, this paper examines the role of Type II feedback, which was omitted in the prior XOR convergence study. Most notably, we analyze the convergence properties of the AND, OR, and XOR operators under noisy training samples, and extend these results to the general $k$-bit case, thereby making the analysis comprehensive and conclusive.

It is worth noting that learning $k$-bit operators with or without noise, is a well-studied problem. For example, numerous studies in concept learning and probably approximately correct learning have extensively explored this topic (Valiant, 1984; Haussler et al., 1994; Mansour & Parnas, 1998; Belaid et al., 2025). While many elegant methods exist for learning conjunctions or disjunctions, their existence does not necessarily imply that the TM converges to such operators in the same manner. TM employs a unique approach, learning from samples to construct conjunctive expressions and coordinating these expressions across various sub-patterns, which merits its own dedicated analysis.

## 2 NOTATIONS OF THE TM

To make the article self-contained, we present the TM notation. For more details on the inference and training concept, please refer to Appendix A.

The input of a TM is indicated as $\mathbf{X} = [x_1, x_2, \ldots, x_o]$, where $x_k \in \{0, 1\}$, $k = 1, 2, \ldots, o$, and $o$ is the number of features. A literal is either $x_k$ in the original form or its negation $\neg x_k$. A clause is a conjunction of literals. Each literal is associated with a TA. The TA is a 2-action learning automaton whose job is to decide whether to Include/Exclude its literal in/from the clause, based on the current state of the TA. A clause is associated with $2o$ TAs, forming a TA team. A TA team is denoted in general as $\mathcal{G}_j^i = \{\text{TA}_{k'}^{i,j} | 1 \leq k' \leq 2o\}$, where $k'$ is the index of the TA, $j$ is the index of the TA team/clause (multiple TA teams form a TM), and $i$ is the index of the TM/class to be identified (A TM identifies a class, multiple TMs identify multiple classes).

Suppose we are investigating the $i^{th}$ TM whose job is to identify class $i$, and that the TM is composed of $m$ TA teams. Then $C_j^i(\mathbf{X})$ can be used to denote the output of the $j^{th}$ TA team, which is a conjunctive clause:

$$\text{Training}: C_j^i(\mathbf{X}) = \begin{cases} \left( \bigwedge_{k \in \xi_j^i} x_k \right) \wedge \left( \bigwedge_{k \in \bar{\xi}_j^i} \neg x_k \right), & \text{for } \xi_j^i, \bar{\xi}_j^i \neq \emptyset, \\ 1, & \text{for } \xi_j^i, \bar{\xi}_j^i = \emptyset. \end{cases} \quad (1)$$

$$\text{Testing}: C_j^i(\mathbf{X}) = \begin{cases} \left( \bigwedge_{k \in \xi_j^i} x_k \right) \wedge \left( \bigwedge_{k \in \bar{\xi}_j^i} \neg x_k \right), & \text{for } \xi_j^i, \bar{\xi}_j^i \neq \emptyset, \\ 0, & \text{for } \xi_j^i, \bar{\xi}_j^i = \emptyset. \end{cases} \quad (2)$$

In Eqs. (1) and (2), $\xi_j^i$ and $\bar{\xi}_j^i$ are defined as the sets of indexes for the literals that have been included in the clause. $\xi_j^i$ contains the indexes of included original inputs, $x_k$, whereas $\bar{\xi}_j^i$ contains the indexes of included negated inputs, $\neg x_k$.

Each clause represents a sub-pattern associated with class $i$ by including a literal (a feature or its negation) if it contributes to the sub-pattern, or excluding it when deemed irrelevant. Multiple clauses, i.e., the TA teams, are assembled into a complete TM to sum up the outputs of the clauses $f_\sum(\mathcal{C}^i(\mathbf{X})) = \sum_{j=1}^{m} C_j^i(\mathbf{X})$, where $\mathcal{C}^i(\mathbf{X})$ is the set of clauses for class $i$. The output of the TM is further determined by the unit step function: $\hat{y}^i = \begin{cases} 0, & \text{for } f_\sum(\mathcal{C}^i(\mathbf{X})) < Th \\ 1, & \text{for } f_\sum(\mathcal{C}^i(\mathbf{X})) \geq Th \end{cases}$, where $Th$ is a predefined threshold for classification. This is indeed a voting scheme.

Note that the TM can assign polarity to each TA team (Granmo, 2018), and one can refer to Appendix A for more information. In this study, for ease of analysis, we consider only positive polarity clauses. Nevertheless, this does not change the nature of TM learning.

**Example:** We use TM learning the OR logic as an example. A sample is classified into the OR class if its two bits and label follow the OR logic: $0\,1 \Rightarrow 1$, $1\,0 \Rightarrow 1$, $1\,1 \Rightarrow 1$, or $0\,0 \Rightarrow 0$. Note that once the TM learns the pattern outputting 1, it inherently learns the complementary pattern outputting 0. Hence the TM's learning and reasoning can be understood primarily as identifying the pattern that results in an output of 1. In the OR logic, sub-patterns outputting 1 are $0\,1$, $1\,0$, and $1\,1$, and can be represented by clauses $\neg x_1 \wedge x_2$, $x_1 \wedge \neg x_2$, and $x_1 \wedge x_2$, respectively.

A clause is learned by a TA team, and a TM can be composed of multiple TA teams. A TA team is a set of TAs, each responsible for handling one literal. A literal is an input feature, in this example, $x_1$ or $x_2$, or the negation of it: $\neg x_1$ or $\neg x_2$. In this example, each TA team consists of four TAs, managing four literals: $x_1$, $\neg x_1$, $x_2$, $\neg x_2$, respectively.

A TA decides, by its current state (which changes according to the state-transition probabilities as shown in Table 1 and Table 2), whether to Include or Exclude its literal in/from the final clause. In a TA team of four TAs, if $TA_1$ includes $x_1$, $TA_2$ excludes $\neg x_1$, $TA_3$ excludes $x_2$, and $TA_4$ includes $\neg x_2$, the resulting clause from this TA team will be $x_1 \wedge \neg x_2$.

A TM learns the pattern of the OR relationship from the input samples that follow the OR logic (training). As the training result, some TA teams converge to clauses like $\neg x_1 \wedge x_2$, others to $x_1 \wedge \neg x_2$, or $x_1 \wedge x_2$, all outputting 1. The process of determining whether an input conforms to the OR logic involves summing the outputs of all the clauses. Let's assume we have three TA teams, each converging to one of the sub-patterns, then the sum is $sum = (\neg x_1 \wedge x_2) + (x_1 \wedge \neg x_2) + (x_1 \wedge x_2)$. If a test sample $\{[x_1, x_2], y\} = \{[0, 1], 1\}$ is put into the TM, the output will be $sum = (1 \wedge 1) + (0 \wedge 0) + (0 \wedge 1) = 1 + 0 + 0 = 1$, indicating one TA team votes for positive classification. If the threshold $Th$ is defined as 1, as $sum \geq Th$, TM evaluates the sample following the OR logic (testing).

**Training:** In the training of a TM, the labeled data $(\mathbf{X} = [x_1, x_2, ..., x_o], y^i)$ is fed into the TM, where the TAs are guided by the feedback defined in Tables 1 and 2. Type I Feedback is triggered when the training sample has a positive label: $y^i = 1$, while Type II feedback is utilized when $y^i = 0$. $s$ controls the granularity of the clauses. NA means not applicable. Examples demonstrating TA state transitions per feedback tables can be found in Section 3.1 in (Zhang et al., 2022). In brief, Type I feedback reinforces true positive and Type II feedback fights against false negative.

| Value of the clause $C_j^i(\mathbf{X})$ | | 1 | | 0 | |
| --- | --- | --- | --- | --- | --- |
| *Value of the Literal $x_k/\neg x_k$* | | 1 | 0 | 1 | 0 |
| **Include Literal** | $P(\text{Reward})$ | $\frac{s-1}{s}$ | NA | 0 | 0 |
| | $P(\text{Inaction})$ | $\frac{1}{s}$ | NA | $\frac{s-1}{s}$ | $\frac{s-1}{s}$ |
| | $P(\text{Penalty})$ | 0 | NA | $\frac{1}{s}$ | $\frac{1}{s}$ |
| **Exclude Literal** | $P(\text{Reward})$ | 0 | $\frac{1}{s}$ | $\frac{1}{s}$ | $\frac{1}{s}$ |
| | $P(\text{Inaction})$ | $\frac{1}{s}$ | $\frac{s-1}{s}$ | $\frac{s-1}{s}$ | $\frac{s-1}{s}$ |
| | $P(\text{Penalty})$ | $\frac{s-1}{s}$ | 0 | 0 | 0 |

Table 1: Type I Feedback — Feedback upon receiving a sample with label $y^i = 1$, for a single TA to decide whether to Include or Exclude a given literal $x_k/\neg x_k$ into $C_j^i$. NA means not applicable (Granmo, 2018).

| Value of the clause $C_j^i(\mathbf{X})$ | | 1 | | 0 | |
|---|---|---|---|---|---|
| Value of the Literal $x_k/\neg x_k$ | | 1 | 0 | 1 | 0 |
| **Include Literal** | $P(\text{Reward})$ | 0 | NA | 0 | 0 |
| | $P(\text{Inaction})$ | 1.0 | NA | 1.0 | 1.0 |
| | $P(\text{Penalty})$ | 0 | NA | 0 | 0 |
| **Exclude Literal** | $P(\text{Reward})$ | 0 | 0 | 0 | 0 |
| | $P(\text{Inaction})$ | 1.0 | 0 | 1.0 | 1.0 |
| | $P(\text{Penalty})$ | 0 | 1.0 | 0 | 0 |

Table 2: Type II Feedback — Feedback upon receiving a sample with label $y^i = 0$, for a single TA to decide whether to Include or Exclude a given literal $x_k/\neg x_k$ into $C_j^i$. (Granmo, 2018).

To avoid situations where a majority of the TA teams learn a *subset* of sub-patterns, forming an incomplete representation[2], the hyperparameter $T$ is used to regulate the resource allocation. The strategy works as follows (Granmo, 2018):

**Generating Type I Feedback.** If the label of the training sample $\mathbf{X}$ is $y^i = 1$, we generate, in probability, *Type I Feedback* for each clause $C_j^i \in \mathcal{C}^i$ according to:

$$u_1 = \frac{T - \max(-T, \min(T, f_\Sigma(\mathcal{C}^i(\mathbf{X}))))}{2T}. \tag{3}$$

**Generating Type II Feedback.** If the label of the training sample $\mathbf{X}$ is $y^i = 0$, we generate, again, in probability, *Type II Feedback* to each clause $C_j^i \in \mathcal{C}^i$ according to:

$$u_2 = \frac{T + \max(-T, \min(T, f_\Sigma(\mathcal{C}^i(\mathbf{X}))))}{2T}. \tag{4}$$

Here $T$ is a positive integer, with its maximum value equal to the total number of clauses. When multiple sub-patterns exist, $T$ limits the maximum number of clauses that can be allocated to each sub-pattern. Briefly speaking, when the number of clauses representing one sub-pattern increases, learning from samples that correspond to that sub-pattern will decrease as the probability of triggering update will decrease. Once at least $T$ clauses have learned a particular sub-pattern, any samples matching that sub-pattern will no longer trigger TM updates (the probability of triggering feedback is 0). This prevents additional clause resources from being spent on a sub-pattern that is already considered learned (with $T$ clauses representing it). With an appropriate choice of $T$, the clause resources can be balanced across different sub-patterns, ensuring convergence of the system. In addition, $T$ plays a crucial role in maintaining convergence when irrelevant bits are present. Further insights and discussions on the role of $T$ are provided in the proofs.

## 3 CONVERGENCE ANALYSIS OF THE AND OPERATOR

A TM has converged when the states of its TAs do not change any longer. We assume that the training samples are noise free, i.e., $P(y = 1|x_1 = 1, x_2 = 1) = 1, P(y = 0|x_1 = 0, x_2 = 1) = 1, P(y = 0|x_1 = 1, x_2 = 0) = 1, P(y = 0|x_1 = 0, x_2 = 0) = 1$. We assume that the training samples are independently drawn from a distribution with full support over the four cases, i.e., each case has strictly positive probability and therefore occurs infinitely often almost surely as time progresses. The same sampling assumption is maintained throughout all subsequent operator analyses.

Because the considered AND operator has only one sub-pattern of input, i.e., $x_1 = 1, x_2 = 1$, that will trigger a true output, we employ one clause in this TM, and we thus can ignore the indices of the classes and the clauses in the notation in the proof. After simplification, $\text{TA}_k^{i,j}$ becomes $\text{TA}_k$, and $C_1^1$ becomes $C$. Since there are two input parameters, namely $x_1$ and $x_2$, we implement four TAs in the clause, i.e., $\text{TA}_1, \text{TA}_2, \text{TA}_3$, and $\text{TA}_4$. $\text{TA}_1$ has two actions, i.e., including or excluding $x_1$. Similarly, $\text{TA}_2$ corresponds to including or excluding $\neg x_1$. $\text{TA}_3$ and $\text{TA}_4$ determine the behavior of $x_2$ and $\neg x_2$, respectively.

Once the TM converge correctly to the intended operation, the resulting clause will be $x_1 \wedge x_2$, with the actions of $\text{TA}_1, \text{TA}_2, \text{TA}_3$, and $\text{TA}_4$ being I, E, I, and E, respectively. Here we use "I" and "E" as abbreviations for include and exclude respectively.

---

[2]In the OR example, one should avoid to have a majority of TA teams converge to $\neg x_1 \wedge x_2$ to represent the sub-pattern of $[0, 1]$, and ignore the other sub-patterns $[1, 0]$ and $[1, 1]$.

**Theorem 1.** *Any clause will converge almost surely to $x_1 \wedge x_2$ given noise free AND training samples in infinite time when $u_1 > 0$ and $u_2 > 0$.*

The complete proof of Theorem 1 is in Appendix B. We here outline the main steps of the proof.

The condition $u_1 > 0$ and $u_2 > 0$ guarantees that all types of samples are provided to the TM and no specific type is blocked by Eqs. (3) and (4) during training. The goal of the proof is to show that the system transitions will guarantee that there is a unique absorbing state of the TM and the absorbing state has the actions of $TA_1$, $TA_2$, $TA_3$, and $TA_4$ to be I, E, I, E, respectively, corresponding to the expression $x_1 \wedge x_2$.

To simplify the analysis of joint TA transitions, we use quasi-stationary analysis by freezing the transitions of the TAs for the first input bit and focusing on the transitions of the TAs corresponding to the second input bit. Clearly, there are four possibilities when freezing the first bit $x_1$. We name them as cases: **Case 1:** $TA_1 = E$, $TA_2 = I$, i.e., include $\neg x_1$. **Case 2:** $TA_1 = I$, $TA_2 = E$, i.e., include $x_1$. **Case 3:** $TA_1 = E$, $TA_2 = E$, i.e., exclude both $x_1$ and $\neg x_1$. **Case 4:** $TA_1 = I$, $TA_2 = I$, i.e., include both $x_1$ and $\neg x_1$.

In each of the above four cases, we analyze individually the transition of $TA_3$ ($TA_4$) with a given current action, under different actions of $TA_4$ ($TA_3$). We index the possibilities as situations: **Situation 1.** We study the transition of $TA_3$ when its current action is "Include", and when $TA_4$ is frozen to be "Include" or "Exclude". **Situation 2.** We study the transition of $TA_3$ when its current action is "Exclude", and when $TA_4$ is frozen to be "Include" or "Exclude". **Situation 3.** We study the transition of $TA_4$ when its current action is "Include", and when $TA_3$ is frozen to be "Include" or "Exclude". **Situation 4.** We study the transition of $TA_4$ when its current action is "Exclude", and when $TA_3$ is frozen to be "Include" or "Exclude".

Within each of the situation, there are 8 possible instances, determined by 4 possible combinations of the input samples of $x_1$ and $x_2$, and the two possible frozen TA actions, i.e., Include and Exclude.

As an example, we randomly select an instance in Case 1, Situation 1. The selected instance is when the training sample is $([x_1 = 1, x_2 = 1], y = 1)$, and $TA_4$ is E. For this instance, the training sample will trigger Type I feedback because $y = 1$. Based on the current status of the TAs, the clause is in the form $C = \neg x_1 \wedge x_2$, which evaluates to 0 based on the input training sample. In Situation 1, the studied TA is $TA_3$, whose corresponding literal is $x_2 = 1$. Given $y = 1$, clause value 0, literal value 1, we go to Table 1, the third column of transition probabilities for "Include Literal", and find the transition of $TA_3$ to be: the penalty probability $\frac{1}{s}$ and the inaction probability $\frac{s-1}{s}$. To indicate the transitions of $TA_3$, we have plotted the transition diagram in Fig. 1. Note that the overall transition probability is $u_1 \frac{1}{s}$, where $u_1$ is defined in Eq. (3). Here, we have assumed $u_1 > 0$.

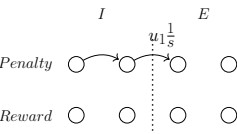

Figure 1: Transition of $TA_3$ when its current action is Include, $TA_1$, $TA_2$, and $TA_4$'s actions are Exclude, Include, and Exclude, respectively, upon a training sample $([x_1 = 1, x_2 = 1], y = 1)$.

Similar to the example instance, we derive a total of 128 transition instances, which can be further summarized into the overall transition behavior of $TA_3$ and $TA_4$. These overall transitions reveal the directional dynamics of the two TAs, from which we observe that the unique absorbing state for $TA_3$ and $TA_4$ is $(I, E)$, given that $TA_1$ and $TA_2$ are fixed in states I and E, respectively.

The transitions of $TA_1$ and $TA_2$ can be analyzed in the same manner as those of $TA_3$ and $TA_4$. Based on this, we conclude that the system has a unique absorbing state in its full dynamics, with $TA_1$, $TA_2$, $TA_3$, and $TA_4$ adopting the actions I, E, I, and E, respectively, and the TAs ultimately settling in their respective deepest states.

## 4 CONVERGENCE ANALYSIS OF THE OR OPERATOR

We assume the training samples for the OR operator are noise free (i.e., Eq. (5)), and are independently drawn from a distribution with full support over the four cases.

$$P(y = 1|x_1 = 1, x_2 = 1) = 1, P(y = 1|x_1 = 0, x_2 = 1) = 1, \tag{5}$$
$$P(y = 1|x_1 = 1, x_2 = 0) = 1, P(y = 0|x_1 = 0, x_2 = 0) = 1.$$

**Theorem 2.** *The clauses in a TM can almost surely learn the 2-bit OR logic given noise free training samples (shown in Eq. (5)) in infinite time, when $T \leq \lfloor \frac{m}{2} \rfloor$.*

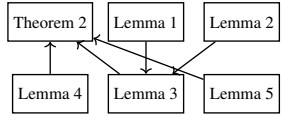

Figure 2: The dependence for the proof of Theorem 2.

The proof of the theorem requires Lemma 1-Lemma 5 and their dependence is shown in Fig. 2. Clearly, there are three sub-patterns for the OR operator. In Lemma 1, we will show that any clause is able to converge and absorb to an intended sub-pattern when the training sample of only one sub-pattern is given, and when $u_1 > 0$ and $u_2 > 0$. In Lemma 2, we will show that the TM will not absorb when more sub-patterns jointly appear in the training samples and when $u_1 > 0$ and $u_2 > 0$. These two lemmas will be utilized in the proof of Lemma 3. Lemma 2 also reveals the non-absorbing nature of TM for the OR operator when the functionality of $T$ is not enabled, i.e., when $u_1 > 0$ and $u_2 > 0$. This confirms the necessity of enabling the functionality of $T$ in order to converge to an absorbing state that fulfills the OR operator, to be indicated by Lemma 3-Lemma 5. Specifically, Lemma 3-Lemma 5 analyze the system behavior when $T$ is enabled and how $T$ should be configured for the TM to converge to the OR operator. They guarantee that when the system reaches an absorbing state, the intended sub-patterns will have a number of clauses no less than $T$ while the unintended sub-pattern will have 0 clause. Then the OR operator can be inferred by setting $Th = T$. In what follows, we will present and prove the lemmas.

**Lemma 1.** *For any one of the three sub-patterns resulting in $y = 1$, shown in Eqs. (6)-(8), the TM can converge to the intended sub-pattern when noise free training samples following this sub-pattern, together with samples satisfying $P(y = 0|x_1 = 0, x_2 = 0) = 1$, are provided, and when $u_1 > 0, u_2 > 0$.*

$$P(y = 1|x_1 = 1, x_2 = 1) = P(y = 0|x_1 = 0, x_2 = 0) = 1, \tag{6}$$
$$P(y = 1|x_1 = 0, x_2 = 1) = P(y = 0|x_1 = 0, x_2 = 0) = 1, \tag{7}$$
$$P(y = 1|x_1 = 1, x_2 = 0) = P(y = 0|x_1 = 0, x_2 = 0) = 1. \tag{8}$$

The proof of Lemma 1 involves demonstrating convergence for three sub-patterns: those governed by Eqs. (6), (7), and (8). These analyses build upon the convergence proofs for the XOR and AND operators. For the sub-pattern in Eq. (6), transition diagrams in Appendix B confirm that the TAs converge to $TA_1 = I$, $TA_2 = E$, $TA_3 = I$, and $TA_4 = E$, when input samples $[\boldsymbol{x}_1 = 0, \boldsymbol{x}_2 = 1]$ and $[\boldsymbol{x}_1 = 1, \boldsymbol{x}_2 = 0]$ are excluded. The other two sub-patterns are proven using similar principles. Full details are provided in Appendix C.

From Lemma 1, we show that the clauses converge to the intended sub-pattern when the training samples follow that specific sub-pattern. In contrast, Lemma 2 will show that the system becomes non-absorbing when training samples contain two or more sub-patterns. In particular, we prove that the TM is non-absorbing for samples following Eq. (5) and Eqs. (9)–(11) when $u_1 > 0$ and $u_2 > 0$.

$$P(y = 1|x_1 = 1, x_2 = 1) = P(y = 1|x_1 = 1, x_2 = 0) \tag{9}$$
$$= P(y = 0|x_1 = 0, x_2 = 0) = 1,$$
$$P(y = 1|x_1 = 1, x_2 = 1) = P(y = 1|x_1 = 0, x_2 = 1) \tag{10}$$
$$= P(y = 0|x_1 = 0, x_2 = 0) = 1,$$
$$P(y = 1|x_1 = 1, x_2 = 0) = P(y = 1|x_1 = 0, x_2 = 1) \tag{11}$$
$$= P(y = 0|x_1 = 0, x_2 = 0) = 1.$$

**Lemma 2.** *The TM becomes non-absorbing if any two or more of the three sub-patterns jointly appear in the training samples, as shown in Eqs. (5), (9)-(11), when $u_1 > 0, u_2 > 0$.*

The proof of Lemma 2 can be found in Appendix D. Lemma 2 tells us that if we always give TM the training samples from all sub-patterns without blocking the learnt patterns by using $T$ via Eqs. (3) and (4), the system is non-absorbing. In other words, if we want to have the TM converge to the OR operator in an absorbing state, it is critical to utilize the feature of $T$ to block any incoming training samples from updating the learnt sub-patterns. Specifically, we need to configure $T$ (1) so that the absorbing states exist and (2) confirm that the absorbing states follows the OR operator. In what follows, we will, through Lemmas 3-5, show how $T$ via Eqs. (3) and (4) can guarantee the convergence and how the value of $T$ should be configured.

Let's revisit the functionality of $T$. $T$ can block the training samples from updating a learnt sub-pattern. More specifically, if the number of the clauses reaches $T$ for a certain sub-pattern, the new training samples of this sub-pattern will be blocked by the TM. There are three sub-patterns in OR operator. When the number of clauses for each of the three sub-patterns reaches $T$, all training samples associated with Type I feedback are blocked. Simultaneously, if none of the samples for Type II feedback trigger any change to the states of the TAs, the TM reaches an absorbing state. In Lemma 3, we detail the necessity and sufficiency of the absorbing state.

**Lemma 3.** *The system is absorbed if and only if (1) the number of clauses for each intended sub-pattern reaches $T$, i.e., $f_\Sigma(\mathcal{C}^i(\mathbf{X})) = T$, $\forall \mathbf{X} = [x_1 = 0, x_2 = 1]$ or $[x_1 = 1, x_2 = 1]$ or $[x_1 = 1, x_2 = 0]$, and (2) no clause is formed only by a negated literal or negated literals.*

The proof of Lemma 3 can be found in Appendix E. In Lemma 3, we find the conditions of the absorbing state. In the next Lemma, we will show how to set up the value of $T$ so that the number of clauses for each intended sub-pattern can indeed reach $T$.

**Lemma 4.** $T \leq \lfloor m/2 \rfloor$ *is required so that the number of clauses for each intended sub-pattern can reach $T$.*

**Proof of Lemma 4:** There are three intended sub-patterns in the OR operator. Given $m$ clauses in total, to make sure each one has at least $T$ votes, we have $3T \leq m$. This requires $T \leq \lfloor m/3 \rfloor$ ($T$ is an integer). However, the nature of the OR operator offers the possibility to represent 2 sub-patterns jointly. For example, $T$ clauses in the form of $x_1$ will result in the number of clauses being $T$ for each of the following sub-patterns, i.e., $[x_1 = 1, x_2 = 0]$ and $[x_1 = 1, x_2 = 1]$. If there are other $T$ clauses representing the remaining sub-pattern, in total $2T$ clauses can garantee that each of the intended sub-patterns is represented by $T$ clauses. We thus have $T \leq \lfloor m/2 \rfloor$. Note that the fact that two sub-patterns can be jointly represented by one clause has been observed and confirmed in experiments shown in Appendix I.

When we have a smaller $T$, different sub-patterns may be represented by distinct clauses. However, when $T > \lfloor m/2 \rfloor$, there will always be one or two sub-patterns that cannot obtain a number of $T$ clauses to represent them. For this reason, the maximum integer value is $T = \lfloor m/2 \rfloor$. ∎

In Lemma 5, we show that the input sample $[x_1 = 0, x_2 = 0]$ will never cause the number of clauses associated with this unintended sub-pattern to reach or exceed $T$. This is to avoid any possible false positive upon input $[x_1 = 0, x_2 = 0]$ in testing.

**Lemma 5.** *When absorbing, the sample from the unintended sub-pattern, i.e., $[x_1 = 0, x_2 = 0]$, will never lead to the number of clauses representing this unintended sub-pattern becoming greater than or equal to $T$.*

**Proof of Lemma 5:** To have a positive output from $[x_1 = 0, x_2 = 0]$, the clause should be in the form of $C = \neg x_1$ or $C = \neg x_2$ or $C = \neg x_1 \wedge \neg x_2$. It has already shown in the proof of Lemma 3 that Type II feedback will eliminate such clauses. In fact, when the system is absorbed, no clause will be in the form of $C = \neg x_1$ or $C = \neg x_2$ or $C = \neg x_1 \wedge \neg x_2$. For this reason, $[x_1 = 0, x_2 = 0]$ will never lead to the number of clauses greater than or equal to $T$. ∎

**Proof of Theorem 2:** Based on Lemmas 3–5, we understand that if $T \leq \lfloor m/2 \rfloor$ holds, Type I feedback will eventually be blocked and Type II feedback will eventually only give "inaction" feedback. In this situation, no actual transition will be triggered and thus the system reaches the absorbing state. Before absorbed, the system moves back and forth in the intermediate states. Once absorbed, any one of the intended sub-patterns will have the number of clauses for that sub-pattern no less than $T$ and the unintended sub-pattern will have 0 clauses. We thus have the OR logic almost surely by setting a threshold $Th = T$ and conclude the proof. ∎

Now let's study a simple example with $m = 2$, $T = 1$. Here, $C_1 = x_1$ and $C_2 = x_2$ can be an instance for an absorbing case. $C_1 = x_1$ and $C_2 = \neg x_1 \wedge x_2$ also works. Clearly, the clauses can be in various forms, as long as the conditions in Lemma 3 fulfill. These converged clauses are not necessarily in the exact form of the three sub-patterns, which is distinct to that of the XOR operator.

**Remark 1.** *Although both AND and OR operators converge, the approaches are different. For AND operator, the system is converged because the clauses become eventually absorbed to the intended pattern upon Type I and Type II feedback, even if the functionality of $T$ is disabled ($u_1 > 0$ and $u_2 > 0$). As the TM enables the functionality of $T$ by default, the system will be absorbed when $T$ clauses converge to $x_1 \wedge x_2$, before all clauses converge to this pattern. However, for the OR operator, the functionality of $T$ is critical because the TM is non-absorbing if $u_1 > 0$ and $u_2 > 0$. The absorbing state of the OR operator is achieved because the functionality of $T$ blocks all Type I feedback and Type II feedback gives only "Inaction" feedback. The concept of convergence for the OR operator is similar to that of XOR, but the form of clauses after absorbing varies due to the possible joint representation of sub-patterns in OR.*

**Remark 2.** *When $T$ is greater than half the number of clauses, i.e., $T > \lfloor m/2 \rfloor$, the system will not have an absorbing state. We conjecture that the system can still learn the sub-patterns in an unbalanced manner, as long as $T$ is not configured too close to the total number of clauses $m$.*

Given $T > \lfloor m/2 \rfloor$, Type I feedback cannot be completely blocked and the TM is non-absorbing. Nevertheless, if $T$ is not close to $m$, there will be clauses that possibly learn distinct sub-patterns. In addition, Type II feedback can avoid the form of $C = \neg x_1$ or $C = \neg x_2$ or $C = \neg x_1 \wedge \neg x_2$ from happening. Therefore, with $Th > 0$, the TM may still learn the OR operator with high probability.

## 5 REVISIT THE XOR OPERATOR

Let us revisit the proof of XOR operator. As stated in (Jiao et al., 2023), when the system is absorbed, the clauses follow the format $C = x_1 \wedge \neg x_2$ or $C = \neg x_1 \wedge x_2$ precisely. In other words, a clause with just one literal, such as $C = x_1$, cannot absorb the system. The reason is that the sub-patterns in XOR operator are mutual exclusive, i.e., the sub-patterns cannot be merged in any way. Although Type I feedback can be blocked when $T$ clauses represent one sub-pattern using one literal, the Type II feedback can force the other missing literal to be included. For example, when $T$ clauses happens to converge to $C = x_1$, the Type I feedback from any input samples of ($[x_1 = 1, x_2 = 0], y = 1$) will be blocked. In this situation, the Type II feedback from ($[x_1 = 1, x_2 = 1], y = 0$) will encourage the clause to include $\neg x_2$. This is because upon a sample ($[x_1 = 1, x_2 = 1], y = 0$), we have Type II feedback, $C = x_1 = 1$, and the studied literal is $\neg x_2 = 0$. When the TA for excluding $\neg x_2$ is considered, a large penalty, i.e., a penalty in probability 1, is given to the TA, moving it towards action *Include*, and thus $C = x_1$ eventually becomes $C = x_1 \wedge \neg x_2$. Following the same concept, we can analyze the development for $C = \neg x_1$, $C = x_2$, and $C = \neg x_2$, which will eventually converge to $C = \neg x_1 \wedge x_2$ or $C = x_1 \wedge \neg x_2$, upon Type II feedback.

## 6 CONVERGENCE ANALYSIS UNDER RANDOM NOISE

We studied the convergence properties of AND, OR, and XOR operators under training samples with noise. The noise type is *noisy completely at random* (Frénay & Verleysen, 2013), categorized as wrong labels and irrelevant input variables. A wrong label refers to an input that should be labeled as 1 but is instead labeled as 0, or vice versa. An irrelevant input variable, on the other hand, is one that does not contribute to the classification. We demonstrate that, with wrong labels, the TM does not converge to the intended operators but can still learn efficiently. With irrelevant variables, the TM converges to the intended operators almost surely. Experimental results confirmed these findings (Appendix J). We summarize the main findings in this section. The proof details can be found in Appendix F and Appendix G.

**Theorem 3.** *The TM is non-absorbing given training samples with wrong labels for the AND, OR, and XOR operators.*

**Remark 3.** *The non-absorbing property of TM indicates that there is a non-zero probability that it cannot learn the intended operator. The primary reason for the non-absorbing behavior when wrong labels are present is the statistically conflicting labels for the same input samples. These*

*inconsistency causes the TAs within a clause to learn conflicting outcomes for the same input. When a clause learns to evaluate an input as 1 based on Type I feedback, samples with a label of 0 for the same input prompt it to learn the opposite. This conflict in labels confuses the TM, leading to back-and-forth learning.*

**Remark 4.** *Although wrong labels will make the TM not converge (not absorbing with 100% accuracy for the intended logic), via experiments, we can still find that the TM are able to learn the operators efficiently, shown in Appendix J. This property aligns with the concept of PAC learnable (Mansour & Parnas, 1998) or $\epsilon$-optimality (Zhang et al., 2020), although a formal proof remains open.*

**Theorem 4.** *The clauses in a TM can almost surely learn the 2-bit AND logic given training samples with $q$ irrelevant input variables in infinite time, $q > 0$, when $T \leq m$.*

**Theorem 5.** *The clauses in a TM can almost surely learn the 2-bit XOR and OR logic given training samples with $q$ irrelevant input variables in infinite time, $q > 0$, when $T \leq \lfloor m/2 \rfloor$.*

The proofs of Theorems 4 and 5 follow the same underlying methodology (see Appendix G). We identify the conditions under which the TM becomes absorbed, and verify that the absorbing states correspond to the intended sub-pattern(s), and no other absorbing states exist. From these proofs, it becomes clear that $T$ is critical for convergence. The presence of irrelevant bits can make the TM non-absorbing if $T$ is not functioning, whereas an appropriate configuration of $T$ guarantees convergence to the correct intended sub-patterns.

**Remark 5.** *An interesting observation is that the TM does not always exclude all irrelevant literals. Our analysis and experiments reveal two distinct mechanisms through which TMs exhibit robustness to irrelevant bits. When sufficient clause resources are available, the $T$ clauses assigned to a sub-pattern may include irrelevant bits while another $T$ clauses include their negations, yet both sets vote for the same target sub-pattern, effectively canceling out their influence. When clause resources are limited, irrelevant bits tend to be excluded, and therefore do not affect the classification outcome.*

## 7 CONVERGENCE ANALYSIS FOR $k$-BIT CASE

The analyses above focus on the 2-bit cases. In this section, we extend the results to the general $k$-bit setting, where $k > 2$. Since the 2-bit analyses rely heavily on exhaustive search, increasing the number of bits immediately leads to a combinatorial explosion. To avoid this, we go from literal level to clause level, by clustering the clause representations into three categories. By analyzing the transition properties among these categories, rather than the literal states, we can demonstrate the convergence behavior without being hindered by the exponential growth. We first present the main theorems, followed by an outline of the proof. The full proofs are provided in Appendix H.

We begin with the noise-free case. For the $k$-bit setting, the convergence analysis naturally splits into two subcategories: cases with a single sub-pattern (analogous to the AND operator in the 2-bit case) and cases with multiple sub-patterns (2 or more sub-patterns exist, analogous to OR or XOR in the 2-bit case). Formally, the single–sub-pattern category corresponds to the existence of a unique sub-pattern among the $2^k$ possible input combinations that is labeled as 1, while all others are labeled as 0 or remain undefined (where "undefined" means unlabeled). In contrast, the multiple–sub-pattern category includes scenarios where more than one sub-pattern is labeled as 1.

**Theorem 6.** *In the $k$-bit single sub-pattern category, any clause will converge almost surely to the intended sub-pattern given noise free training samples in infinite time when $u_1 > 0$ and $u_2 > 0$.*

To prove Theorem 6, instead of examining all possible states of literals, we group the clause forms into three categories and summarize their possible transitions. The clause forms are defined as follows: (1) **Exact match:** The clause matches the intended sub-pattern exactly. A clause in this form outputs 1 when the intended sub-pattern is presented (e.g., $x_1 \wedge x_2$ for the AND operator). (2) **Partial match:** The clause does not fully match the intended sub-pattern but matches a subset of it. Such a clause also outputs 1 for the intended sub-pattern (e.g., $x_1$ in the AND case). (3) **Non-match:** The clause matches neither the intended sub-pattern nor any subset of it. A clause in this category outputs 0 when the intended sub-pattern is given (e.g., $\neg x_1$ for the AND case). The proof shows that once the TM reaches a clause of type (1), the system becomes absorbed, whereas (2) and (3) do not. This guarantees the existence of a unique absorbing clause that represents the intended sub-pattern.

**Theorem 7.** *The clauses in a TM can almost surely learn the $k$-bit multiple–sub-pattern logic from noise-free training samples in infinite time, provided that $T \leq \lfloor m/e \rfloor$, where $e$ is the number of sub-pattern clusters.*

In Theorem 7, a *sub-pattern cluster* is defined as a group of sub-patterns that share one or more common 1s in their corresponding input bits. For example, in the OR case, $(0, 1)$ and $(1, 1)$ belong to the same cluster because they share a 1 in $x_2$. In contrast, $(1, 0)$ and $(0, 1)$ do not belong to the same cluster, as they do not share any common 1s. We introduce the notion of sub-pattern clusters because sub-patterns within the same cluster can potentially be represented jointly by a clause that learns their shared feature. The proof of Theorem 7 follows the same structure as the proof of Theorem 2. We first show that when multiple sub-patterns are present, the system becomes non-absorbing if $u_1 > 0$ and $u_2 > 0$. Then, by configuring the hyperparameter $T$ appropriately, we can suppress further feedback once the clauses have captured all individual sub-patterns, thereby ensuring convergence.

With irrelevant bits, we have the following conclusions.

**Theorem 8.** *The clauses in a TM can almost surely learn the $k$-bit single sub-pattern logic given training samples with $q$ irrelevant input variables in infinite time, $k \geq 2$, $q > 0$, when $T \leq m$.*

**Theorem 9.** *Consider training samples of fixed length $n$, and a set of sub-patterns indexed by $i$, where the $i$-th sub-pattern contains $k_i \geq 2$ informative bits and $q_i = n - k_i > 0$ irrelevant bits. A TM can almost surely learn all such multi-sub-pattern logic in infinite time when $T \leq \lfloor m/e \rfloor$.*

The proofs of Theorems 8 and 9 follow the same concept as those of Theorems 4 and 5. Experimental insights of the convergence of the $k$-bit cases can be found in Appendix K.

**Remark 6.** *By moving from the restrictive 2-bit setting to a $k$-bit formulation, the results now capture the actual operating regime of general TM applications. Since practical TM systems universally rely on a preprocessing to transform any input data into booleanized $k$-bit feature representations, the generalized theory developed here provides the first principled explanation of the mechanism that govern the TM behavior in practical settings. Consequently, the theoretical findings presented here offer a broadly applicable explanation for the convergence properties and performance patterns repeatedly observed across diverse prior empirical studies.*

## 8 INSIGHTS FOR PRACTICAL USAGE

We summarize here the insights from the proofs that are useful for the practical application of the TM. First, the hyperparameter $T$ plays a crucial role. If one can estimate the number of sub-patterns and their clustering structure in a classification task, it becomes easier to select a good initial value for $T$, reducing tuning effort and improving convergence. Second, while joint learning of sub-patterns enables clauses to capture concepts more compactly, it also means that individual sub-patterns may not appear explicitly, which can hinder interpretability since a single clause may represent several sub-patterns simultaneously. Third, our robustness analysis shows that clauses can accumulate irrelevant literals when many clauses are used, adding further interpretability challenges. For applications where transparency is essential, limiting clause length or the number of clauses may therefore be beneficial. To demonstrate practical relevance, we also include real-world TM applications on publicly available benchmark datasets in Appendix L.

## 9 CONCLUSIONS

This work establishes a comprehensive theoretical framework for understanding the convergence behavior of the TM. By extending prior results from the 1-bit setting to the 2-bit AND, OR, and XOR operators, and further to the general $k$-bit case, we demonstrate that the TM reliably converges under noise-free training and irrelevant variables. The analysis highlights the critical role of $T$ in both multi–sub-pattern scenarios and those involving irrelevant bits. The analysis also reveals structural properties unique to certain operators, such as the ability of OR clauses to jointly encode multiple sub-patterns. These insights not only clarify the learning dynamics of the TM but also provide practical guidance for model design, hyper-parameter selection, and interpretability. Collectively, the results reinforce the TM as a theoretically grounded and practically effective approach to interpretable machine learning.

ACKNOWLEDGMENT

This work was partially supported by NORCE Research and by the project *Spacetime Vision: Towards Unsupervised Learning in the 4D World*, financed by the EEA and Norway Grants 2014–2021 under Grant No. EEA-RO-NO-2018-04. It was also partially supported by the Research Council of Norway (NFR) through the project *CaReLearner – Causal Reasoning with Logical Interpretable Learning* (Project No. 335700).

The authors would like to thank the two reviewers for their valuable feedback. One reviewer provided critical comments that motivated us to extend the work to noisy cases. The other offered insightful suggestions that led us to generalize the proof from 2 bits to $k$ bits. Their feedback significantly improved the quality and scope of this paper.

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

# A    BRIEF OVERVIEW OF THE TM

We present the basics of TM here. Those who already are familiar with the concept and notations of TM can ignore this appendix.

## A.1    BASIC CONCEPT OF THE TM

The input to a TM is denoted by $\mathbf{X} = [x_1, x_2, \ldots, x_o]$, where $x_k \in \{0, 1\}$ for $k = 1, 2, \ldots, o$, and $o$ denotes the number of input features. For each input $x_k$, two literals are defined: the literal $x_k$ and its negation $\neg x_k$. Consequently, each clause is associated with a total of $2o$ literals. A clause is defined as a conjunction of a subset of these literals, to be explained later.

Each literal is controlled by an associated TA. The TA is a two-action learning automaton that decides whether its corresponding literal should be *included in* or *excluded from* the clause. This decision is determined by the current internal state of the TA. Figure 3 illustrates the structure of a TA with $2N$ states, where $N$ is the number of states for each action. This study considers $N$ as a finite number, which is practical for real-world applications. When the TA is in any state between $0$ to $N - 1$, the action "Include" is selected. The action becomes "Exclude" when the TA is in any state between $N$ to $2N - 1$. The transitions among the states are triggered by a reward or a penalty that the TA receives from the environment, which, in this case, is determined by different types of feedback defined in the TM (to be explained later). A larger $N$ expands the depth of the TA's action-state space, enhancing its robustness. This benefit, however, comes at the cost of longer convergence times to innermost states and greater memory requirements.

A clause is associated with $2o$ TAs, forming a TA team. A TA team is denoted in general as $\mathcal{G}_j^i = \{\text{TA}_{k'}^{i,j} | 1 \leq k' \leq 2o\}$, where $k'$ is the index of the TA, $j$ is the index of the TA team/clause (multiple TA teams form a TM), and $i$ is the index of the TM/class to be identified (A TM identifies a class, multiple TMs identify multiple classes).

Suppose we are investigating the $i^{th}$ TM whose job is to identify class $i$, and that the TM is composed of $m$ TA teams. Then $C_j^i(\mathbf{X})$ can be used to denote the output of the $j^{th}$ TA team, which is a conjunctive clause.
For training:

$$C_j^i(\mathbf{X}) = \begin{cases} \left( \bigwedge_{k \in \xi_j^i} x_k \right) \wedge \left( \bigwedge_{k \in \bar{\xi}_j^i} \neg x_k \right), & \text{for } \xi_j^i, \bar{\xi}_j^i \neq \emptyset, \\ 1, & \text{for } \xi_j^i, \bar{\xi}_j^i = \emptyset. \end{cases} \tag{12}$$

For inference:

$$C_j^i(\mathbf{X}) = \begin{cases} \left( \bigwedge_{k \in \xi_j^i} x_k \right) \wedge \left( \bigwedge_{k \in \bar{\xi}_j^i} \neg x_k \right), & \text{for } \xi_j^i, \bar{\xi}_j^i \neq \emptyset, \\ 0, & \text{for } \xi_j^i, \bar{\xi}_j^i = \emptyset. \end{cases} \tag{13}$$

In Eqs. (12) and (13), $\xi_j^i$ and $\bar{\xi}_j^i$ are defined as the sets of indexes for the literals that have been included in the clause. $\xi_j^i$ contains the indexes of included original (non-negated) inputs, $x_k$, whereas $\bar{\xi}_j^i$ contains the indexes of included negated inputs, $\neg x_k$. $\xi_j^i, \bar{\xi}_j^i = \emptyset$ means not a single literal (a feature or its negation) is included in the clause. Note that in propositional logic, an empty clause is typically defined as having a value of 1. However, empirical results indicate that TMs generally achieve higher test accuracy on new data when empty clauses are 0-valued. Therefore, during TM

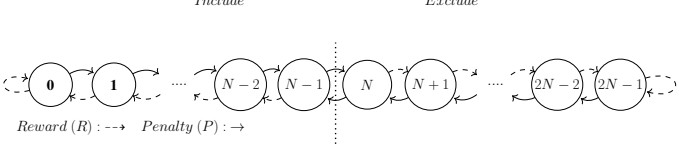

Figure 3: A two-action Tsetlin automaton with $2N$ states (Jiao et al., 2023).

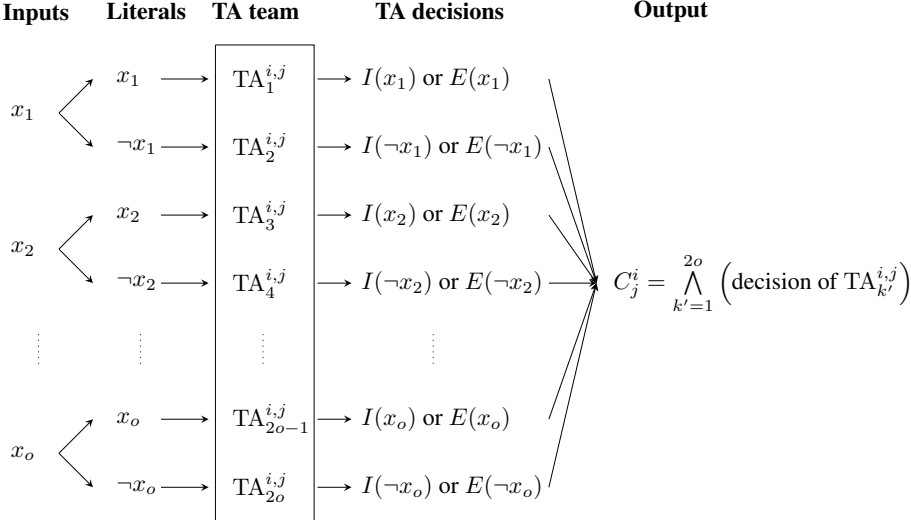

Figure 4: A TA team $G_j^i$ consisting of $2o$ TAs (Zhang et al., 2022). Here $I(x_1)$ means "include $x_1$" and $E(x_1)$ means "exclude $x_1$".

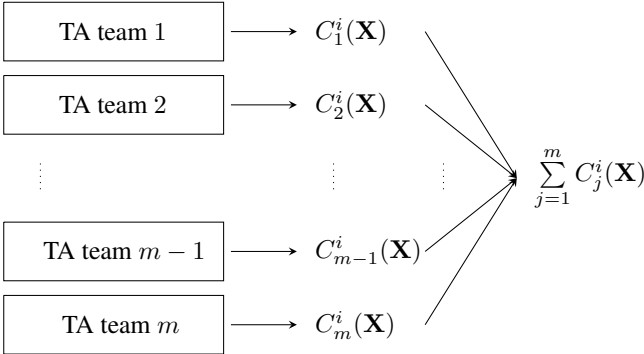

Figure 5: TM voting architecture (Jiao et al., 2023).

training, an "empty" clause outputs 1 to encourage the TAs to include literals, following the feedback mechanisms of the TM. In contrast, during TM inference, an "empty" clause outputs 0, indicating that it does not influence the final classification decision since it does not represent any specific sub-pattern.

Figure 4 illustrates the structure of a clause and its relationship to its literals. Here, for ease of notation, we define $I(x) = x$, $I(\neg x) = \neg x$, and $E(x) = E(\neg x) = 1$ in the analysis of the training procedure, with the latter meaning that an excluded literal does not contribute to the output.

Multiple clauses, i.e., the TA teams, each of which in conjunctive form, are assembled into a complete TM. There are two architectures for clause assembling: Disjunctive Normal Form Architecture and Voting Architecture. In this study, we focus on the latter one, as shown in Figure 5. The voting consists of summing the outputs of the clauses:

$$f_{\sum}(\mathcal{C}^i(\mathbf{X})) = \sum_{j=1}^m C_j^i(\mathbf{X}), \tag{14}$$

where $\mathcal{C}^i(\mathbf{X})$ is the set of trained clauses for class $i$.

The output of the TM, in turn, is decided by the unit step function:

$$\hat{y}^i = \begin{cases} 0 & \text{for } f_{\sum}(\mathcal{C}^i(\mathbf{X})) < Th, \\ 1 & \text{for } f_{\sum}(\mathcal{C}^i(\mathbf{X})) \geq Th, \end{cases} \tag{15}$$

where $Th$ is a predefined threshold for classification. For example, the classifier $(x_1 \wedge \neg x_2) + (\neg x_1 \wedge x_2)$ captures the XOR-relation given $Th = 1$, meaning if any sub-pattern is satisfied, the input will be identified as following the XOR logic.

Note that for the voting architecture, the TM can assign polarity to each TA team (Granmo, 2018). Specifically, TA teams with odd indices have positive polarity, learning from training samples with label 1, while those with even indices have negative polarity, learning from training samples with label 0. The only difference between these polarities is that the output of a clause associated with an even-indexed TA team will be flipped to its negative. The voting consists of summing the polarized clause outputs, and the threshold $Th$ is set to zero. For example, for the XOR operator with four clauses, the learned clauses with positive polarity can be $C_1 = x_1 \wedge \neg x_2$ and $C_3 = \neg x_1 \wedge x_2$, while the ones with negative polarity can be $C_2 = x_1 \wedge x_2$ and $C_4 = \neg x_1 \wedge \neg x_2$. In this case, when the testing sample $[x_1 = 1, x_2 = 0]$ arrives, the sum of the clause values is 1. On the contrary, when the testing sample $[x_1 = 0, x_2 = 0]$ arrives, the sum of the clause values is $-1$. In this way, with $Th = 0$, the system's decision range and tolerance is expected to be larger.

In this study, we consider only positive-polarity clauses for two reasons. First, the learning and reasoning process of the TM can be completely explained from the perspective of learning patterns that output 1, and negative-polarity clauses, which learn patterns that output 0, follow the same procedure. Second, this simplification offers easier analysis and better understanding.

## A.2 TRAINING PROCESS OF THE TM

The training process is built on letting all the TAs take part in a decentralized game. Training data $(\mathbf{X} = [x_1, x_2, ..., x_o], \ y^i)$ is obtained from a data set $\mathcal{S}$, distributed according to the probability distribution $P(\mathbf{X}, y^i)$. In the game, each TA is guided by Type I Feedback and Type II Feedback defined in Table 3 and Table 4, respectively. Type I Feedback is triggered when the training sample has a positive label, i.e., $y^i = 1$, meaning that the sample belongs to class $i$. When the training sample is labeled as not belonging to class $i$, i.e., $y^i = 0$, Type II Feedback is utilized for generating feedback. Examples demonstrating TA state transitions per feedback tables can be found in Section 3.1 in (Zhang et al., 2022). In brief, Type I feedback is to reinforce true positive and Type II feedback is to fight against false negative.

The hyperparameter $s$ controls the granularity of the clauses and a larger $s$ encourages more literals to be included in each clause, which also accelerates convergence and improves stability, but at the cost of an increased risk of overfitting in practice. Smaller $s$ generally leads to shorter clauses and slower convergence in practice. A more detailed analysis on hyperparameters $s$ and $N$ can be found in (Zhang et al., 2022).

| *Value of the clause $C_j^i(\mathbf{X})$* | | 1 | | 0 | |
|---|---|---|---|---|---|
| *Value of the Literal $x_k/\neg x_k$* | | 1 | 0 | 1 | 0 |
| **Include Literal** | $P(\text{Reward})$ | $\frac{s-1}{s}$ | NA | 0 | 0 |
| | $P(\text{Inaction})$ | $\frac{1}{s}$ | NA | $\frac{s-1}{s}$ | $\frac{s-1}{s}$ |
| | $P(\text{Penalty})$ | 0 | NA | $\frac{1}{s}$ | $\frac{1}{s}$ |
| **Exclude Literal** | $P(\text{Reward})$ | 0 | $\frac{1}{s}$ | $\frac{1}{s}$ | $\frac{1}{s}$ |
| | $P(\text{Inaction})$ | $\frac{1}{s}$ | $\frac{s-1}{s}$ | $\frac{s-1}{s}$ | $\frac{s-1}{s}$ |
| | $P(\text{Penalty})$ | $\frac{s-1}{s}$ | 0 | 0 | 0 |

Table 3: Type I Feedback — Feedback upon receiving a sample with label $y = 1$, for a single TA to decide whether to Include or Exclude a given literal $x_k/\neg x_k$ into $C_j^i$. NA means not applicable (Granmo, 2018).

| Value of the clause $C_j^i(\mathbf{X})$ | | 1 | | 0 | |
|---|---|---|---|---|---|
| Value of the Literal $x_k/\neg x_k$ | | 1 | 0 | 1 | 0 |
| **Include Literal** | $P(\text{Reward})$ | 0 | NA | 0 | 0 |
| | $P(\text{Inaction})$ | 1.0 | NA | 1.0 | 1.0 |
| | $P(\text{Penalty})$ | 0 | NA | 0 | 0 |
| **Exclude Literal** | $P(\text{Reward})$ | 0 | 0 | 0 | 0 |
| | $P(\text{Inaction})$ | 1.0 | 0 | 1.0 | 1.0 |
| | $P(\text{Penalty})$ | 0 | 1.0 | 0 | 0 |

Table 4: Type II Feedback — Feedback upon receiving a sample with label $y = 0$, for a single TA to decide whether to Include or Exclude a given literal $x_k/\neg x_k$ into $C_j^i$. NA means not applicable (Granmo, 2018).

To avoid the situation that a majority of the TA teams learn only one sub-pattern (or a subset of sub-patterns) while ignore other sub-patterns, forming an incomplete representation[3], the hyperparameter $T$ is used to regulate the resource allocation. If the votes, i.e., the summation $f_{\sum}(\mathcal{C}^i(\mathbf{X}))$, for a certain sub-pattern $\mathbf{X}$ already reach a total of $T$ or more, neither rewards nor penalties are provided to the TAs when more training samples of this particular sub-pattern are given. In this way, we can ensure that each specific sub-pattern can be captured by a limited number, i.e., $T$, of available clauses, allowing sparse sub-pattern representations among competing sub-patterns. Formally, the strategy works as follows:

**Generating Type I Feedback.** If the label of the training sample $\mathbf{X}$ is $y^i = 1$, we generate, in probability, *Type I Feedback* to each clause $C_j^i \in \mathcal{C}^i$. The probability of generating Type I Feedback is (Granmo, 2018):

$$u_1 = \frac{T - \max(-T, \min(T, f_{\sum}(\mathcal{C}^i(\mathbf{X}))))}{2T}. \tag{16}$$

**Generating Type II Feedback.** If the lable of the training sample $\mathbf{X}$ is $y^i = 0$, we generate, again, in probability, *Type II Feedback* to each clause $C_j^i \in \mathcal{C}^i$. The probability is (Granmo, 2018):

$$u_2 = \frac{T + \max(-T, \min(T, f_{\sum}(\mathcal{C}^i(\mathbf{X}))))}{2T}. \tag{17}$$

After Type I Feedback or Type II Feedback is generated for a clause, each individual TA within each clause is given a reward/penalty/inaction according to the probability defined in the Type I and Type II feedback tables, and then the state of the corresponding TA is updated.

---

[3]For example, for the OR operator, one should avoid the situation that a majority of TA teams converge to $\neg x_1 \wedge x_2$ to represent the sub-pattern of $[0, 1]$, and ignore the other sub-patterns $[1, 0]$ and $[1, 1]$, making the learning outcome biased/unbalanced. A proper configuration of $T$ can avoid this situation.

## B  DETAILED PROOF OF THE CONVERGENCE OF THE AND OPERATOR

**Proof:** In this Appendix, we will prove Theorem 1. The condition $u_1 > 0$ and $u_2 > 0$ guarantees that all types of samples for AND operator, following Eq. (18), are always given and no specific type is blocked during training. The goal of the proof is to show that the system transitions will guarantee the actions of $TA_1$, $TA_2$, $TA_3$, and $TA_4$ to be I, E, I, E, and these actions correspond to the unique absorbing state of the system.

$$P\left(y = 1 | x_1 = 1, x_2 = 1\right) = 1,$$ 

$$P\left(y = 0 | x_1 = 0, x_2 = 1\right) = 1,$$

$$P\left(y = 0 | x_1 = 1, x_2 = 0\right) = 1,$$

$$P\left(y = 0 | x_1 = 0, x_2 = 0\right) = 1.$$

(18)

In Subsections B.1, we will describe the transitions of the system in an exhaustive manner. Thereafter, in the Subsection B.2, we summarize the transitions in Subsection B.1 and reveal the absorbing state of the system, which is the intended AND operator.

### B.1  THE TRANSITIONS OF THE TAS

In order to analyze the transitions of the system, we freeze the transition of the two TAs for the first bit of the input and study the transition of the second bit of input. Clearly, there are four cases for the first bit, $x_1$, as:

- Case 1: $TA_1$ = E, $TA_2$ = I, i.e., include $\neg x_1$.
- Case 2: $TA_1$ = I, $TA_2$ = E, i.e., include $x_1$.
- Case 3: $TA_1$ = E, $TA_2$ = E, i.e., exclude both $x_1$ and $\neg x_1$.
- Case 4: $TA_1$ = I, $TA_2$ = I, i.e., include both $x_1$ and $\neg x_1$.

In what follows, we will analyze the transition of the TAs for $x_2$, given the TAs of $x_1$ frozen in the above four distinct cases, one by one.

#### B.1.1  CASE 1: INCLUDE $\neg x_1$

In this subsection, we assume that the TAs for first bit is frozen as $TA_1$ = E and $TA_2$ = I, and thus the overall joint actions of TAs for the first bit give "$\neg x_1$". In this case, we have 4 situations to study, detailed below:

- Situation1: We study the transition of $TA_3$ when it has "Include" as its current action, given different actions of $TA_4$ (i.e., when the action of $TA_4$ is frozen as "Include" or "Exclude").

- Situation 2: We study the transition of $TA_3$ when it has "Exclude" as its current action, given different actions of $TA_4$ (i.e., when the action of $TA_4$ is frozen as "Include" or "Exclude").

- Situation 3: We study the transition of $TA_4$ when it has "Include" as its current action, given different actions of $TA_3$ (i.e., when the action of $TA_3$ is frozen as "Include" or "Exclude").

- Situation 4: We study the transition of $TA_4$ when it has "Exclude" as its current action, given different actions of $TA_3$ (i.e., when the action of $TA_3$ is frozen as "Include" or "Exclude").

In what follows, we will go through, exhaustively, the four situations.

*B.1.1.1  Study $TA_3$ with Action Include*

Here we study the transitions of $TA_3$ when its current action is *Include*, given different actions of $TA_4$ and input samples. For ease of expressions, the self-loops of the transitions are not depicted in the transition diagram. Clearly, this situation has eight instances, depending on the variations of

the training samples and the status of $\text{TA}_4$, where the first four correspond to the instances with $\text{TA}_4 = \text{E}$ while the remaining four represent the instances with $\text{TA}_4 = \text{I}$.

Now we study the first instance, with $x_1 = 1$, $x_2 = 1$, $y = 1$, and $\text{TA}_4 = \text{E}$. Clearly, this training sample will trigger Type I feedback because $y = 1$. Together with the current status of the other TAs, the clause is determined to be $C = \neg x_1 \wedge x_2 = 0$ and the literal is $x_2 = 1$. From Table 3, we know that the penalty probability is $\frac{1}{s}$ and the inaction probability is $\frac{s-1}{s}$. To indicate the transitions, we have plotted the diagram, with the transitions for penalty below. Note that the overall transition probability is $u_1 \frac{1}{s}$, where $u_1$ is defined in Eq. (3). Here, we have assumed $u_1 > 0$.

Condition: $x_1 = 1$, $x_2 = 1$, $y = 1$,
$\text{TA}_4 = \text{E}$.
Thus, Type I, $x_2 = 1$,
$C = \neg x_1 \wedge x_2 = 0$.

We here continue with analyzing another example shown below. In this instance, it covers the training samples: $x_1 = 1$, $x_2 = 0$, $y = 0$, and $\text{TA}_4 = \text{E}$. Clearly, the training sample will trigger Type II feedback because $y = 0$. The clause output becomes $C_3 = \neg x_1 \wedge x_2 = 0$. Because we now study $\text{TA}_3$, the corresponding literal is $x_2 = 0$. Based on the information above, we can check from Table 4 and find the probability of "Inaction" is 1. For this reason, the transition diagram does not have any arrow, indicating that there is "No transition" for $\text{TA}_3$.

Condition: $x_1 = 1$, $x_2 = 0$, $y = 0$,
$\text{TA}_4 = \text{E}$.
Thus, Type II, $x_2 = 0$,
$C = \neg x_1 \wedge x_2 = 0$.

No transition

The same analytical principle applies to all the other instances, and we therefore will not explain them in detail. Instead, we just list the transition diagrams.

Condition: $x_1 = 0$, $x_2 = 1$, $y = 0$,
$\text{TA}_4 = \text{E}$.
Thus, Type II, $x_2 = 1$,
$C = \neg x_1 \wedge x_2 = 1$.

No transition

Condition: $x_1 = 0$, $x_2 = 0$, $y = 0$,
$\text{TA}_4 = \text{E}$.
Thus, Type II, $x_2 = 0$,
$C = \neg x_1 \wedge x_2 = 0$.

No transition

Condition: $x_1 = 1$, $x_2 = 1$, $y = 1$,
$\text{TA}_4 = \text{I}$.
Thus, Type I, $x_2 = 1$,
$C = \neg x_1 \wedge x_2 \wedge \neg x_2 = 0$.

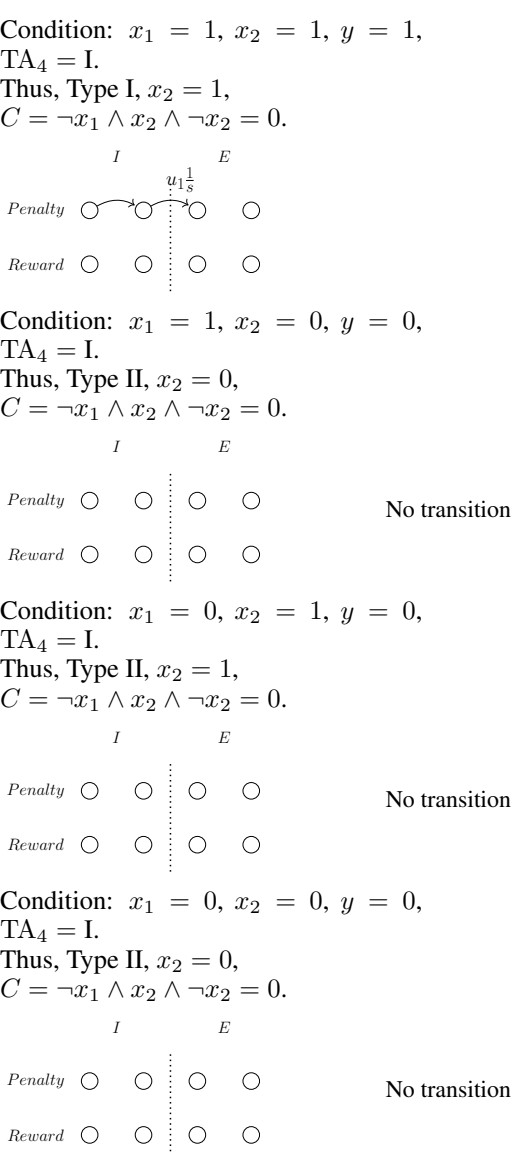

Condition: $x_1 = 1$, $x_2 = 0$, $y = 0$,
$\text{TA}_4 = \text{I}$.
Thus, Type II, $x_2 = 0$,
$C = \neg x_1 \wedge x_2 \wedge \neg x_2 = 0$.

No transition

Condition: $x_1 = 0$, $x_2 = 1$, $y = 0$,
$\text{TA}_4 = \text{I}$.
Thus, Type II, $x_2 = 1$,
$C = \neg x_1 \wedge x_2 \wedge \neg x_2 = 0$.

No transition

Condition: $x_1 = 0$, $x_2 = 0$, $y = 0$,
$\text{TA}_4 = \text{I}$.
Thus, Type II, $x_2 = 0$,
$C = \neg x_1 \wedge x_2 \wedge \neg x_2 = 0$.

No transition

### B.1.1.2 Study $\text{TA}_3$ with Action Exclude

Here we study the transitions of $\text{TA}_3$ when its current action is *Exclude*, given different actions of $\text{TA}_4$ and input samples. This situation has eight instances, depending on the variations of the training samples and the status of $\text{TA}_4$. In this subsection and the following subsections, we will not plot the transition diagrams for "No transition".

Condition: $x_1 = 1$, $x_2 = 1$, $y = 1$,
$\text{TA}_4 = \text{E}$.
Thus, Type I, $x_2 = 1$,
$C = \neg x_1 = 0$.

Condition: $x_1 = 0$, $x_2 = 0$, $y = 0$,
$\text{TA}_4 = \text{E}$.
Thus, Type II, $x_2 = 0$,
$C = \neg x_1 = 1$.

Condition: $x_1 = 1$, $x_2 = 1$, $y = 1$,
$\text{TA}_4 = \text{I}$.
Thus, Type I, $x_2 = 1$,
$C = \neg x_1 \wedge \neg x_2 = 0$.

Condition: $x_1 = 0$, $x_2 = 0$, $y = 0$,
$\text{TA}_4 = \text{I}$.
Thus, Type II, $x_2 = 0$,
$C = \neg x_1 \wedge \neg x_2 = 1$.

### B.1.1.3 Study $\text{TA}_4$ with Action Include

Here we list the transitions for $\text{TA}_4$ when its current action is *Include*.

Condition: $x_1 = 1$, $x_2 = 1$, $y = 1$,
$\text{TA}_3 = \text{E}$.
Thus, Type I, $\neg x_2 = 0$,
$C = \neg x_1 \wedge \neg x_2 = 0$.

Condition: $x_1 = 1$, $x_2 = 1$, $y = 1$,
$\text{TA}_3 = \text{I}$.
Thus, Type I, $\neg x_2 = 0$,
$C = \neg x_1 \wedge x_2 \wedge \neg x_2 = 0$.

### B.1.1.4 Study $\text{TA}_4$ with Action Exclude

Here we list the transitions for $\text{TA}_4$ when its current action is *Exclude*.

Condition: $x_1 = 1$, $x_2 = 1$, $y = 1$,
$TA_3 = E$.
Thus, Type I, $\neg x_2 = 0$,
$C = \neg x_1 = 0$.

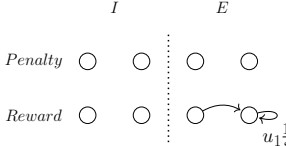

Condition: $x_1 = 0$, $x_2 = 1$, $y = 0$,
$TA_3 = E$.
Thus, Type II, $\neg x_2 = 0$,
$C = \neg x_1 = 1$.

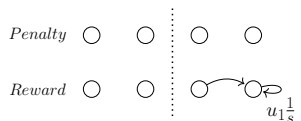

Condition: $x_1 = 1$, $x_2 = 1$, $y = 1$,
$TA_3 = I$.
Thus, Type I, $\neg x_2 = 0$,
$C = \neg x_1 \wedge x_2 = 0$.

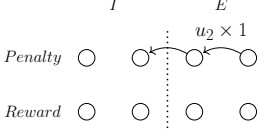

Condition: $x_1 = 0$, $x_2 = 1$, $y = 0$,
$TA_3 = I$.
Thus, Type II, $\neg x_2 = 0$,
$C = \neg x_1 \wedge x_2 = 1$.

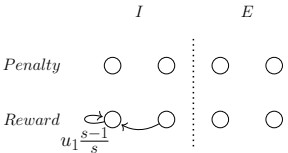

### B.1.2 CASE 2: INCLUDE $x_1$

For Case 2, we assume that the actions of the TAs for the first bit are frozen as $TA_1 = I$ and $TA_2 = E$, and thus the overall joint action for the first bit is "$x_1$". Similar to Case 1, we also have four situations.

#### B.1.2.1 Study $TA_3$ with Action Include

Condition: $x_1 = 1$, $x_2 = 1$, $y = 1$,
$TA_4 = E$.
Thus, Type I, $x_2 = 1$,
$C = x_1 \wedge x_2 = 1$.

Condition: $x_1 = 1$, $x_2 = 1$, $y = 1$,
$TA_4 = I$.
Thus, Type I, $x_2 = 1$,
$C = x_1 \wedge x_2 \wedge \neg x_2 = 0$.

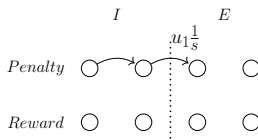

### B.1.2.2 Study $TA_3$ with Action Exclude

Condition: $x_1 = 1$, $x_2 = 1$, $y = 1$,
$TA_4 = E$.
Thus, Type I, $x_2 = 1$,
$C = x_1 = 1$.

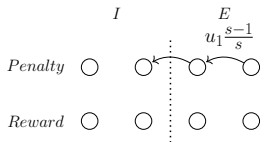

Condition: $x_1 = 1$, $x_2 = 0$, $y = 0$,
$TA_4 = E$.
Thus, Type II, $x_2 = 0$,
$C = x_1 = 1$.

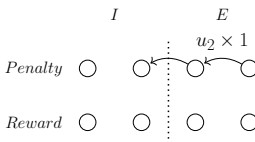

Condition: $x_1 = 1$, $x_2 = 1$, $y = 1$,
$TA_4 = I$.
Thus, Type I, $x_2 = 1$,
$C = x_1 \wedge \neg x_2 = 0$.

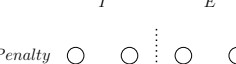

Condition: $x_1 = 1$, $x_2 = 0$, $y = 0$,
$TA_4 = I$.
Thus, Type II, $x_2 = 0$,
$C = x_1 \wedge \neg x_2 = 1$.

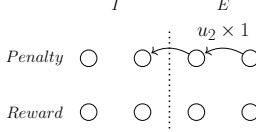

### B.1.2.3 Study $TA_4$ with Action Include

Condition: $x_1 = 1$, $x_2 = 1$, $y = 1$,
$TA_3 = E$.
Thus, Type I, $\neg x_2 = 0$,
$C = x_1 \wedge \neg x_2 = 0$.

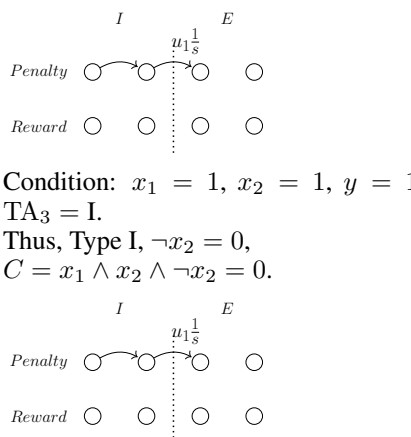

Condition: $x_1 = 1$, $x_2 = 1$, $y = 1$,
$TA_3 = I$.
Thus, Type I, $\neg x_2 = 0$,
$C = x_1 \wedge x_2 \wedge \neg x_2 = 0$.

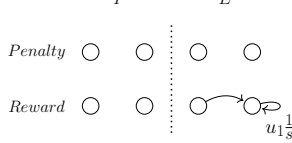

### B.1.2.4 Study $TA_4$ with Action Exclude

Condition: $x_1 = 1$, $x_2 = 1$, $y = 1$,
$TA_3 = E$.
Thus, Type I, $\neg x_2 = 0$,
$C = x_1 = 1$.

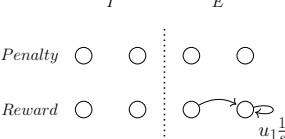

Condition: $x_1 = 1$, $x_2 = 1$, $y = 1$,
$TA_3 = I$.
Thus, Type I, $\neg x_2 = 0$,
$C = x_1 \wedge x_2 = 1$.

Condition: $x_1 = 1$, $x_2 = 1$, $y = 1$,
$TA_3 = I$.
Thus, Type I, $\neg x_2 = 0$,
$C = x_1 \wedge x_2 = 1$.

### B.1.3 CASE 3: EXCLUDE BOTH $\neg x_1$ AND $x_1$

For Case 3, we assume that the actions of TAs for the first bit are frozen as $TA_1 = E$ and $TA_2 = E$, with four situations. Note that in the training process, when all literals are excluded, $C$ is assigned to 1.

### B.1.3.1 Study $TA_3$ with Action Include

Condition: $x_1 = 1$, $x_2 = 1$, $y = 1$,
$TA_4 = E$.
Thus, Type I, $x_2 = 1$,
$C = x_2 = 1$.

Condition: $x_1 = 1$, $x_2 = 1$, $y = 1$,
$TA_4 = I$.
Thus, Type I, $x_2 = 1$,
$C = 0$.

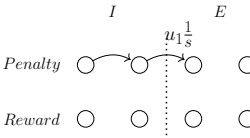

### B.1.3.2 Study TA₃ with Action Exclude

Condition: $x_1 = 1$, $x_2 = 1$, $y = 1$,
$TA_4 = E$.
Thus, Type I, $x_2 = 1$,
$C = 1$.

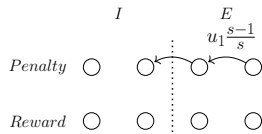

Condition: $x_1 = 1$, $x_2 = 0$, $y = 0$,
$TA_4 = E$.
Thus, Type II, $x_2 = 0$,
$C = 1$.

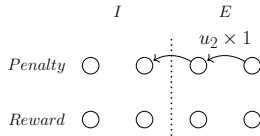

Condition: $x_1 = 0$, $x_2 = 0$, $y = 0$,
$TA_4 = E$.
Thus, Type II, $x_2 = 0$,
$C = 1$.

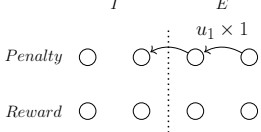

Condition: $x_1 = 1$, $x_2 = 1$, $y = 1$,
$TA_4 = I$.
Thus, Type I, $x_2 = 1$,
$C = 0$.

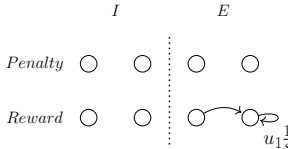

Condition: $x_1 = 1$, $x_2 = 0$, $y = 0$,
$TA_4 = I$.
Thus, Type II, $x_2 = 0$,
$C = 1$.

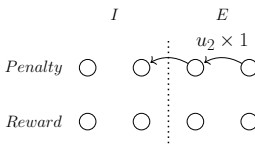

Condition: $x_1 = 0$, $x_2 = 0$, $y = 0$,
TA$_4$ = I.
Thus, Type II, $x_2 = 0$,
$C = 1$.

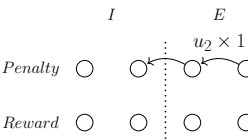

### B.1.3.3 Study TA$_4$ with Action Include

Condition: $x_1 = 1$, $x_2 = 1$, $y = 1$,
TA$_3$ = E.
Thus, Type I, $\neg x_2 = 0$,
$C = \neg x_2 = 0$.

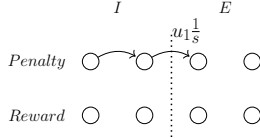

Condition: $x_1 = 1$, $x_2 = 1$, $y = 1$,
TA$_3$ = I.
Thus, Type I, $\neg x_2 = 0$,
$C = \neg x_2 \wedge x_2 = 0$.

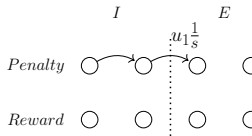

### B.1.3.4 Study TA$_4$ with Action Exclude

Condition: $x_1 = 1$, $x_2 = 1$, $y = 1$,
TA$_3$ = E.
Thus, Type I, $\neg x_2 = 0$,
$C = 1$.

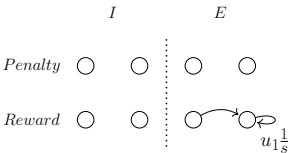

Condition: $x_1 = 0$, $x_2 = 1$, $y = 0$,
TA$_3$ = E.
Thus, Type II, $\neg x_2 = 0$,
$C = 1$.

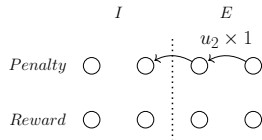

Condition: $x_1 = 1$, $x_2 = 1$, $y = 1$,
TA$_3$ = I.
Thus, Type I, $\neg x_2 = 0$,
$C = 1$.

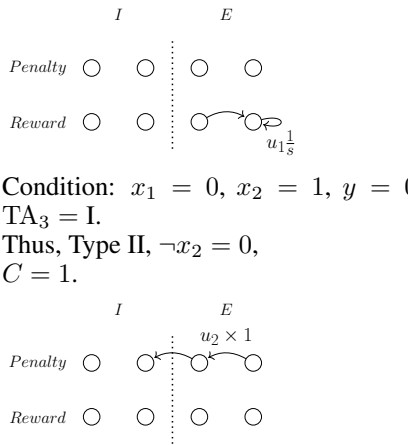

Condition: $x_1 = 0$, $x_2 = 1$, $y = 0$,
$\text{TA}_3 = \text{I}$.
Thus, Type II, $\neg x_2 = 0$,
$C = 1$.

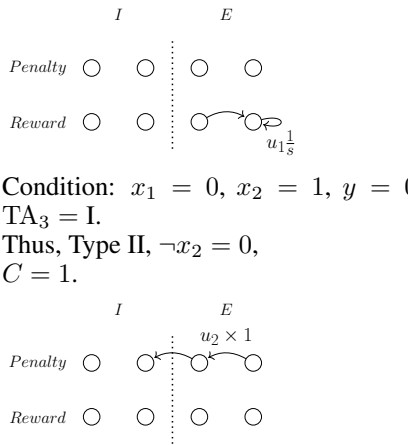

### B.1.4 CASE 4: INCLUDE BOTH $\neg x_1$ AND $x_1$

For Case 4, we assume that the actions of TAs for the first bit are frozen as $\text{TA}_1 = \text{I}$ and $\text{TA}_2 = \text{I}$, and thus $C = \textbf{0 always}$. Similarly, we also have four situations, detailed below.

#### B.1.4.1 *Study* $\text{TA}_3$ *with Action Include*

Condition: $x_1 = 1$, $x_2 = 1$, $y = 1$,
$\text{TA}_4 = \text{E}$.
Thus, Type I, $x_2 = 1$,
$C = 0$.

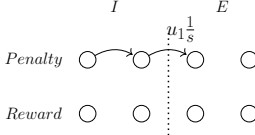

Condition: $x_1 = 1$, $x_2 = 1$, $y = 1$,
$\text{TA}_4 = \text{I}$.
Thus, Type I, $x_2 = 1$,
$C = 0$.

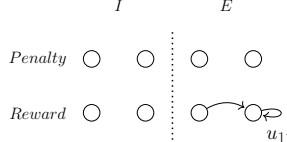

#### B.1.4.2 *Study* $\text{TA}_3$ *with Action Exclude*

Condition: $x_1 = 1$, $x_2 = 1$, $y = 1$,
$\text{TA}_4 = \text{E}$.
Thus, Type I, $x_2 = 1$,
$C = 0$.

Condition: $x_1 = 1$, $x_2 = 1$, $y = 1$,
$\text{TA}_4 = \text{I}$.
Thus, Type I, $x_2 = 1$,
$C = 0$.

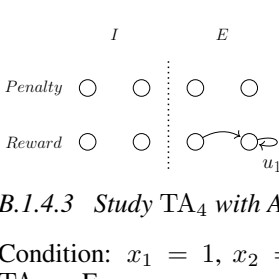

### B.1.4.3 Study $\text{TA}_4$ with Action Include

Condition: $x_1 = 1$, $x_2 = 1$, $y = 1$,
$\text{TA}_3 = \text{E}$.
Thus, Type I, $\neg x_2 = 0$,
$C = 0$.

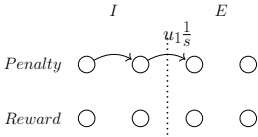

Condition: $x_1 = 1$, $x_2 = 1$, $y = 1$,
$\text{TA}_3 = \text{I}$.
Thus, Type I, $\neg x_2 = 0$,
$C = 0$.

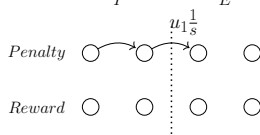

### B.1.4.4 Study $\text{TA}_4$ with Action Exclude

Condition: $x_1 = 1$, $x_2 = 1$, $y = 1$,
$\text{TA}_3 = \text{E}$.
Thus, Type I, $\neg x_2 = 0$,
$C = 0$.

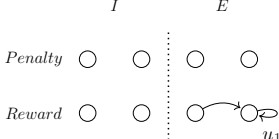

Condition: $x_1 = 1$, $x_2 = 1$, $y = 1$,
$\text{TA}_3 = \text{I}$.
Thus, Type I, $\neg x_2 = 0$,
$C = 0$.

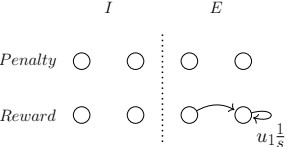

So far, we have gone through, exhaustively, the transitions of $\text{TA}_3$ and $\text{TA}_4$ for all the cases (all possible training samples and system states). Hereafter, we can summarize the direction of transitions and study the convergence properties of the system for the given training samples, to be detailed in the next subsection.

## B.2 Summarize of the Directions of Transitions in Different Cases

Based on the analysis above, we summarize here what happens to $\text{TA}_3$ and $\text{TA}_4$, given different status (Cases) of $\text{TA}_1$ and $\text{TA}_2$. More specifically, we will summarize here the directions of the

transitions for the TAs. For example, "$TA_3 \Rightarrow$ E" means that $TA_3$ will move towards the action "Exclude", while "$TA_4 \Rightarrow$ E or I" means $TA_4$ transits towards either "Exclude" or "Include".

**Scenario 1:** Study $TA_3 =$ I and $TA_4 =$ I.

| **Case 1**, we have: | **Case 3**, we have: |
|---|---|
| $TA_3 \Rightarrow$ E. | $TA_3 \Rightarrow$ E. |
| $TA_4 \Rightarrow$ E. | $TA_4 \Rightarrow$ E. |
| **Case 2**, we have: | **Case 4**, we have: |
| $TA_3 \Rightarrow$ E. | $TA_3 \Rightarrow$ E. |
| $TA_4 \Rightarrow$ E. | $TA_4 \Rightarrow$ E. |

From the facts presented above, we can confirm that regardless the state of $TA_1$ and $TA_2$, if $TA_3 =$ I and $TA_4 =$ I, they ($TA_3$ and $TA_4$) will eventually move out of their states.

**Scenario 2:** Study $TA_3 =$ I and $TA_4 =$ E.

| **Case 1**, we have: | **Case 3**, we have: |
|---|---|
| $TA_3 \Rightarrow$ E. | $TA_3 \Rightarrow$ I. |
| $TA_4 \Rightarrow$ E or I. | $TA_4 \Rightarrow$ E or I. |
| **Case 2**, we have: | **Case 4**, we have: |
| $TA_3 \Rightarrow$ I. | $TA_3 \Rightarrow$ E. |
| $TA_4 \Rightarrow$ E. | $TA_4 \Rightarrow$ E. |

For Scenario 2 Case 2, we can observe that if $TA_3 =$ I, $TA_4 =$ E, $TA_1 =$ I, and $TA_2 =$ E, $TA_3$ will move deeper to "include" and $TA_4$ will go deeper to "exclude". It is not difficult to derive also that $TA_1$ will move deeper to "include" and $TA_2$ will transfer deeper to "exclude" in this circumstance. This tells us that the TAs in states $TA_3 =$ I, $TA_4 =$ E, $TA_1 =$ I, and $TA_2 =$ E, reinforce each other to move deeper to their corresponding directions and they therefore construct an absorbing state of the system. If it is the only absorbing state, we can conclude that the TM converge to the intended "AND" operation.

In Scenario 2, we can observe for Cases 1, 3, and 4, the actions for $TA_3$ and $TA_4$ are not absorbing because the TAs will not be reinforced to move monotonically deeper to the states of the corresponding actions for difference cases.

For Scenario 2, Case 3, $TA_4$ has two possible directions to transit, I or E, depending on the input of the training sample. For action exclude, it will be reinforced when training sample $x_1 = 1$ and $x_2 = 1$ is given, based on Type I feedback. However, $TA_4$ will transit towards "include" side when training sample $x_1 = 0$ and $x_2 = 1$ is given, due to Type II feedback. Therefore, the direction of the transition for $TA_4$ is I or E, depending on the training samples. In the following paragraphs, when "or" appears in the transition direction, the same concept applies.

**Scenario 3:** Study $TA_3 =$ E and $TA_4 =$ I.

| **Case 1**, we have: | **Case 3**, we have: |
|---|---|
| $TA_3 \Rightarrow$ E or I. | $TA_3 \Rightarrow$ E or I. |
| $TA_4 \Rightarrow$ E. | $TA_4 \Rightarrow$ E. |
| **Case 2**, we have: | **Case 4**, we have: |
| $TA_3 \Rightarrow$ E or I. | $TA_3 \Rightarrow$ E. |
| $TA_4 \Rightarrow$ E. | $TA_4 \Rightarrow$ E. |

In Scenario 3, we can see that the actions for $TA_3 =$ E and $TA_4 =$ I are not absorbing because the TAs will not be reinforced to move deeper to the states of the corresponding actions.

**Scenario 4:** Study $TA_3 =$ E and $TA_4 =$ E.

**Case 1**, we have:      **Case 3**, we have:

$TA_3 \Rightarrow$ I or E.      $TA_3 \Rightarrow$ I.

$TA_4 \Rightarrow$ I or E.      $TA_4 \Rightarrow$ I or E.

**Case 2**, we have:      **Case 4**, we have:

$TA_3 \Rightarrow$ I.      $TA_3 \Rightarrow$ E.

$TA_4 \Rightarrow$ E.      $TA_4 \Rightarrow$ E.

In Scenario 4, we see that, the actions for $TA_3 = E$ and $TA_4 = E$ seem to be an absorbing state, because the states of TAs will move deeper in Case 4. After a revisit of the condition for Case 4, i.e., include both $\neg x_1$ and $x_1$, we understand that this condition is not absorbing. In fact, when $TA_1$ and $TA_2$ both have "Include" as their actions, they monotonically move towards "Exclude". Therefore, from the overall system's perspective, the system state $TA_1 = I$, $TA_2 = I$, $TA_3 = E$, and $TA_4 = E$ is not absorbing. For the other cases in this scenario, there is no absorbing state.

Based on the above analysis, we understand that there is only one absorbing condition in the system, namely, $TA_1 = I$, $TA_2 = E$, $TA_3 = I$, and $TA_4 = E$, for the given training samples with AND logic. The same conclusion applies when we freeze the transition of the two TAs for the second bit of the input and study behavior of the first bit of input. Therefore, we can conclude that the TM with a clause can learn to be the intended AND operator, almost surely, in infinite time horizon. We thus complete the proof of Theorem 1. ∎

## C  PROOF OF LEMMA 1

The probability of the training samples for the noise-free OR operator can be presented by the following equations.

$$P(y = 1|x_1 = 1, x_2 = 1) = 1, \tag{19}$$
$$P(y = 1|x_1 = 0, x_2 = 1) = 1,$$
$$P(y = 1|x_1 = 1, x_2 = 0) = 1,$$
$$P(y = 0|x_1 = 0, x_2 = 0) = 1.$$

Clearly, there are three sub-patterns of $x_1$ and $x_2$ that will give $y = 1$, i.e., $[x_1 = 1, x_2 = 1]$, $[x_1 = 1, x_2 = 0]$, and $[x_1 = 0, x_2 = 1]$. More specifically, Eq. (19) can be split into three cases, corresponding to the three sub-patterns:

$$P(y = 1|x_1 = 1, x_2 = 1) = 1, \tag{20}$$
$$P(y = 0|x_1 = 0, x_2 = 0) = 1,$$

$$P(y = 1|x_1 = 0, x_2 = 1) = 1, \tag{21}$$
$$P(y = 0|x_1 = 0, x_2 = 0) = 1,$$

and

$$P(y = 1|x_1 = 1, x_2 = 0) = 1, \tag{22}$$
$$P(y = 0|x_1 = 0, x_2 = 0) = 1.$$

In what follows, we will show the convergence of each of the three sub-patterns, i.e., Lemma 1.

The convergence analyses of the above three sub-patterns can be derived by reusing the analyses of the sub-patterns of the XOR operator plus the AND operator. For the sub-pattern described by Eq. (20), we can confirm that the TAs will indeed converge to $TA_1 = I$, $TA_2 = E$, $TA_3 = I$, and $TA_4 = E$, by studying the transition diagrams in Subsection B when input samples of $[x_1 = 0, x_2 = 1]$ and $[x_1 = 1, x_2 = 0]$ are removed. In this case, the directions of the transitions for different scenarios are summarized below.

**Scenario 1:** Study $TA_3 = I$ and $TA_4 = I$.

**Case 1**, we have:
$TA_3 \Rightarrow E$.
$TA_4 \Rightarrow E$.
**Case 2**, we have:
$TA_3 \Rightarrow E$.
$TA_4 \Rightarrow E$.

**Case 3**, we have:
$TA_3 \Rightarrow E$.
$TA_4 \Rightarrow E$.
**Case 4**, we have:
$TA_3 \Rightarrow E$.
$TA_4 \Rightarrow E$.

**Scenario 2:** Study $TA_3 = I$ and $TA_4 = E$.

**Case 1**, we have:
$TA_3 \Rightarrow E$.
$TA_4 \Rightarrow E$.
**Case 2**, we have:
$TA_3 \Rightarrow I$.
$TA_4 \Rightarrow E$.

**Case 3**, we have:
$TA_3 \Rightarrow$ I.
$TA_4 \Rightarrow$ E.
**Case 4**, we have:
$TA_3 \Rightarrow$ E.
$TA_4 \Rightarrow$ E.

**Scenario 3:** Study $TA_3 =$ E and $TA_4 =$ I.

**Case 1**, we have:
$TA_3 \Rightarrow$ E or I.
$TA_4 \Rightarrow$ E.
**Case 2**, we have:
$TA_3 \Rightarrow$ E.
$TA_4 \Rightarrow$ E.

**Case 3**, we have:
$TA_3 \Rightarrow$ E or I.
$TA_4 \Rightarrow$ E.
**Case 4**, we have:
$TA_3 \Rightarrow$ E.
$TA_4 \Rightarrow$ E.

**Scenario 4:** Study $TA_3 =$ E and $TA_4 =$ E.

**Case 1**, we have:
$TA_3 \Rightarrow$ I or E.
$TA_4 \Rightarrow$ E.
**Case 2**, we have:
$TA_3 \Rightarrow$ I.
$TA_4 \Rightarrow$ E.

**Case 3**, we have:
$TA_3 \Rightarrow$ I.
$TA_4 \Rightarrow$ E.
**Case 4**, we have:
$TA_3 \Rightarrow$ E.
$TA_4 \Rightarrow$ E.

Comparing the analysis with the one in Subsection B.2, there is apparently another possible absorbing case, which can be observed in Scenario 2, Case 3, where $TA_3 =$ I and $TA_4 =$ E, given $TA_1 =$ E and $TA_2 =$ E. However, given $TA_3 =$ I and $TA_4 =$ E, the TAs for the first bit, i.e., $TA_1 =$ E and $TA_2 =$ E, will not move only towards Exclude. Therefore, they do not reinforce each other to move to deeper states for their current actions. For this reason, the system in $TA_3 =$ I, $TA_4 =$ E, $TA_1 =$ E, and $TA_2 =$ E, is not in an absorbing state. In addition, given $TA_3 =$ I and $TA_4 =$ E, $TA_1$ and $TA_2$ with actions E and E will transit towards I and E, encouraging the overall system to move towards I, E, I, and E. Consequently, the system state with $TA_1 =$ I, $TA_2 =$ E, $TA_3 =$ I, and $TA_4 =$ E is still the only absorbing case for the given training samples following Eq. (20).

For Eq. (21), similar to the proof of in Lemma 1 in (Jiao et al., 2023), we can derive that the TAs will converge in $TA_1 =$ E, $TA_2 =$ I, $TA_3 =$ I, and $TA_4 =$ E. The transition diagrams for the samples of Eq. (21) are in fact a subset of the ones presented in Subsection 3.2.1 and Appendix 2 of (Jiao et al., 2023), when the input samples of $[x_1 = 1$ and $x_2 = 1]$ are removed. We summarize below only the directions of transitions.

The directions of the transitions of the TAs for the second input bit, i.e., $x_2/\neg x_2$, when the TAs for the first input bit are frozen, are summarized as follows (based on the subset of the transition diagrams in Subsection 3.2.1 of (Jiao et al., 2023)).

**Scenario 1:** Study $TA_3 =$ I and $TA_4 =$ I.

**Case 1:** we have
$TA_3 \rightarrow E$
$TA_4 \rightarrow E$
**Case 2:** we have
$TA_3 \rightarrow E$
$TA_4 \rightarrow E$

**Case 3:** we have
$TA_3 \rightarrow E$
$TA_4 \rightarrow E$
**Case 4:** we have
$TA_3 \rightarrow E$
$TA_4 \rightarrow E$

**Scenario 2:** Study $TA_3 = I$ and $TA_4 = E$.

**Case 1:** we have
$TA_3 \rightarrow I$
$TA_4 \rightarrow E$
**Case 2:** we have
$TA_3 \rightarrow E$
$TA_4 \rightarrow E$

**Case 3:** we have
$TA_3 \rightarrow I$
$TA_4 \rightarrow E$
**Case 4:** we have
$TA_3 \rightarrow E$
$TA_4 \rightarrow E$

**Scenario 3:** Study $TA_3 = E$ and $TA_4 = I$.

**Case 1:** we have
$TA_3 \rightarrow I$, or E
$TA_4 \rightarrow E$
**Case 2:** we have
$TA_3 \rightarrow E$
$TA_4 \rightarrow E$

**Case 3:** we have
$TA_3 \rightarrow I$, or E
$TA_4 \rightarrow E$
**Case 4:** we have
$TA_3 \rightarrow E$
$TA_4 \rightarrow E$

**Scenario 4:** Study $TA_3 = E$ and $TA_4 = E$.

**Case 1:** we have
$TA_3 \rightarrow I$
$TA_4 \rightarrow E$
**Case 2:** we have
$TA_3 \rightarrow E$
$TA_4 \rightarrow E$

**Case 3:** we have
$TA_3 \rightarrow I$
$TA_4 \rightarrow E$
**Case 4:** we have
$TA_3 \rightarrow E$
$TA_4 \rightarrow E$

The directions of the transitions of the TAs for the first input bit, i.e., $x_1/\neg x_1$, when the TAs for the second input bit are frozen, are summarized as follows (based on the subset of the transition diagrams in Appendix 2 of (Jiao et al., 2023)).

**Scenario 1:** Study $TA_1 = I$ and $TA_2 = I$.

**Case 1:** we have
$TA_1 \to E$
$TA_2 \to E$
**Case 2:** we have
$TA_1 \to E$
$TA_2 \to E$

**Case 3:** we have
$TA_1 \to E$
$TA_2 \to E$
**Case 4:** we have
$TA_1 \to E$
$TA_2 \to E$

**Scenario 2:** Study $TA_1 = I$ and $TA_2 = E$.

**Case 1:** we have
$TA_1 \to E$
$TA_2 \to E$
**Case 2:** we have
$TA_1 \to E$
$TA_2 \to E$

**Case 3:** we have
$TA_1 \to E$
$TA_2 \to E$
**Case 4:** we have
$TA_1 \to E$
$TA_2 \to E$

**Scenario 3:** Study $TA_1 = E$ and $TA_2 = I$.

**Case 1:** we have
$TA_1 \to I$, or E
$TA_2 \to E$
**Case 2:** we have
$TA_1 \to E$
$TA_2 \to I$

**Case 3:** we have
$TA_1 \to I$
$TA_2 \to I$
**Case 4:** we have
$TA_1 \to E$
$TA_2 \to E$

**Scenario 4:** Study $TA_1 = E$ and $TA_2 = E$.

**Case 1:** we have
$TA_1 \to I$, or E
$TA_2 \to E$
**Case 2:** we have
$TA_1 \to E$
$TA_2 \to I$

**Case 3:** we have
$TA_1 \rightarrow E$
$TA_2 \rightarrow E$
**Case 4:** we have
$TA_1 \rightarrow E$
$TA_2 \rightarrow E$

By analyzing the transitions of TAs for the two input bits with samples following Eq. (21), we can conclude that $TA_1 = E$, $TA_2 = I$, $TA_3 = I$, and $TA_4 = E$ is an absorbing state, as the actions of $TA_1$–$TA_4$ reinforce each other to transit to deeper states for the current actions upon various input samples. There are a few other cases in different scenarios that seem to be absorbing, but in fact not. For example, the status $TA_3 = I$ and $TA_4 = E$ seems also absorbing in Scenario 2, Case 3, i.e., when $TA_1 = E$ and $TA_2 = E$ hold. However, to make $TA_1 = E$ and $TA_2 = E$ absorbing, the condition is $TA_3 = I$ and $TA_4 = I$, or $TA_3 = E$ and $TA_4 = E$. Clearly, the status $TA_3 = I$ and $TA_4 = I$ is not absorbing. For $TA_3 = E$ and $TA_4 = E$ to be absorbing, it is required to have $TA_1 = I$ and $TA_2 = I$ to be absorbing, or $TA_1 = I$ and $TA_2 = E$ to be absorbing, which are not true. Therefore, all those absorbing-like states are not absorbing. In fact, when $TA_3 = I$, $TA_4 = E$, $TA_1 = E$, and $TA_2 = E$ hold, the condition $TA_3 = I$, $TA_4 = E$ will reinforce $TA_1$ and $TA_2$ to move towards E, I, which is the absorbing state of the system. Based on the above analysis on the transition directions, we can thus confirm the convergence of TM when training samples from Eq. (21) are given.

Following the same principle, we can also confirm that the TAs will converge to $TA_1 = I$, $TA_2 = E$, $TA_3 = E$, and $TA_4 = I$ when training samples from Eq. (22) are given, according to the proof of Lemma 2 in (Jiao et al., 2023).

## D Proof of Lemma 2

**Proof of Lemma 2:** To show the non-absorbing property when samples following Eq. (9) are given, we need to show that the absorbing states for Eq. (6) disappear when ($[x_1 = 1, x_2 = 0], y = 1$) is given in addition, and the same applies to Eq. (8) when ($[x_1 = 1, x_2 = 1], y = 1$) is given.

We first show that the absorbing state of $TA_1 = I$, $TA_2 = E$, $TA_3 = I$, $TA_4 = E$, for sub-pattern ($[x_1 = 1, x_2 = 1], y = 1$) as shown in Eq. (6), disappears when sub-pattern ($[x_1 = 1, x_2 = 0], y = 1$) is given in addition. Indeed, $TA_3$ will move toward E when ($[x_1 = 1, x_2 = 0], y = 1$) is given, because a penalty is given to $TA_3$ as shown in Fig. 6.

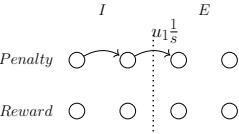

Figure 6: Transition of $TA_3$ when its current action is Include, $TA_1$, $TA_2$, and $TA_4$'s actions are Include, Exclude, and Exclude, respectively, upon a training sample ($[x_1 = 1, x_2 = 0], y = 1$).

Clearly, when ($[x_1 = 1, x_2 = 0], y = 1$) is given in addition, $TA_3$ has a non-zero probability to move towards "Exclude". Therefore, "Include" is not the only direction that $TA_3$ moves to upon the new input. In other words, ($[x_1 = 1, x_2 = 0], y = 1$) will make the state $TA_1 = I$, $TA_2 = E$, $TA_3 = I$, $TA_4 = E$, not absorbing any longer. For other states, the newly added training sample will not remove any transition from the previous case. For this reason, the system will not have any new absorbing state. Therefore, when ($[x_1 = 1, x_2 = 0], y = 1$) is given in addition, the absorbing state disappears and the system will not have any new absorbing state.

Following the same concept, we show that the absorbing state for ($[x_1 = 1, x_2 = 0], y = 1$) shown in Eq. (8), i.e., $TA_1 = I$, $TA_2 = E$, $TA_3 = E$, $TA_4 = I$, disappears when sub-pattern ($[x_1 = 1, x_2 = 1], y = 1$) is given in addition. Indeed, $TA_4$ will also move towards E when ($[x_1 = 1, x_2 = 1], y = 1$) is given, as shown in Fig. 7.

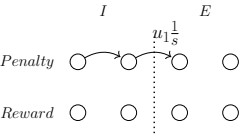

Figure 7: Transition of $TA_4$ when its current action is Include, $TA_1$, $TA_2$, and $TA_3$'s actions are Include, Exclude, and Exclude, respectively, upon a training sample ($x_1 = 1, x_2 = 1, y = 1$).

Understandably, because of the newly added sub-patterns, the absorbing states in Eqs. (6) and (8) disappear and no new absorbing states are generated. In other words, the TM trained based on samples from Eq. (9) becomes non-absorbing.

Following the same concept, we can show that the system becomes non-absorbing for Eqs. (5), (10), and (11) as well. For the sake of conciseness, we will not provide the details here. In general, any newly added sub-pattern will involve a probability for the learnt sub-pattern to move outside the learnt state, making the system non-absorbing. ∎

## E    PROOF OF LEMMA 3

**Proof of Lemma 3:** In Lemma 2, the TM is non-absorbing if the functionality of $T$ is disabled (i.e., when $u_1 > 0, u_2 > 0$ always hold). Therefore, for the OR operator to converge, the functionality of $T$ is critical to block any feedback in order to form an absorbing state.

By design, TM will either be updated via Type I feedback or Type II feedback. We show via (1) the condition when Type I feedback is blocked and then show via (2) when any update from Type II feedback is not triggered. When both happen, the system will not be updated anymore and thus absorbed.

To prove (1) in Lemma 3, we show that the system is not absorbed when 0 or 1 intended sub-pattern is blocked by $T$. When 2 intended sub-patterns are blocked, the system will guide the clauses to learn the remaining intended sub-pattern. Only when all 3 intended sub-patterns are blocked by $T$, the system will stop updating based on Type I feedback.

Clearly, when no intended sub-pattern is blocked by $T$, the training samples provided to the system follow Eq. (5). In other words, no samples corresponding to a specific sub-pattern are blocked. Under such training conditions, as shown in Lemma 2, the TM is non-absorbing. When only 1 intended sub-pattern is blocked by $T$, the system is updated based on samples following Eqs. (9), (10), or (11), which is also non-absorbing.

We look at the cases when two intended sub-patterns are blocked by $T$ but the third one is not blocked. In other words, the number of clauses for each of the two intended sub-patterns reaches at least $T$, and the number of clauses for the remaining sub-pattern is less than $T$. In this case, only one type of samples from Eqs. (6) or (7) or (8) will be provided to the TM[4]. Based on Lemma 1, we understand that all clauses, including the ones that have learnt the two blocked sub-patterns, will be forced to learn the not-yet-blocked sub-pattern. This is due to the fact that only the samples following the not-yet-blocked sub-pattern are triggering the update for the TM. In this circumstance, as soon as the not-yet-blocked sub-pattern also has $T$ clauses, i.e., when all three sub-patterns are blocked by $T$ at the same time, Type I feedback are blocked completely.

Note that the samples corresponding to the not-yet-blocked sub-pattern will encourage the learnt clauses (i.e., the clauses for the blocked sub-patterns) to move out from the learnt sub-patterns, and this may cause the number of clauses for the blocked sub-pattern being lower than $T$ (thus unblocked), again. If this happens before the number of clauses for the not-yet-blocked sub-pattern reaches $T$, at least two sub-patterns will be in the non-blocked state, and the system becomes one of the three cases described by Eqs. (9), (10) or (11). In other words, even if an absorbing state exists after two intended sub-patterns are blocked by $T$, the system may not monotonically move towards the absorbing state. Nevertheless, as soon as all three intended sub-patterns are blocked by reaching $T$ clauses, the Type I feedback will be blocked.

Here we prove (2) in Lemma 3. Type II feedback is only triggered by training sample ($[x_1 = 0, x_2 = 0]$, $y = 0$) in the OR operator. For Type II feedback, based on Table 2, a transition is triggered only when a penalty occurs, i.e., when the excluded literal has a value of 0 and the clause evaluates to 1. Specifically for the OR operation, this only happens when $C = \neg x_1 \wedge \neg x_2$ or $C = \neg x_1$ or $C = \neg x_2$. For $C = \neg x_1 \wedge \neg x_2$, based on the Type II feedback, the TA with the action "excluding $x_1$" and the TA with the action "excluding $x_2$" will be penalized. In other words, the actions of the two TAs for $x_1$ and $x_2$ will be encouraged to move from exclude to include side. As soon as one of the TAs (or occasionally both of them) becomes include, the clause will become $C = x_1 \wedge \neg x_1 \wedge \neg x_2$ or $C = \neg x_1 \wedge x_2 \wedge \neg x_2$ (or occasionally $C = x_1 \wedge \neg x_1 \wedge x_2 \wedge \neg x_2$). In this case, input $[x_1 = 0, x_2 = 0]$ will always result in 0 as the clause value and then the Type II feedback will not update the system any longer. Following the same concept, for $C = \neg x_2$, the Type II feedback will encourage the excluded $x_1$ to be included so that the clause becomes $C = x_1 \wedge \neg x_2$. The same applies to $C = \neg x_1$, which will eventually become $C = \neg x_1 \wedge x_2$ upon Type II feedback. When all clauses in $C = \neg x_2$ or $C = \neg x_1$ are also updated to $C = x_1 \wedge \neg x_2$ or $C = \neg x_1 \wedge x_2$, no Type II feedback is triggered up on any input sample.

We summarize the requirements for an absorbing state:

---

[4]More precisely speaking, all samples will be fed into the TM, but only samples corresponding to the not-yet-blocked sub-pattern will be used by the TM for training purpose.

- For any sample $\mathbf{X}$ following sub-pattern $[x_1 = 1, x_2 = 1]$, or $[x_1 = 1, x_2 = 0]$, or $[x_1 = 0, x_2 = 1]$, the number of clauses for that sub-pattern, i.e., $f_\Sigma(\mathcal{C}^i(\mathbf{X}))$, must be at least $T$, no matter in which form the clauses are constructed. This will block Type I feedback.

- There are no clauses with literal(s) in only negated form, such as $C = \neg x_1$ or $C = \neg x_2$ or or $C = \neg x_1 \wedge \neg x_2$. This guarantees that no transition will happen upon any Type II feedback. ∎

# F    ANALYSIS OF THE TM WITH WRONG TRAINING LABELS

In this appendix, we analyze the transition properties of the TM when training samples contain wrong labels.

There are two types of wrong labels:

- Inputs labeled as 0, which should be 1.
- Inputs labeled as 1, which should be 0.

We begin by examining the first type of wrong label, followed by the second type, and then address the general case.

## F.1    THE AND OPERATOR WITH THE FIRST TYPE OF WRONG LABELS

To formally define training samples with the first type of wrong label, we use the following formulas:

$$P\left(y = 1 | x_1 = 1, x_2 = 1\right) = a, a \in (0, 1) \tag{23}$$
$$P\left(y = 0 | x_1 = 1, x_2 = 1\right) = 1 - a,$$
$$P\left(y = 0 | x_1 = 0, x_2 = 1\right) = 1,$$
$$P\left(y = 0 | x_1 = 1, x_2 = 0\right) = 1,$$
$$P\left(y = 0 | x_1 = 0, x_2 = 0\right) = 1.$$

In this case, the label for training samples representing the intended logic $[x_1 = 1, x_2 = 1]$ is $y = 1$ with probability $a$ and $y = 0$ with probability $1 - a$. In other words, in addition to the training samples detailed in Subsection B, a new training sample will appear to the system, namely $([x_1 = 1, x_2 = 1], y = 0)$.

**Lemma 6.** *The TM exhibits non-absorbing for the training samples defined in Eq. (23).*

**Proof:** To prove this lemma, we analyze the TM's transitions as follows. First, we examine the transitions assuming $u_1 > 0$ and $u_2 > 0$, similar to the analysis in Subsection B, as detailed in Subsection F.1.1. Next, we study the impact of $T$ to determine whether it leads to convergence (absorption), as discussed in Subsection F.1.2.

### F.1.1    TRANSITION OF TM WITH AND OPERATOR GIVEN $u_1 > 0$ AND $u_2 > 0$

Following the approach in Subsection B, we examine the transitions of $\text{TA}_3$ and $\text{TA}_4$ when the additional training sample $([x_1 = 1, x_2 = 1], y = 0)$ is introduced, considering Cases 1 to 4 as defined in Subsection B. Since $y = 0$ for this sample, only Type II feedback can be triggered to cause transitions. As $\text{TA}_3$ is responsible for the literal $x_2$, which is always 1 for this sample, Type II feedback does not trigger any transitions for $\text{TA}_3$. Therefore, we focus on studying the potential transitions of $\text{TA}_4$ in the four cases defined in Subsection B.1.

In Case 1, where $\text{TA}_1 = \text{E}$ and $\text{TA}_2 = \text{I}$, the clause value will always be 0 for the training sample because $\neg x_1$ is included in the clause, regardless of the action $\text{TA}_4$ takes. According to the Type II feedback transition table, no transition occurs when $C = 0$, so no transitions are triggered for $\text{TA}_4$. Similarly, in Case 4, where $\text{TA}_1 = \text{I}$ and $\text{TA}_2 = \text{I}$, the clause value will always be 0 due to the presence of $x_1 \wedge \neg x_1$ in the clause. As a result, there are no transitions for $\text{TA}_4$.

In Case 2, where $\text{TA}_1 = \text{I}$ and $\text{TA}_2 = \text{E}$, the literal $x_1$ will always appear in the clause. When $\text{TA}_4 = \text{I}$, the clause includes the literal $\neg x_2$, which results in a clause value of 0. Therefore, no transition is triggered. However, when $\text{TA}_4 = \text{E}$, the literal $x_1$ will always appear in the clause, and the value of $x_2$ is 1, making the clause value 1 regardless of $\text{TA}_3$'s action (whether it includes or excludes $x_2$). According to the Type II feedback table, with the literal value of $\neg x_2$ being 0 and the clause value being 1, the transition for $\text{TA}_4 = \text{E}$ is:

Condition: $x_1 = 1$, $x_2 = 1$, $y = 0$.
Thus, Type II, $\neg x_2 = 0$,
$C = 1$.

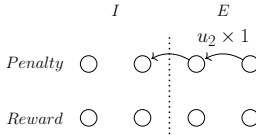

In Case 3, where $TA_1 = E$ and $TA_2 = E$, the clause value is fully determined by $TA_3$ and $TA_4$. When $TA_4$'s action is to include, the clause value is 0 for this sample because it includes the literal $\neg x_2$, resulting in no transition for $TA_4$. However, when $TA_4$'s action is to exclude, the clause value is always 1, regardless of $TA_3$'s action. Specifically, when $TA_3$ includes $x_2$, the clause value is 1, as the literal value of $x_2$ is 1. When it is exclude, all literals are excluded and then the clause value becomes 1 by definition. By examining the transitions of $TA_4$, we can summarize the following graph:

Condition: $x_1 = 1$, $x_2 = 1$, $y = 0$.
Thus, Type II, $\neg x_2 = 0$,
$C = 1$.

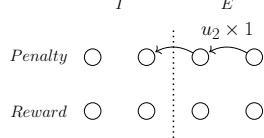

We summarize the directions of the transitions when the new wrongly labeled sample is added, with the newly added actions highlighted in red.

**Scenario 1:** Study $TA_3 = I$ and $TA_4 = I$.

**Case 1**, we have:  **Case 3**, we have:
$TA_3 \Rightarrow E$.  $TA_3 \Rightarrow E$.
$TA_4 \Rightarrow E$.  $TA_4 \Rightarrow E$.
**Case 2**, we have:  **Case 4**, we have:
$TA_3 \Rightarrow E$.  $TA_3 \Rightarrow E$.
$TA_4 \Rightarrow E$.  $TA_4 \Rightarrow E$.

**Scenario 2:** Study $TA_3 = I$ and $TA_4 = E$.

**Case 1**, we have:  **Case 3**, we have:
$TA_3 \Rightarrow E$.  $TA_3 \Rightarrow I$.
$TA_4 \Rightarrow E$ or I.  $TA_4 \Rightarrow E$ or I.
**Case 2**, we have:  **Case 4**, we have:
$TA_3 \Rightarrow I$.  $TA_3 \Rightarrow E$.
$TA_4 \Rightarrow E$ or I.  $TA_4 \Rightarrow E$.

**Scenario 3:** Study $TA_3 = E$ and $TA_4 = I$.

**Case 1**, we have:  **Case 3**, we have:
$TA_3 \Rightarrow E$ or I.  $TA_3 \Rightarrow E$ or I.
$TA_4 \Rightarrow E$.  $TA_4 \Rightarrow E$.
**Case 2**, we have:  **Case 4**, we have:
$TA_3 \Rightarrow E$ or I.  $TA_3 \Rightarrow E$.
$TA_4 \Rightarrow E$.  $TA_4 \Rightarrow E$.

**Scenario 4:** Study $TA_3 = E$ and $TA_4 = E$.

**Case 1**, we have:  **Case 3**, we have:
$TA_3 \Rightarrow I$ or E.  $TA_3 \Rightarrow I$.
$TA_4 \Rightarrow I$ or E.  $TA_4 \Rightarrow I$ or E.
**Case 2**, we have:  **Case 4**, we have:
$TA_3 \Rightarrow I$.  $TA_3 \Rightarrow E$.
$TA_4 \Rightarrow E$ or I.  $TA_4 \Rightarrow E$.

Clearly, the only absorbing state ($TA_3 = I$ and $TA_4 = E$) becomes non-absorbing due to the newly added transition (the red I for $TA_4$). As a result, the system is non-absorbing when $u_1 > 0$ and $u_2 > 0$.

### F.1.2 Transition of TM with AND Operator when $T$ can block Type I feedback

Based on the above analysis, we understand that the system is non-absorbing when $u_1 > 0$ and $u_2 > 0$. Next, we examine whether it is possible for the system to become absorbing when $T$ can block Type I feedback.

When $T$ clauses have learned the intended pattern $\mathbf{X} = [x_1 = 1, x_2 = 1]$, i.e., when $f_{\sum}(\mathcal{C}^i(\mathbf{X})) = T$, then $u_1 = 0$ holds, and Type I feedback is blocked. In this situation, only Type II feedback can occur. Due to the presence of the wrong label, i.e., ($[x_1 = 1, x_2 = 1], y = 0$), Type II feedback triggers transitions in the TAs that have already learned the intended logic ((($[x_1 = 1, x_2 = 1], y = 1$)). For example, Type II feedback will cause a transition in TAs of a learned clause $C = x_1 \wedge x_2$, making the clause deviate from its learned state (e.g., changing from $x_1 \wedge x_2$ to $x_1 \wedge x_2 \wedge \neg x_2$). Once this happens, $u_1 > 0$ holds, and Type I feedback is triggered by samples of ($[x_1 = 1, x_2 = 1], y = 1$), encouraging TAs in this clause to move back toward the action Exclude. Thus, even when $T$ blocks all Type I feedback samples (setting $u_1 = 0$), the system remains non-absorbing due to the wrong label and Type II feedback. Notably, no value of $f_{\sum}(\mathcal{C}^i(\mathbf{X}))$ can make both $u_1 = 0$ and $u_2 = 0$ simultaneously[5]. Therefore, Type I and Type II feedback cannot be blocked simultaneously, ensuring the system is non-absorbing. ∎

### F.2 The AND Operator with the Second Type of Wrong Labels

To properly define the training samples with the second type of wrong label, we employ the following formulas:

$$P(y = 1 | x_1 = 1, x_2 = 1) = 1, \tag{24}$$
$$P(y = 0 | x_1 = 1, x_2 = 0) = a, a \in (0, 1)$$
$$P(y = 1 | x_1 = 1, x_2 = 0) = 1 - a,$$
$$P(y = 0 | x_1 = 0, x_2 = 1) = 1,$$
$$P(y = 0 | x_1 = 0, x_2 = 0) = 1.$$

In this case, clearly, label of the training samples $[x_1 = 1, x_2 = 0]$ are wrongly labeled as 1 with probability $1 - a$. In other words, in addition to the training samples detailed in Subsection B, a new type (wrongly labeled) of training sample will appear to the system, namely ($[x_1 = 1, x_2 = 0], y = 1$).

**Lemma 7.** *The TM is non-absorbing for the training samples given by Eq. (24).*

**Proof:** Similar to the proof of Lemma 6, we first consider the transitions of TM with $u_1 > 0$ and $u_2 > 0$, and then examine the impact of $T$ for the system transition.

When $u_1 > 0$ and $u_2 > 0$, there is a non-zero probability in which the training sample ($[x_1 = 1, x_2 = 0], y = 1$) will appear to the system. The appearance of this sample will involve transition of $TA_3$ moving from action Include toward Exclude, as shown in Fig. 6, making the system non-absorbing.

When $T$ clauses have learned the intended pattern $\mathbf{X} = [x_1 = 1, x_2 = 1]$, i.e., $f_{\sum}(\mathcal{C}^i(\mathbf{X})) = T$, then $u_1 = 0$, and thus Type I feedback is blocked for this training sample. In this situation, the TM can only see the training samples of the following:

$$P(y = 0 | x_1 = 1, x_2 = 0) = a, a \in (0, 1) \tag{25}$$
$$P(y = 1 | x_1 = 1, x_2 = 0) = 1 - a,$$
$$P(y = 0 | x_1 = 0, x_2 = 1) = 1,$$
$$P(y = 0 | x_1 = 0, x_2 = 0) = 1.$$

---

[5]In this study, we focus only on positive polarity thus $u_2 > 0$ always holds. When negative polarity is enabled (i.e., when a set of clauses learns sub-patterns with label $y = 0$), $u_2$ becomes 0 when $T$ clauses learn a sample with $y = 0$. However, it remains true that no value of $f_{\sum}(\mathcal{C}^i(\mathbf{X}))$ can make both $u_1$ and $u_2$ equal to 0 simultaneously.

Following the same concept as the proof of Lemma 6, we can conclude that the TM is non-absorbing for the samples in Eq. (25). Clearly, the system is non-absorbing, regardless of the value of $u_1$. Therefore, we can conclude that the TM is non-absorbing for the training samples described in Eq. (24).

Following the same principle, we can also prove that the TM is non-absorbing when other training samples, i.e., $[x_1 = 0, x_2 = 1]$, and $[x_1 = 0, x_2 = 0]$, or their combinations, have wrong labels. We thus can conclude that the TM is non-absorbing for the second type of wrong labels. ∎

So far, we have proven that the TM is non-absorbing when only one type of wrong label exists for the AND operator. It is straightforward to conclude that the TM remains non-absorbing when both types of wrong labels are present. The key reason is that adding both types of wrong labels does not eliminate any transitions between system states in non-absorbing systems. Therefore, the TM is non-absorbing for training samples with general wrong labels for the AND operator. Using the same reasoning, we can extend this conclusion to the XOR and OR operators. Thus, the following theorem holds.

**Theorem 10.** *The TM is non-absorbing given training samples with wrong labels for the AND, OR, and XOR operators.*

**Remark 7.** *The primary reason for the non-absorbing behavior of the TM when wrong labels are present is the introduction of statistically conflicting labels for the same input samples. These inconsistency causes the TAs within a clause to learn conflicting outcomes for the same input due to the corresponding Type I and Type II feedback for label 1 and 0 respectively. When a clause learns to evaluate an input as 1 based on Type I feedback, samples with a label of 0 for the same input prompt it to learn the input as 0 through Type II feedback. This conflict in labels confuses the TM, leading to back-and-forth learning.*

**Remark 8.** *Note that although wrong labels will make the TM not converge (not absorbing with 100% accuracy for the intended logic), via simulations, we find that the TM can still learn the operators efficiently, which has been demonstrated in Section J, especially when the probability of wrong label is small. Interestingly, when the probability of the second type of wrong label is large, TM will consider it as a sub-pattern, and learn it, which aligns with the nature of learning.*

# G  ANALYSIS OF THE TM WITH AN IRRELEVANT INPUT VARIABLE

In this appendix, we examine the impact of irrelevant input noise on the TM. Irrelevant noise refers to an input bit with a random value that does not affect the classification result. For instance, in the AND operator, a third input bit, $x_3$, may appear in the training sample with random 1 and 0 values, but its value does not influence the output of the AND operator. In other words, the output is entirely determined by the values of $x_1$ and $x_2$. Formally, we have:

$$P(y = 1 | x_1 = 1, x_2 = 1, x_3 = 0 \text{ or } 1) = 1, \tag{26}$$
$$P(y = 0 | x_1 = 1, x_2 = 0, x_3 = 0 \text{ or } 1) = 1,$$
$$P(y = 0 | x_1 = 0, x_2 = 1, x_3 = 0 \text{ or } 1) = 1,$$
$$P(y = 0 | x_1 = 0, x_2 = 0, x_3 = 0 \text{ or } 1) = 1.$$

Here $x_3 = 0 \text{ or } 1$ means $P(x_3 = 0) = a$, $P(x_3 = 1) = 1 - a$, $a \in (0, 1)$.

## G.1  CONVERGENCE ANALYSIS OF THE AND OPERATOR WITH IRRELEVANT VARIABLE

**Theorem 11.** *The clauses in a TM can almost surely learn the AND logic given training samples in Eq. (26) in infinite time, when $T \leq m$.*

**Proof:** The proof of Theorem 11 consists of two steps: (1) Identifying a set of absorbing conditions and confirming that the TM, when in these conditions, satisfies the requirements of the AND operator. (2) Demonstrating that any state of the TM that deviates from the conditions defined in step (1) is not absorbing.

The TM will be absorbed when the following conditions fulfill:

1. Condition to block Type I feedback: For any input sample $\mathbf{X} = [x_1 = 1, x_2 = 1, x_3]$, regardless of whether $x_3 = 1$ or 0, the TM has at least $T$ clauses that output 1.

2. Conditions to guarantee no action upon Type II feedback:

   (a) When $x_3$ or $\neg x_3$ appears in a clause in the TM: The literals that are included in the clause for the first two input variables must result in a clause value of 0 for the input samples $\mathbf{X} = [x_1 = 0, x_2 = 1, x_3]$, $\mathbf{X} = [x_1 = 1, x_2 = 0, x_3]$ and $\mathbf{X} = [x_1 = 0, x_2 = 0, x_3]$. This ensures that $C = 0$ for these input samples, regardless of the value of $x_3$, thereby preventing transitions caused by any Type II feedback. The portion of the clause involving the first two input variables can be, e.g., $x_1 \wedge x_2$ or $x_1 \wedge \neg x_1 \wedge x_2$, while the overall clauses can be, e.g., $C = x_1 \wedge x_2 \wedge x_3$, or $C = x_1 \wedge \neg x_1 \wedge x_2 \wedge \neg x_3$, as long as the resulted clause value is 0 for those input samples.

   (b) When $x_3$ or $\neg x_3$ does NOT appear in a clause in the TM: There is no clause that is in the form of $C = x_1$, $C = x_2$, $C = x_1 \wedge \neg x_2$, $C = \neg x_1 \wedge x_2$, $C = \neg x_1$, $C = \neg x_2$, or $C = \neg x_1 \wedge \neg x_2$.

Clearly, when the above conditions fulfill, the system has absorbed because no feedback appears to the system. Additionally, this absorbing state follows AND operator. Based on the statement of the condition to block Type I feedback, there are at least $T$ clauses that output 1 for input sample $\mathbf{X} = [x_1 = 1, x_2 = 1, x_3]$, regardless $x_3 = 1$ or 0. Studying the conditions for Type II feedback, we can conclude that the clause outputs 0 for all input samples $\mathbf{X} = [x_1 = 1, x_2 = 0, x_3]$, $\mathbf{X} = [x_1 = 0, x_2 = 1, x_3]$, or $\mathbf{X} = [x_1 = 0, x_2 = 0, x_3]$. We can then setup the $Th = T$ to confirm the AND logic.

The next step is to show that any state of the TM deviating from the above conditions is not absorbing. To demonstrate this, we can simply confirm that transitions, which might change the current actions of the TAs, will occur due to updates from Type I or Type II feedback.

When literal $x_3$ or literal $\neg x_3$ is included as a part of the clause, the probability for $C = 0$ is non-zero due to the randomness of input variable $x_3$. As a result, Type I Feedback will encourage the TA for the included literal $x_3$ or $\neg x_3$ to move away from its current action, thus preventing the system from becoming absorbing.

For the case where literal $x_3$ or literal $\neg x_3$ is not included in the clause, the system operates purely based on the first two input variables, namely $x_1$ and $x_2$. According our previous analysis for

the noise free AND case (Theorem 1), there is only one absorbing status, which is $C = x_1 \wedge x_2$. However, this absorbing state disappears because Type I feedback will encourage the excluded literal $x_3$ to be included when $x_3 = 1$, and similarly encourage the excluded literal $\neg x_3$ to be included when $x_3 = 0$. Once either $x_3$ or $\neg x_3$ is included, the analysis in the previous paragraph applies, and thus the system is not absorbing.

From the above discussion, it is clear that Type I feedback is the key driver of action changes in non-absorbing cases. If Type I feedback is not blocked, the system cannot reach an absorbing state. Therefore, blocking Type I feedback is critical for achieving convergence. The condition $T < m$ is to guarantee that $T$ should not be greater than the total number of clauses, making it feasible to block Type I feedback. ∎

**Remark 9.** *Due to the existence of the irrelevant input $x_3$, the system requires the functionality of $T$ to block Type I feedback in order to converge. This contrasts with the noise-free case, where the TM will almost surely converge to the AND operator even when Type I feedback is consistently present ($u_1 > 0$).*

### G.2 Convergence Analysis of the OR Operator with Irrelevant Variable

For the OR case, we have

$$P\left(y = 1 | x_1 = 1, x_2 = 1, x_3 = 0 \text{ or } 1\right) = 1, \tag{27}$$
$$P\left(y = 1 | x_1 = 1, x_2 = 0, x_3 = 0 \text{ or } 1\right) = 1,$$
$$P\left(y = 1 | x_1 = 0, x_2 = 1, x_3 = 0 \text{ or } 1\right) = 1,$$
$$P\left(y = 0 | x_1 = 0, x_2 = 0, x_3 = 0 \text{ or } 1\right) = 1.$$

**Theorem 12.** *The clauses in a TM can almost surely learn the OR logic given training samples in Eq. (27) in infinite time, when $T \leq \lfloor m/2 \rfloor$.*

**Proof:** The proof of Theorem 12 follows a similar structure to that of the AND case and involves two steps: (1) Identifying a set of absorbing conditions and verifying that, under these conditions, the TM satisfies the requirements of the OR operator. (2) demonstrating that any state of the TM deviating from these conditions is not absorbing.

1. Condition to block Type I feedback: For any input sample $\mathbf{X} = [x_1 = 1, x_2 = 1, x_3]$, $\mathbf{X} = [x_1 = 1, x_2 = 0, x_3]$, and $\mathbf{X} = [x_1 = 0, x_2 = 1, x_3]$ regardless of whether $x_3 = 1$ or $0$, the TM has at least $T$ clauses that output 1.

2. Conditions to guarantee no action upon Type II feedback:

   (a) When $x_3$ or $\neg x_3$ appears in a clause in the TM: The literals included in the clause for the first two input variables must ensure a clause value of 0 for the input samples $\mathbf{X} = [x_1 = 0, x_2 = 0, x_3]$. This is to guarantee that $C = 0$ for those input samples, irrespective of the value of $x_3$, thereby preventing any transitions caused by Type II feedback. The portion of the clause involving the first two input variables can take the form such as $x_1, x_1 \wedge \neg x_2, x_1 \wedge x_2, x_1 \wedge \neg x_1 \wedge x_2$. Correspondingly, the overall clauses can take the form such as $C = x_1 \wedge \neg x_3$, $C = x_1 \wedge \neg x_2 \wedge x_3$, $C = x_1 \wedge x_2 \wedge x_3$, or $C = x_1 \wedge \neg x_1 \wedge x_2 \wedge \neg x_3$, as long as the resulted clause value is 0 for those input samples.

   (b) When $x_3$ or $\neg x_3$ does not appear in a clause in the TM: There are no clauses with literal(s) in only negated form, such as $C = \neg x_1$, $C = \neg x_2$, or $C = \neg x_1 \wedge \neg x_2$.

Clearly, when the above conditions fulfill, the system is absorbing because no feedback triggers state transitions in the system. Additionally, this absorbing state adheres to the OR operator. Based on the condition required to block Type I feedback, there are at least $T$ clauses that output 1 for input sample $\mathbf{X} = [x_1 = 1, x_2 = 1, x_3]$, $\mathbf{X} = [x_1 = 1, x_2 = 0, x_3]$, or $\mathbf{X} = [x_1 = 0, x_2 = 1, x_3]$ regardless of whether $x_3 = 1$ or $0$. Analyzing the conditions for Type II feedback, we find that the clause outputs 0 for all input samples $\mathbf{X} = [x_1 = 0, x_2 = 0, x_3]$. We can then setup the $Th = T$ to confirm the OR logic.

The next step is to demonstrate that any state of the TM that deviates from the above conditions outlined above is not absorbing. To do this, we can confirm that transitions which may alter the current actions of the TAs will occur due to updates from Type I and Type II feedback.

When literal $x_3$ or literal $\neg x_3$ is included in the clause, there is a non-zero probability for $C = 0$ due to the randomness of the input variable $x_3$. In this case, Type I Feedback will move the included literal $x_3$ or $\neg x_3$ towards action Exclude, preventing the system from being absorbing.

For the case where literal $x_3$ or literal $\neg x_3$ is not included as a part of the clause, the system operates purely based on the first two input variables, namely $x_1$ and $x_2$. Based on our previous analysis for the noise free OR case shown in Lemma 2, the system is non-absorbing. This non-absorbing behavior can also lead the system to a state where the excluded literal, either $x_3$ or $\neg x_3$, is encouraged to be included. For example, if the TM has a clause $C = x_1 \wedge x_2$, upon a training sample $\mathbf{X} = [x_1 = 1, x_2 = 1, x_3 = 0]$, the Type I feedback will encourage the excluded literal $\neg x_3$ to be included. Once one of the excluded literal, $x_3$ or $\neg x_3$, is included, the analysis in the previous paragraph applies, meaning the system is not absorbing.

Clearly, if Type I feedback is not blocked, the system will not be absorbing. As blocking Type I feedback is critical, condition $T \leq \lfloor m/2 \rfloor$ is necessary, refer to Lemma 4. ∎

When $T$ clauses have learned the intended sub-patterns of OR operation, the Type I feedback will be blocked. At the same time, Type II feedback will eliminate all clauses that output 1 for input sample following $\mathbf{X} = [x_1 = 0, x_2 = 0, x_3]$, removing false positives. At this point, the system has converged. The presence of $x_3$ does not change the convergence feature, but it adds more dynamics to the TM.

### G.3 CONVERGENCE ANALYSIS OF THE XOR OPERATOR WITH IRRELEVANT VARIABLE

**Theorem 13.** *The clauses in a TM can almost surely learn the XOR logic given training samples in Eq. (28) in infinite time, when $T \leq \lfloor m/2 \rfloor$.*

$$
\begin{aligned}
P(y = 0 | x_1 = 1, x_2 = 1, x_3 = 0 \text{ or } 1) &= 1, \\
P(y = 1 | x_1 = 1, x_2 = 0, x_3 = 0 \text{ or } 1) &= 1, \\
P(y = 1 | x_1 = 0, x_2 = 1, x_3 = 0 \text{ or } 1) &= 1, \\
P(y = 0 | x_1 = 0, x_2 = 0, x_3 = 0 \text{ or } 1) &= 1.
\end{aligned}
\tag{28}
$$

The proof for XOR follows the same principles as the AND and OR cases, and therefore, we do not present it explicitly here.

### G.4 CONVERGENCE ANALYSIS OF THE OPERATORS WITH MULTIPLE IRRELEVANT VARIABLES

In the previous subsections, we demonstrated that if a single irrelevant bit is present in the training samples, the system will almost surely converge to the intended operators. This conclusion can be readily extended to scenarios involving multiple irrelevant variables. Here, "multiple irrelevant variables" refers to the presence of additional variables, beyond $x_3$, in the training samples that do not contribute to the classification.

**Theorem 14.** *The clauses in a TM can almost surely learn the 2-bit AND logic given training samples with q irrelevant input variables in infinite time, $q > 0$, when $T \leq m$.*

**Theorem 15.** *The clauses in a TM can almost surely learn the 2-bit XOR and OR logic given training samples with q irrelevant input variables in infinite time, $q > 0$, when $T \leq \lfloor m/2 \rfloor$.*

**Proof:** The proofs of Theorems 14 and 15 are straightforward. It suffices to verify whether the conditions for blocking Type I and Type II feedback remain valid when multiple irrelevant variables are present.

The condition for blocking Type I feedback remains valid because Type I feedback is only determined by the first two input bits and is not a function of the irrelevant variables. For Type II feedback, its effect depends on whether the literals for the irrelevant inputs are present in the clause. In cases where the literals of the irrelevant bits are not included in the clause, the analysis holds, as those literals are absent. When the literals of the irrelevant bits are included, their number does not impact the analysis. This is because the clause value is entirely determined by the first two bits, and the clause value remains $C = 0$, regardless of the number of irrelevant variables. ∎

# H    CONVERGENCE ANALYSIS OF TM IN $k$-BIT CASES

## H.1    PROOF OF THEOREM 6

**Proof:** In this setting, the training samples are noise-free, and there exists exactly one intended sub-pattern to be learned among the $2^k$ possible combinations. The conditions $u_1 > 0$ and $u_2 > 0$ ensure that $T$ has no effect. In particular, training samples are always presented to the TM, and no sample type is suppressed.

To establish Theorem 6, we avoid enumerating all possible states in literal level and instead group the clause forms into the following three categories:

(1) **Exact match:** The clause matches the intended sub-pattern exactly (e.g., $C = x_1 \wedge x_2$ in the 2-bit AND case). Such a clause outputs 1 when the intended sub-pattern is presented.

(2) **Partial match:** The clause matches a strict subset of the intended sub-pattern (e.g., $C = x_1$ in the 2-bit AND case). Such clauses also output 1 when the intended sub-pattern is presented.

(3) **Non-match:** The clause matches neither the intended sub-pattern nor any of its subsets (e.g., $C = \neg x_1$ in the 2-bit AND case). Such clauses output 0 when the intended sub-pattern is presented.

We show that clauses of type (1) are absorbing, whereas clauses of types (2) and (3) are non-absorbing. Consequently, the system possesses a unique absorbing clause form corresponding to the intended sub-pattern.

**Type (1): Exact match is absorbing.**

A clause of type (1) is absorbing because once it matches the intended sub-pattern, no transition can alter its form. Under Type I feedback (i.e., when the unique positive sample with $y = 1$ is presented), the clause outputs 1. All included literals evaluate to 1, and all excluded literals evaluate to 0. The Type I feedback table prescribes *reward* for both included and excluded literals in this situation, meaning that no TA changes its action. Therefore, the clause remains unchanged.

Under Type II feedback (i.e., when samples with $y = 0$ are presented), the clause outputs 0, since it matches the positive sub-pattern exactly and thus rejects all negative samples. According to the Type II feedback rules, no updates are applied when the clause output is 0.

Thus, neither Type I nor Type II feedback can modify the clause. Type (1) clauses are therefore absorbing.

**Type (2): Partial match is non-absorbing.**

We next show that clauses of type (2) are non-absorbing. Under Type I feedback, such clauses output 1. Any literal that *should* be part of the exact match but is currently in the *exclude* action receives a penalty (Type I table, case $C = 1$, literal value $= 1$ on the excluded side), encouraging a transition from *exclude* to *include*. Thus the clause will eventually change its form.

To prove non-absorbency, it suffices to exhibit a single transition with non-zero probability. Nevertheless, we also examine Type II feedback for completeness. A transition under Type II feedback occurs whenever the clause outputs 1 and an excluded literal has value 0. Such literals receive a penalty and are encouraged to shift from *exclude* to *include*. This situation can arise for partial-match clauses. For example, in the 2-bit AND case, the clause $C = x_1$ outputs 1 for the negative sample $x_1 = 1, x_2 = 0$ (with $y = 0$). The excluded literal corresponding to $x_2$ takes value 0, so Type II feedback encourages it to transition to *include*. The clause therefore moves toward $x_1 \wedge x_2$.

A special instance of this category is the empty clause, where all literals are in the *exclude* action. Such a clause outputs 1 for all samples by definition in the training process. Under Type I feedback, all literals belonging to the intended sub-pattern are encouraged to move from *exclude* to *include*. Under Type II feedback, every literal with value 0 is likewise encouraged to transition from *exclude* to *include*. Hence the empty clause cannot remain unchanged.

Thus, clauses of type (2) are non-absorbing.

**Type (3): Non-match is non-absorbing.**

Finally, clauses of type (3) are also non-absorbing. Such clauses output $0$ for the intended sub-pattern. Under Type I feedback, when $C = 0$, the Type I transition table prescribes penalties for all included literals, encouraging them to move from *include* to *exclude*. Hence the clause cannot remain in its current form.

Under Type II feedback, depending on the clause, it is possible for the clause to output $1$ for some negative sample. When this occurs, any excluded literal with value $0$ receives a penalty and is encouraged to shift from *exclude* to *include*. For example, in the 2-bit AND case, the clause $C = \neg x_1$ outputs $1$ on the sample $x_1 = 0, x_2 = 0$ (with $y = 0$). Type II feedback then penalizes both excluded literals, pushing them toward inclusion. The clause therefore changes form. Thus, clauses of type (3) are non-absorbing.

Although the clause space grows combinatorially, grouping clauses into the three categories above reveals that absorbing/non-absorbing behavior is identical within each category, regardless of the specific form of a clause. Since type (1) clauses are absorbing whereas types (2) and (3) are non-absorbing, the TM has a unique absorbing state: the exact match of the intended sub-pattern. Consequently, convergence is guaranteed given infinite time. ∎

## H.2 PROOF OF THEOREM 7

The proof of Theorem 7 follows the same structure as the proof of Theorem 2. We show that when two or more sub-patterns appear in the training samples, the absorbing clauses that exist in the single–sub-pattern setting disappear. In other words, the system no longer possesses any absorbing state as in the single sub-pattern case. To restore an absorbing state, the role of the threshold $T$ becomes critical. Analogous to the OR proof, we first show that the system becomes non-absorbing when multiple sub-patterns are present, and then show how to configure $T$ so that convergence is guaranteed.

We begin by showing that if two or more sub-patterns occur in the training samples, the system is non-absorbing. In the $k$-bit setting, the existence of multiple sub-patterns implies that there is at least one bit whose value differs across sub-patterns. That is, the bit is $1$ in one sub-pattern but $0$ in another. For example, in the 2-bit OR case, between the sub-patterns $(1, 0)$ and $(1, 1)$, $x_2$ is a conflicting bit: it is $0$ in the former sub-pattern and $1$ in the latter. We refer to such bits as *conflicting bits*. Because conflicting bits are present, the absorbing clauses that existed in the single–sub-pattern setting can no longer remain absorbing. The core argument is to show that any clause that was absorbing in the single–sub-pattern case ceases to be absorbing once additional sub-patterns, and thus conflicting bits, appear.

Without loss of generality, assume that the third bit is the conflicting bit in the multiple–sub-pattern setting. Suppose the clause has reached the absorbing form

$$([x_i = *, \, x_3 = 0], \, y = 1),$$

where $i \in \{1, \ldots, k\} \setminus \{3\}$, and "$*$" denotes an arbitrary assignment to the non-conflicting bits. To show that the system is non-absorbing, we must demonstrate that this clause loses its absorbing property once an additional sub-pattern

$$([x_i = *, \, x_3 = 1], \, y = 1)$$

is introduced.

In the absorbing form, the clause must include the literal $\neg x_3$ in order to match the sub-pattern exactly. However, when training samples corresponding to

$$([x_i = *, \, x_3 = 1], \, y = 1)$$

appear, the clause will output $0$ due to the conflicting bit $x_3 = 1$. Consequently, Type I feedback will reinforce the exclusion of the currently included literals, regardless of their literal value. This process breaks the absorbing condition, meaning the clause is no longer absorbing.

By the same reasoning, if the absorbing sub-pattern were

$$([x_i = *, \, x_3 = 1], \, y = 1),$$

then introducing samples of

$$([x_i = *, \ x_3 = 0], \ y = 1)$$

would likewise eliminate the absorbing property. Therefore, the specific literal value (0 or 1) of the conflicting bit in the original absorbing state does not matter. Moreover, as additional sub-patterns introduce more conflicting bits, the clauses remain non-absorbing. This clearly indicates that when $u_1 > 0$ and $u_2 > 0$, the clauses are non-absorbing in the presence of multiple sub-patterns. Consequently, the functionality of $T$ must be enabled to ensure that the clauses become absorbing, by preventing Type I feedback from being triggered.

Similar to Lemma 3, we obtain the following result.

**Lemma 8.** *The system is absorbed if and only if (1) the number of clauses for each sub-pattern reaches T, and (2) no clause outputs* 1 *for training samples with label* 0*, i.e., no false positives occur.*

The proof of Lemma 8 is immediate. Condition (1) blocks all Type I feedback, and the argument follows directly from Lemma 3. Condition (2) ensures that no transitions are triggered by Type II feedback. When Type I feedback is blocked and Type II feedback induces no further transitions, the system is absorbing.

To establish Theorem 7, it remains to verify that the condition $T \le \lfloor \frac{m}{e} \rfloor$ is sufficient to guarantee that all sub-patterns are covered by at least $T$ clauses. Since $e$ denotes the number of sub-pattern clusters, that is, the number of clusters in which all sub-patterns share one or more bits in common, we may represent each such cluster with a single clause. Thus, if $T \le \lfloor \frac{m}{e} \rfloor$, then at least $T$ clauses can be assigned to each cluster. Therefore, the convergence requirement in condition (1) is satisfied. We can now prove Theorem 7.

**Proof:** From the above arguments, we observe that if

$$T \le \left\lfloor \frac{m}{e} \right\rfloor$$

holds, then Type I feedback will eventually be completely blocked, and Type II feedback will eventually produce only "inaction" responses. In this situation, no further state transitions occur, and the system reaches an absorbing state. Prior to absorption, the system may move back and forth among intermediate states, but it will not become absorbed until the above condition is met.

Once absorbed, every sub-pattern with label 1 will have at least $T$ clauses assigned to it, while the sub-pattern with label 0 will have none. This means that the TM can almost surely learn the intended multiple sub-patterns in infinite time. Once learnt, for inference, it can classify the class by setting the threshold $Th = T$. This completes the proof. ∎

### H.3 PROOF OF THEOREM 8 AND THEOREM 9

**Proof of Theorem 8:** The proof of Theorem 8 consists of two steps: (1) Identifying a set of absorbing conditions and confirming that the TM, when in these conditions, satisfies the requirements of the intended unique sub-pattern. (2) Demonstrating that any state of the TM that deviates from the conditions defined in step (1) is not absorbing. Without lose of generality, we consider a general $k + q$ input Boolean vector with $k$-bit useful bits plus $q$ bit irrelevant bits, as $\mathbf{X} = [x_1, x_2, \ldots x_k, x_{k+1}, \ldots, x_{k+q}]$.

The TM will be absorbed when the following conditions are satisfied:

1. **Condition to block Type I feedback:** For any input sample belonging to the intended sub-pattern, and for any bits $x_j \in \{0, 1\}$ with $j \in [k + 1, k + q]$, the TM must have at least $T$ clauses that output 1.

2. **Condition to guarantee no transitions under Type II feedback:** No clause outputs 1 for training samples with label 0. That is, no false positives occur.

Clearly, when the above conditions are satisfied, the system is absorbed, since no further feedback is produced. Moreover, this absorbing state corresponds to the intended sub-pattern. It is worth noting that the irrelevant bits do not induce any transitions under Type II feedback once no clause outputs 1

for training samples with label 0, i.e., once no false positives occur. This is because the absence of false positives implies that all samples labeled 0 produce a clause output of 0, fully determined by the learned sub-patterns from the $k$ useful bits. Since the irrelevant bits do not influence the label, they likewise do not affect the clause output once learning has converged. In other words, regardless of the values of the irrelevant bits, the clause output is already determined by the $k$-bit useful pattern, giving 0 in this case. Hence, the randomness of the irrelevant bits cannot change the clause output from 0. According to the Type II feedback table, no transitions occur when $C = 0$.

The next step is to show that any state of the TM that violates these conditions is not absorbing. To establish this, it suffices to confirm that such states will trigger transitions, arising from either Type I or Type II feedback, that modify the current form of the clauses.

We begin with Type I feedback. Before this feedback is blocked by the threshold $T$, the random irrelevant bits make the system non-absorbing. The reason is analogous to the conflict-bit argument presented in Subsection H.2. The conflict bits take the value 1 for some sub-patterns and 0 for others, pushing the system to learn in inconsistent directions. The random irrelevant bits behave in exactly the same manner. Therefore, the mechanism provided by $T$ is essential. The condition $T < m$ guarantees that $T$ does not exceed the total number of clauses, ensuring that blocking Type I feedback is feasible.

For Type II feedback, any false positive will trigger transitions that move literals with value 0 (and currently excluded) toward the include side until all false positives have been eliminated. As long as false positives remain, Type II feedback continues to update the system, and thus the state is not absorbing.

Based on the above discussion, it follows that any violation of the listed conditions prevents the system from being absorbing. Therefore, only the listed conditions fulfill the absorbing states, which covers the intended sub-pattern. ∎

**Proof of Theorem 9:** The proof of Theorem 9 follows the same structure and reasoning as the proof of Theorem 8. In particular, we identify two conditions that ensure the system is absorbing.

1. **Condition to block Type I feedback:** For any input samples of fixed length $n$, containing multiple intended sub-patterns where the $i$-th sub-pattern consists of $k_i$ informative bits and $n - k_i$ irrelevant bits (with the positions of both bit types arbitrary), the TM must have at least $T$ clauses that output 1 for each intended sub-pattern.

2. **Condition to prevent transitions under Type II feedback:** No clause outputs 1 for training samples with label 0, i.e., no false positives occur.

Different from the condition $T \leq m$ in Theorem 8, here we require $T \leq \lfloor m/e \rfloor$. The arguments showing that these conditions lead to an absorbing state, as well as the arguments establishing that any other clause configuration is non-absorbing, are identical to those presented in the proof of Theorem 8. We therefore omit the details to avoid repetition. ∎

# I  EXPERIMENT RESULTS OF 2-BIT CASES WITH NOISE-FREE TRAINING SAMPLES

To validate the theoretical analyses, we here present the experiment results[6] for both the AND and the OR operators.

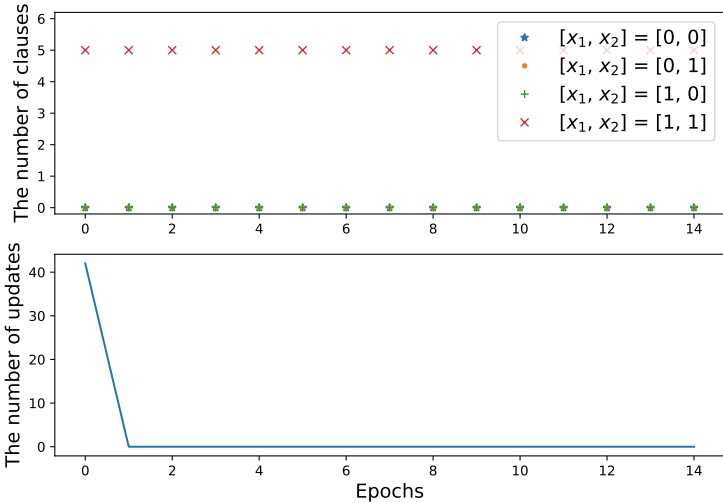

Figure 8: The convergence of a TM with 7 clauses when $T = 5$ for the AND operator.

Figure 8 shows the convergence of TM for the AND operator when $m = 7$, $T = 5$, $s = 4$, and $N = 50$ ($N$ is the number of states for each action in each TA). More specifically, we plot the number of clauses that learn the AND operator, namely, $x_1 = x_2 = 1$, and the number of system updates as a function of epochs. From these figures, we can clearly see that after a few epochs, the TM has 5 clauses that learn the AND operator and then the system stops updating because no update is triggered anymore. Note that if we control $T$ so that $u_1 > 0$ always holds, all clauses will converge to the AND operator, which has been validated via experiments. These observations confirm Theorem 1. Although the theorem says it may require infinite time in principle, the actual convergence can be much faster.

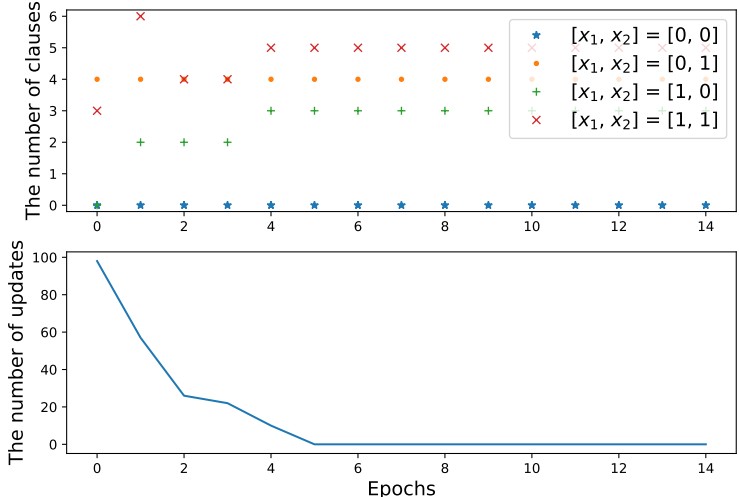

Figure 9: The convergence of a TM with 7 clauses when $T = 3$ for the OR operator.

---

[6]The code for validating the convergence can be found at https://github.com/JaneGlim/Convergence-of-Tsetline-Machine-for-the-AND-OR-operators.

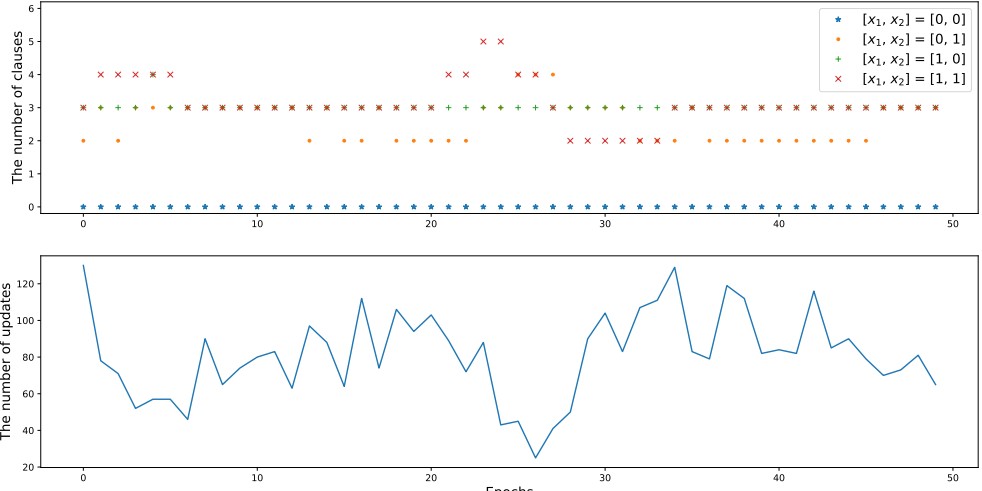

Figure 10: The behavior of a TM with 7 clauses when $T = 4$ for the OR operator.

In Fig. 9, we illustrate the number of clauses in distinct sub-patterns when we employ $m = 7$, $T = 3$, $s = 4$, and $N = 50$ for the OR operator. Based on the analytical result, i.e., Theorem 2, the system will be absorbed, where each sub-pattern will have at least 3 clauses and no update will happen afterwards. From the figure, we can clearly observe that after a few epochs, the system becomes indeed absorbed as no updates are observed. When absorbed, the three intended sub-patterns have 3, 4, 5 clauses to represent them respectively, while the unintended sub-pattern has 0 clause, which is consistent with the theorem. Indeed, the list of the converged clauses are: $C_1 = x_1$, $C_2 = x_1$, $C_3 = x_2$, $C_4 = x_1 \wedge \neg x_2$, $C_5 = x_1 \wedge x_2$, $C_6 = x_2$, and $C_7 = \neg x_1 \wedge x_2$, explaining the number of converged clauses in different sub-patterns shown in the figure. Clearly, in this example, some clauses, i.e., $C_1$, $C_2$, $C_3$ and $C_6$, can each cover multiple sub-patterns. This indicates that in real world applications, if distinct sub-patterns have certain bits in common, which can be used to differentiate it from other classes, it is possible for TM to learn those bits as joint features, confirming the efficiency of the TM.

Note that there are many other possible absorbing states that are different from the shown example, which have been observed when we run multiple instances of the experiments. As long as each intended sub-pattern is represented by at least $T$ clauses in the OR operator, the system converges.

In Fig. 10, the configuration is identical to that in Fig. 9 except that $T = 4$. In this case, as stated in Remark 2, the system will not become absorbing, but will still cover the intended sub-patterns with high probability. From this figure, we can observe that each intended sub-pattern is represented by at least two clauses, and that the unintended sub-pattern has zero clause. At the same time, the TAs do not stop updating their states, which can be seen in the bottom figure. It is worth mentioning that we have occasionally observed in other rounds of experiments, that one intended sub-pattern is covered by only 1 clause. In this case, it is still possible to set up $Th \geq 1$ to have successful classification. Nevertheless, there is no guarantee that each intended sub-pattern will be represented by at least one (or $Th$) clause(s) in this configuration, thus no guaranteed successful classification.

## J  EXPERIMENT RESULTS OF 2-BIT CASES WITH NOISY TRAINING SAMPLES

We present the experimental results for the operators under noisy conditions. First, we show the results when incorrect labels are present, followed by the results involving irrelevant variables. The final subsection addresses a case where both incorrect labels and irrelevant variables are present.

### J.1  EXPERIMENT RESULTS FOR WRONG LABELS

To evaluate the performance of the TM when exposed to mislabeled samples, we introduced incorrectly labeled data into the system. The key observation is that the TM does not converge to the intended logic, meaning it does not absorb into a state where the correct logic is consistently represented. However, with carefully chosen hyperparameters, the TM can still learn the intended logic with high probability.

To demonstrate the TM's behavior, we first conduct experiments on the AND operator, which satisfies the following equation:

$$P\left(y = 1 | x_1 = 1, x_2 = 1\right) = 95\%, \tag{29}$$
$$P\left(y = 0 | x_1 = 1, x_2 = 0\right) = 1,$$
$$P\left(y = 0 | x_1 = 1, x_2 = 0\right) = 1,$$
$$P\left(y = 0 | x_1 = 0, x_2 = 0\right) = 1.$$

In this scenario, 5% of the input samples that should be labeled as 1 were incorrectly labeled as 0. To train the TM and evaluate its performance, we used the following hyperparameters: $m = 7$, $T = 4$, $Th = 2$, $s = 3$, and $N = 100$. The experimental results indicate that the TM does not converge, as the number of updates does not approach zero, which is consistent with our theoretical findings. However, an analysis of the learned clauses shows that in most runs the TM is still able to capture the AND operator after training, since at least two clauses conform to the AND pattern, even though the convergence is not achieved.

Here we also conduct experiments on the OR operator, which satisfies the following equation:

$$P\left(y = 1 | x_1 = 1, x_2 = 1\right) = 90\%, \tag{30}$$
$$P\left(y = 1 | x_1 = 1, x_2 = 0\right) = 90\%,$$
$$P\left(y = 1 | x_1 = 1, x_2 = 0\right) = 90\%,$$
$$P\left(y = 0 | x_1 = 0, x_2 = 0\right) = 1.$$

In this scenario, 10% of the input samples that should be labeled as 1 were incorrectly labeled as 0. To train the TM and evaluate its performance, we used the following hyperparameters: $m = 7$, $T = 4$, $Th = 2$, $s = 3$, and $N = 100$. Fig. 11 shows the number of updates and the number of clauses that learn distinct sub-patterns, as a function of epochs. As shown in Fig. 11, the number of updates is big, and thus the system did not converge. Nevertheless, when examining the number of clauses associated with each sub-pattern, we observed that each sub-pattern was covered by at least two clauses, ensuring that the OR operator remained valid. Similar results were observed in experiments conducted on the XOR operators.

Interestingly and understandably, when the proportion of mislabeled samples increases to an extreme level, where inputs that should be labeled as 0 are instead labeled as 1, the TM begins to treat the noise as a sub-pattern. For instance, consider the AND operator with input $\mathbf{X} = [x_1 = 0, x_2 = 1]$, which is mislabeled as 1 in 90% of the cases, as shown in Eq. (31). Using the hyperparameters $m = 7$, $T = 3$, $s = 3.0$, and $N = 100$, we observed from experiments that the TM generates three clauses with an output of 1 for $\mathbf{X} = [x_1 = 0, x_2 = 1]$ and another three clauses with an output of 1 for $\mathbf{X} = [x_1 = 1, x_2 = 1]$. This behavior indicates that the TM has incorporated the noise as a learned sub-pattern. Such outcomes align with the TM's underlying principle of learning, where it identifies and models sub-patterns associated with the label 1.

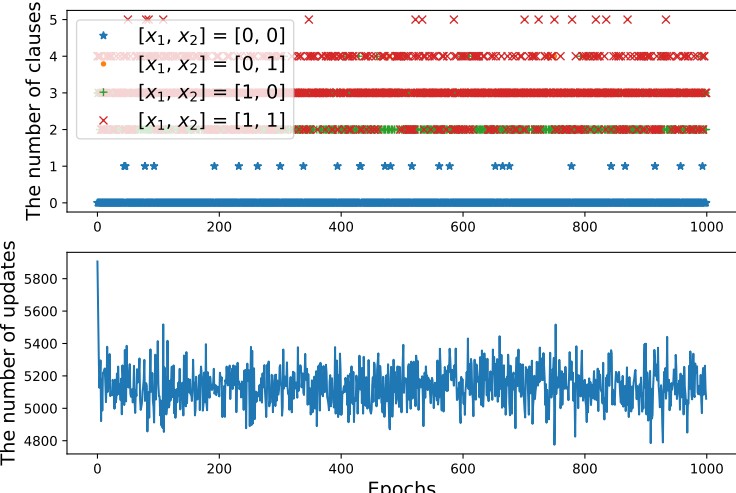

Figure 11: The behavior of TM when $m = 7, T = 4$ for the OR operator with wrong training labels.

$$P\left(y = 1 | x_1 = 1, x_2 = 1\right) = 1,$$
$$P\left(y = 0 | x_1 = 1, x_2 = 0\right) = 1, \tag{31}$$
$$P\left(y = 0 | x_1 = 0, x_2 = 1\right) = 10\%,$$
$$P\left(y = 0 | x_1 = 0, x_2 = 0\right) = 1.$$

### J.2 EXPERIMENT RESULTS FOR IRRELEVANT VARIABLE

To confirm the convergence property of TM with irrelevant variable, we setup the experiments for the AND, OR, and XOR operators when one irrelevant variable, namely, $x_3$, exists. The probability of $x_3$ being 1 in the training and testing samples is 50%.

For the AND operator, we use the hyperparameters $m = 5, T = 2, s = 3, Th = 2$, and $N = 100$. Fig. 12 illustrates the convergence of TM for the AND operator in the presence of an irrelevant bit. The results confirm that the TM can correctly learn the AND operator without uncertainty, validating the correctness of Theorem 11.

Interestingly, upon convergence, the form of the included literals varies. For instance, with the aforementioned hyperparameters, we observe that the converged TM includes two clauses of the form $x_1 \wedge x_2 \wedge x_3$ and another two clauses of the form $x_1 \wedge x_2 \wedge \neg x_3$. This suggests that, instead of excluding the irrelevant bit $x_3$, the TM includes at least $T$ clauses containing $x_3$ and at least $T$ clauses containing $\neg x_3$, which ensures correct classification regardless of the value of $x_3$. However, when the hyperparameters are set to $m = 1, T = 1, s = 3, Th = 1$, and $N = 100$, where only a single clause exists in the TM, the converged clause takes the form $x_1 \wedge x_2$, excluding the literals $x_3$ and $\neg x_3$.

As $T$ increases ($T > m/2$), we observe that convergence becomes challenging. This difficulty arises because the TM cannot simultaneously learn $T$ clauses containing $x_3$ and another $T$ clauses containing $\neg x_3$. In such cases, the TM must rely on $T$ clauses in the form $x_1 \wedge x_2$ to achieve convergence, which can be particularly demanding.

For the OR operator, we use the hyperparameters $m = 5, T = 2, s = 3, Th = 2$, and $N = 100$. Figure 13 illustrates the convergence of the TM for the OR operator in the presence of an irrelevant bit. The results confirm that TM successfully learns the OR operator without ambiguity, validating the correctness of Theorem 12. The results also confirm that the TM is capable of presenting two sub-patterns jointly.

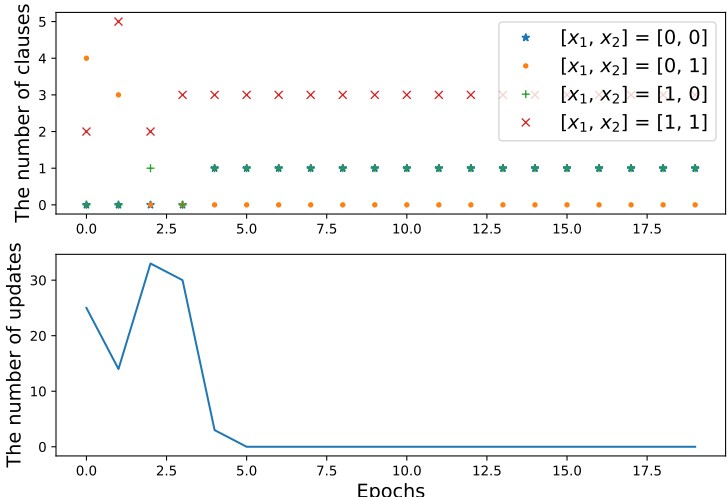

Figure 12: Convergence of TM when $m = 5$, $T = 2$ for the AND operator with an irrelevant variable.

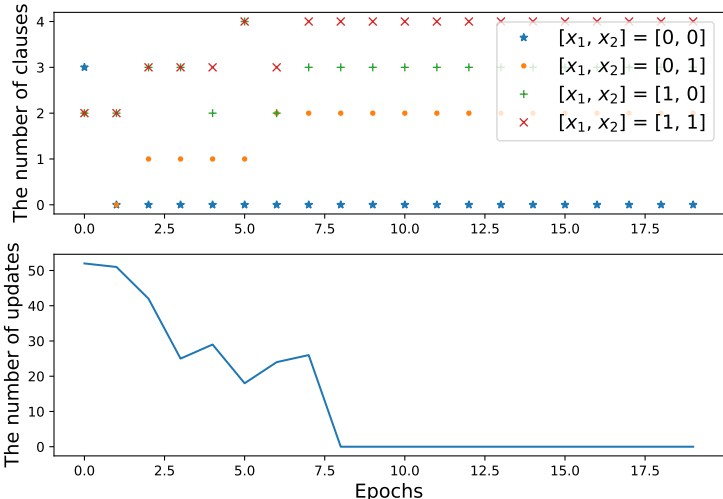

Figure 13: Convergence of TM when $m = 5$, $T = 2$ for the OR operator with an irrelevant variable.

Indeed, the OR operator has multiple absorbing states, corresponding to multiple clause forms. Some clause forms may include $x_3$ or $\neg x_3$, depending on the hyperparameter configuration. Regardless of the value of $x_3$, as long as the vote sum of the clauses is greater than or equal to $T$, the correct classification can be guaranteed.

We have also studied the XOR operator. The convergence instance is shown in Fig. 14, confirming Theorem 13. Here we use $m = 7$, $T = 2$, $s = 3$, $Th = 2$.

### J.3 EXPERIMENT RESULTS FOR BOTH WRONG LABELS AND IRRELEVANT VARIABLES

In this experiment, we assess the performance of the TM in the presence of both mislabeled data and irrelevant variables. Specifically, we evaluate the TM's ability to learn the XOR operator when 40%

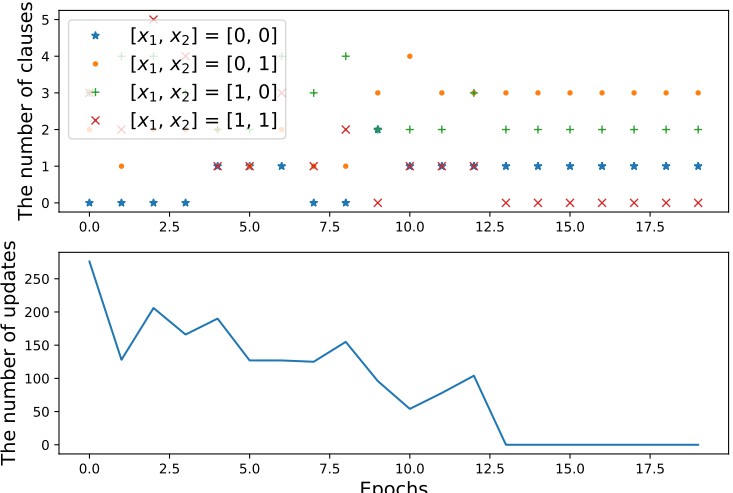

Figure 14: Convergence of TM when $m = 7$, $T = 2$ for the XOR operator with an irrelevant variable.

of the samples are incorrectly labeled, and 10 irrelevant variables are added. The input comprises 12 bits, with only the first two bits determining the output based on the XOR logic.

The hyperparameters are configured as follows: $m = 20$, $T = 15$, $s = 3.9$, and $N = 100$ with polarity enabled. Experimental results reveal that the TM successfully learns the XOR operator in 99% of 200 independent runs. These findings demonstrate the robustness of the TM training in noisy environments.

In another experiment, we configured the TM to learn a noisy XOR function with 2 useful input bits and 18 irrelevant input bits (hyper parameters: $N = 128$, $m = 20$, $T = 10$, $s = 3$, label noise 0.1). Remarkably, the TM was still able to learn the XOR operator with 100% accuracy using just 5000 training samples. If all possible input combinations were required in the training samples, it would require $2^{20} = 1048576$ samples. Clearly, the TM does not rely on the entire combinatorial input space to learn effectively.

When many variables are irrelevant, the training set may not cover all possible input combinations due to the exponential size of the input space. Although not yet theoretically established, polynomial-sized training sets appear sufficient for the TM, as confirmed by experiments. This is because each TA within a clause updates independently once the clause value and the literal value are determined by a sample. The resulting Type I and Type II transitions are then fully specified. Consequently, the TM does not need to observe every combination of irrelevant inputs. It only requires enough samples to reveal their statistical irrelevance, which triggers the appropriate TA transitions. Furthermore, the influence of irrelevant bits can be neutralized when $T$ clauses capture their negated form and $T$ clauses capture their original form for the same sub-pattern. Together, these properties allow the TM to learn effectively without exhaustive coverage of the input space.

## K    EXPERIMENT RESULTS FOR THE $k$-BIT CASES

### K.1    EXPERIMENT RESULTS FOR NOISE FREE CASE

In this experiment[7], we tested the TM on a 5-bit scenario using both a single sub-pattern and multiple sub-patterns. For the single sub-pattern case, the pattern (1, 0, 1, 0, 1) was used with parameters $m = 3$, $T = 5$, $s = 3$, $Th = 3$, and $N = 100$. Here we configure $T > m$ deliberately to make $u_1 > 0$ always true. For the multiple sub-patterns case, the pattern (1, 0, 1, 0, 0) was added in addition with $m = 5$, $T = 2$, $s = 3$, $Th = 2$, and $N = 100$. In all cases, the TM achieved 100% convergence to the correct intended sub-pattern(s), demonstrating its reliable learning capability in the noise-free case.

### K.2    EXPERIMENT RESULTS WITH IRRELEVANT BITS

We first evaluated convergence in the single–sub-pattern case, where each sample contains 8 bits and includes one intended sub-pattern, (1, 0, 1, 0, x, x, x, x), with x denoting irrelevant bits. The hyperparameters for this experiment were configured as $m = 3$, $T = 2$, $s = 3$, $Th = 2$, and $N = 100$.

For the multiple–sub-pattern case, we first introduced an additional intended sub-pattern, (x, x, x, x, 0, 1, 0, 1), and used the configuration $m = 5$, $T = 2$, $s = 3$, $Th = 2$, and $N = 100$. Thereafter, in order to test when the informative bits are unbalanced in numbers among sub-patterns, we replaced (x, x, x, x, 0, 1, 0, 1) by (x, x, x, x, x, 1, 0, 1).

All experiments demonstrated 100% convergence to the correct intended sub-pattern(s), confirming that the TM consistently identifies and converges to the desired patterns. These results highlight the robustness of the TM for samples with irrelevant bits.

### K.3    EXPERIMENT RESULTS FOR BOTH WRONG LABELS AND IRRELEVANT BITS

To illustrate the performance of the TM in this case, we directly use the Noisy XOR problem and take its results from the literature (Tunheim et al., 2023). The Two-dimensional (2D) Noisy XOR dataset consists of 4 × 4 single-channel Boolean images. Figure 15 shows the patterns for Class 1 (blue) and Class 0 (orange), positioned in the middle of the two upper rows, with x's representing random Boolean values. The dataset is balanced, with equal numbers of examples for each class and sub-pattern. Class 1 corresponds to a diagonal line, while Class 0 represents either a horizontal or vertical line. Each image contains 4 informative bits and 12 irrelevant bits. To test robustness against label noise, 40% of the training labels are randomly inverted.

Using 2,500 training samples and 8,192 test samples, the TM implementation[8] achieves a mean test accuracy of 99.99% (Tunheim et al., 2023), demonstrating strong robustness in the presence of noisy labels and irrelevant features.



Figure 15: Patterns representing Class 1 (blue) and Class 0 (Orange) for the 2D Noisy XOR dataset (Tunheim et al., 2023).

---

[7] The code for validating the convergence can be found at https://github.com/JaneGlim/Convergence-of-Tsetline-Machine-for-the-AND-OR-operators.

[8] Here a Convolutional TM (CTM) is employed, where the learning principle is identical to the TM used in this work. The difference is that CTM processes 2-D images through patches.

## L    APPLICATION EXAMPLES OF TM

Although our manuscript focuses on the theoretical properties of TMs, it is important to note that TMs have already demonstrated strong empirical performance across a broad range of real-world applications. Published results in these areas demonstrate that TMs can serve as competitive, interpretable, and resource-efficient alternatives to neural networks. Below we provide a few examples of real-world experiments from recent literature.

**Low-Power and Edge Computing** A substantial line of research has explored TM deployments on constrained hardware platforms. The REDRESS framework (Maheshwari et al., 2023) demonstrates that TM-based models outperform binarized neural networks across multiple datasets while offering 5–5700× speed and energy gains on microcontroller hardware. Dedicated TM accelerators have achieved state-of-the-art energy efficiency, including a $65nm$ implementation requiring only $8.6nJ$ per MNIST frame (Tunheim et al., 2025b), currently the lowest reported for MNIST inference in digital circuits. TM-based end to end keyword spotting system, TsetlinKWS (Lin et al., 2025), operates at merely $16.58\mu W$ while maintaining high accuracy, enabled by compression techniques, convolutional TM variants, and custom low-power hardware for Google Speech Commands Dataset.

**Contextual Decision Making** TMs have also been successfully integrated into sequential decision-making settings. By framing the classification task as a contextual multi-armed bandit problem, the TM-Thompson sampling method outperforms other algorithms, including neural network-based approaches, on eight of the nine benchmark environments (Iris, Breast Cancer, MNIST, Adult, Covertype, MovieLens, Statlog, Noisy XOR, and Simulated Article) (Seraj et al., 2022).

**Federated Learning** Recent work has explored TMs in privacy-preserving distributed training. FedTMOS (Qi et al., 2025) introduces a one-shot federated learning (OFL) framework that replaces conventional knowledge distillation with a TM-based, data-free mechanism. It significantly outperforms its ensemble counterpart by an average of 6.16%, and the leading state-of-the-art OFL baselines by 7.22% across various OFL settings. In addition, it results in significantly lower complexity, reduced storage requirements, and improved computational and communication efficiency, while retaining strong accuracy and scalability.

**Image Recognition and Classification** TMs have been applied to standard benchmarks such as MNIST (Tunheim et al., 2025b) and CIFAR-10 (Grønningsæter et al., 2024), as well as domain-specific visual tasks. Recent work demonstrates that TM-based models can match or surpass neural network performance while operating at significantly lower computational cost. For example, Mix-CTME (Jeeru et al., 2025b) achieves robust classification of GPS jamming signals in spectrogram data, outperforming conventional deep learning approaches (99.46% vs. 95.72% on an open benchmark dataset).

**Natural Language Processing.** In NLP, TMs offer interpretable alternatives to dense neural embeddings and opaque text classifiers. TM Embeddings (Bhattarai et al., 2024) demonstrate that word semantics can be captured using compact, human-readable logical clauses rather than latent vectors. Other studies highlight the advantages of TM-based reasoning for interpretable and robust text classification (Yadav et al., 2022) as well as fake news detection (Bhattarai et al., 2022), where TMs match or surpass prior baselines while providing transparent, clause-level explanations for their predictions. Their performance has been consistently validated on widely used open benchmark datasets.

