# OpenReview forum: "Convergence Analysis of Tsetlin Machines under Noise-Free and Noisy Training Conditions: From $2$ Bits to $k$ Bits"
_ICLR.cc/2026/Conference — ICLR 2026 Poster_

### Official Review · Reviewer_ThEH · 2025-10-20

**Soundness:** 3
**Presentation:** 2
**Contribution:** 2
**Rating:** 4
**Confidence:** 3

**Summary:**

A theoretical study of the probabilistic training convergence of Tsetlin Machines for basic logical operators with and without noise.

**Strengths:**

Paper is reasonably clear although the presentation could be improved in parts. For example, a figure would shorten and clarify the setup in Section 2.   It would help to explicitly state that Tables 1 and 2 are transition probabilities to bring along readers who are not already familiar with the material.

**Weaknesses:**

The paper, plus 30 pages of appendix, is a detailed treatise on the convergence of Tsetlin machines for basic logical operators.  It is impossible to estimate the validity of the theorems without a deep study of the lengthy appendix.

While the results are of theoretical interest,  we cannot arrive at any conclusions on practical significance without experiments on real-world data sets.   In the absence of such experiments, the paper might be better suited for a refereed journal.

**Questions:**

How about some experimental results on substantive real data sets?
How can you convince the audience that Tsetlin machines may be practical alternatives to deep neural networks?

---

> ### Author Response · Authors · 2025-11-20
>
> Thank you very much for your time on reviewing this work.
>
> Thank you for your suggestion regarding the presentation of the paper. Due to page limitations, we were not able to include the figure in the main body, but it is available in the appendix. We have also explicitly stated the transition probabilities as suggested.
>
> In addition, the updated version of the paper presents a general proof for the $k$-bit case, extending beyond the original 2-bit operators and making the overall theoretical contribution more comprehensive.
>
> We thank the reviewer for raising the important question regarding the practical significance of TMs and the need for evidence from real-world applications. To address your concern (and that of potential readers), we have added two components to the revised paper:
>
> (1) A dedicated section (**Section 8**), discussing how the theoretical results guide the practical use of TMs. This section highlights the specific insights that the proofs offer for real-world deployment, including implications for hyper-parameter tuning and interpretability.
>
> (2) A dedicated appendix (**Appendix L**) presenting real-world applications of the TM based on open benchmark datasets, demonstrating its practical effectiveness. This will provide readers who are unfamiliar with the TM with concrete examples and greater confidence in including it in their AI toolbox. While the focus of our manuscript is theoretical, we emphasize that **TMs have already shown strong empirical performance across a wide variety of real-world domains**, including computer vision, edge computing, contextual decision making, federated learning, continual learning, and natural language processing. These published results demonstrate that TMs can indeed serve as competitive and efficient alternatives to deep neural networks in practice.

---

> ### Author Response · Authors · 2025-11-20
>
> Here we briefly summarize the practical application examples of TM (corresponding to Appendix L) for your convenience.
>
> **1. Image Recognition**
> TMs have been evaluated on MNIST, CIFAR-10, and domain-specific image tasks, achieving competitive accuracy with significantly lower computational cost.
> Example:
> >- **MixCTME (IEEE IoT-J 2025)**: Robust classification of GPS jamming signals using spectrogram visualizations，outperforming DNN.
>
> **2. Low-Power Edge Computing**
> A growing body of work shows that TMs can be implemented efficiently in digital hardware, often outperforming neural networks in energy and area efficiency. Representative work includes:
> >- **8.6 nJ/frame TM accelerator (65 nm) (IEEE TCAS-I 2025)** energy-efficient MNIST classification, the lowest-power digital circuit reported for MNIST inference to date.
>
> >- **TsetlinKWS** ultra-low-power TM-based keyword spotting (16.58 μW). Introduces the first competitive Convolutional Tsetlin Machine for keyword spotting, combining new spectral features, model compression, and a custom low-power hardware design to achieve high accuracy with extremely low energy consumption.
>
> >- **REDRESS (IEEE TPAMI 2023)** TM based methodology for low and ultra-low power applications, outperforming BNN models on MNIST, CIFAR2, KWS6, Fashion-MNIST and Kuzushiji-MNIST. On an STM32F746G-DISCO microcontroller, REDRESS delivers 5-5700× speed and energy gains over BNN models.
>
> **3. Contextual Bandit Problems**
> TMs have been successfully applied to sequential decision-making tasks:
> >- **Tsetlin Machine for Contextual Bandits (NeurIPS 2022)**：By framing the classification task as a Contextual bandit problem, Thompson sampling with the Tsetlin Machine outperforms other algorithms, including neural network-based approaches, on eight of the nine benchmark environments (Iris, Breast Cancer, MNIST, Adult, Covertype, MovieLens, Statlog, Noisy XOR, and Simulated Article).
>
> **4. Federated Learning**
> TMs can be trained in privacy-preserving distributed settings:
> >- **FedTMOS (ICLR 2025)** This work presents a one-shot federated learning framework based on Tsetlin Machines, achieving lower complexity, higher computational efficiency, and reduced storage compared with deep neural networks. It provides a data-free, compute-efficient alternative to conventional knowledge distillation approaches, offering advantages in accuracy, communication efficiency, and scalability for OFL.
>
> **5. Natural Language Processing**
> TMs provide interpretable rule-based alternatives to deep text models and have been validated on multiple NLP tasks:
> >- **TM Embeddings (EACL Findings 2024)** Tsetlin Machine provides interpretable, compact, and computationally efficient word embeddings by representing word semantics with human-understandable logical clauses rather than dense, opaque floating-point vectors.
>
> >- **Interpretable Text Classification (IJCAI 2022)** By leveraging negated reasoning through the specificity parameter, Tsetlin Machines learn robust, interpretable rules that avoid spurious correlations and achieve substantially better counterfactual performance than standard models.
>
> >- **Fake News Detection (LREC 2022)** Tsetlin Machines enable accurate and interpretable fake-news detection by capturing lexical and semantic patterns with logic-based clauses, outperforming prior baselines and offering clearer explanations than deep neural network models.
>
> **Summary of Our Response**
>
> Although our manuscript focuses on foundational theory, **extensive prior empirical work already demonstrates that Tsetlin Machines are practical, competitive, and efficient** in real-world applications. Our theoretical results are intended to complement this empirical evidence by providing a deeper understanding of *why* TMs possess these desirable properties, such as robustness to noise and the ability to learn sub-patterns jointly and efficiently.

---

> > ### Comment · Reviewer_ThEH · 2025-11-24
> > **Thanks and response to authors**
> >
> > Thanks for your response and the material on the prior empirical work.
> >
> > Without contextual experiments, it remains difficult to determine the practical impact of the paper's results.   My rating stands.

---

> > > ### Author Response · Authors · 2025-11-26
> > > **Further Clarification of Contextual Experiments and Practical Relevance**
> > >
> > > Thank you again for your feedback and for your attention to the practical impact of this work. We would like to further clarify the contextual experiments and the practical impact of the paper’s results.
> > >
> > > **Contextual Experiments:**
> > > As pointed out, the previous proof indeed had limited contextual significance and impact, as its theoretical conclusions were restricted to the 2-bit case. As suggested and inspired by Reviewer aVVF, we have now successfully extended the theoretical analysis from the original 2-bit setting to the general k-bit case. This extension is substantial, as it positions prior empirical studies on the Tsetlin Machine as contextual experiments that align naturally with our generalized theoretical framework.
> > >
> > > All practical Tsetlin machine applications, whether in image processing, NLP, regression, or other domains, depend on a preprocessing stage that converts inputs into k-bit Boolean feature vectors. The k-bit formulation analyzed in this paper therefore corresponds directly to the standard operational setting of the Tsetlin Machine. For instance, the MNIST dataset with 28×28 pixels becomes a k = 784-bit Boolean representation after preprocessing. The generalized k-bit theoretical results thus provide a unifying foundation that consistently explains the convergence behavior, effectiveness, hyperparameter sensitivity, and interpretability patterns repeatedly observed across prior empirical studies. Likewise, the extensive empirical evidence in different applications accumulated in earlier work offers strong support for the theoretical results established here.
> > > This information was not made explicit in the original paper, and we have now make it clear as explained in Remark 6 in the paper.
> > >
> > > To sum up, the combination of (1) our synthetic experiments validating the theoretical claims in Appendices I, J, and K, and (2) the extensive empirical evidence from earlier TM applications (not limited to the ones in Appendix L), together provide strong contextual experimental support for the theoretical findings presented in this work.
> > >
> > > **Practical Impact:**
> > > The theoretical results provide a stronger foundation for future TM applications by offering formal guarantees in certain circumstances,  which were previously unavailable. As discussed in Section 8 of the main paper, we explicitly outline the practical implications of our results and how the theoretical insights can be used to guide and improve real-world TM applications. These findings directly support more reliable deployment, more principled hyper-parameter selection, and improved interpretability of TM-based systems.
> > >
> > > We hope this further clarifies your concerns regarding contextual validation and practical impact. If not, could you please specify what type of contextual experiments you expect? We would be happy to address any remaining issues. Thanks.

---

> > > > ### Comment · Reviewer_ThEH · 2025-11-26
> > > > **addendum**
> > > >
> > > > Thanks for the clarification and the consistent strength.
> > > >
> > > > Your current summary states " Together, these findings provide a robust and comprehensive theoretical foundation for understanding TM convergence."
> > > >
> > > > Perhaps a stronger statement would be appropriate
> > > > " Together, these findings provide a robust and comprehensive theoretical foundation for analyzing TM convergence."
> > > >
> > > > Increased my rating to 6 in light of the improvements.

---

> ### Author Response · Authors · 2025-11-26
> **Thank you**
>
> Thank you for your professional, prompt, and now positive review. Your feedback has been very helpful in clarifying and strengthening the presentation and impact of this work. We also appreciate your suggestion regarding the statement in the abstract, and we will update the wording accordingly.

---

### Official Review · Reviewer_PyEi · 2025-11-01

**Soundness:** 3
**Presentation:** 2
**Contribution:** 3
**Rating:** 6
**Confidence:** 3

**Summary:**

This work provides detailed proofs on the convergence properties of Tsetlin Machines (TMs). Given that existing literature only discussed TMs convergence in 1-bit settings for the Identity/NOT operator and in 2-bit settings for the XOR operator only, it extends the current theoretical understanding of TMs by providing (1) a noise-free (almost surely) convergence proof for the AND operator in a 2-bit setting, (2) a noise-free (almost surely) convergence proof for the OR operator in a 2-bit setting, (3) (almost surely) convergence/recurrence analysis of AND, OR, XOR operators in a 2-bit setting with noisy data (under two random noise types: mislabeled samples and irrelevant input variables).

**Strengths:**

S1: strong motivation is given for improving the theoretical ground of TMs properties, with a clear identification of the gap (what was already shown in the literature, and what was missing).

S2: self-contained notation for setting up the TM functionalities, consistent notations used throughout proofs.

S3: a complete proof provided for (1), and high level proof sketches provided for (2,3) where complex reasoning involves (proof details are presented in the appendix). Figure 2 helps the reader understand the proof structure.

S4: key findings on the OR operator where it is shown to be able to learn joint sub-patterns, which is novel.

S5: the convergence analysis on TM's behaviour with noisy data is particularly valuable as real-world data is more than often quite noisy.

**Weaknesses:**

W1: limited scope - this work explores the 2-bit settings, however, it is missing the discussion of how this work may contribute to analysis of TMs behaviours at a larger scale (in the real deployment of TMs, it is often at a much larger scale). Or more generally, missing discussion on how the theoretical improvements relates to practical usage of TMs.

W2: Section 6 seems to be weaker than section 3&4, partially due to the defer of (almost all) proof details to the appendix and only leaving the statements in the main text. The key rationales behind these findings could be partially included to enhance the strengths of these statements.

**Questions:**

Q1: The definition of the hyper-parameter "T" does not seem to be well presented. It was not clear to the reader what type of values it takes (it was briefly mentioned in one of the proofs that T is an integer, but no formal definition for T is found in the notation section) and what is its range. It would be nice if the authors can present the definition of T better in revisions as it is heavily used in the proofs.

Q2: as discussed in W1, can the authors explains a bit more on how this work influence the practical usage of TMs? Any insights that we can draw from these analysis to be applied to real world scenarios?

---

> ### Author Response · Authors · 2025-11-20
>
> We sincerely thank the reviewer for the thoughtful and insightful assessment of our work, and we greatly appreciate the positive evaluation and recommendation.
>
> **Reviewer (Q1):**
> > Q1: The definition of the hyper-parameter "T" does not seem to be well presented....
>
> **Response:** We appreciate the reviewer’s comment. In the revised manuscript, we have clarified the definition of T in Section 2 (blue text). Specifically,
> - T is a positive integer，with its maximum value equal to the total number of clauses.
> - T limits the maximum number of clauses that can be allocated to each sub-pattern. Once at least T clauses have learned a particular sub-pattern, any samples matching that sub-pattern will no longer trigger TM updates (the probability of triggering feedback is 0). This prevents additional clause resources from being spent on a sub-pattern that is already considered learned (with $T$ clauses representing it). With an appropriate choice of $T$, the clause resources can be balanced across different sub-patterns, ensuring that the system converges.
> ---
> **Reviewer (W1 Q2):**
> > W1: limited scope - this work explores the 2-bit settings, however, it is missing the discussion of how this work may contribute to analysis of TMs behaviours at a larger scale (in the real deployment of TMs, it is often at a much larger scale). Or more generally, missing discussion on how the theoretical improvements relates to practical usage of TMs.
>
> > Q2: as discussed in W1, can the authors explains a bit more on how this work influence the practical usage of TMs? Any insights that we can draw from these analysis to be applied to real world scenarios?
>
> **Response:**
> We have now extended the proof from the 2-bit setting to the general $k$-bit setting with $k > 2$, thereby making the analysis substantially more general.
> We thank the reviewer for raising the important point regarding the relationship between our theoretical findings and the practical deployment of TMs. The main objective of this paper is to provide insights into the learning dynamics of TMs so that practitioners and researchers can better understand how and why TMs behave as they do in real applications. This includes, among other things, more efficient hyper-parameter configuration and improved interpretability. **To address this, we have added a new section (Section 8) where we discuss practical recommendations and implications based on the insights that we draw from the analysis for guiding real-world TM usage.**
>
> In more detail, our analysis highlights two core mechanisms underlying TM learning:
>
> **Joint learning of sub-patterns**:
> As pointed out by the reviewer in S4, our analysis reveals that TMs are capable of learning joint sub-patterns under the OR operator, which was not previously established in TM literature. This contributes directly to understanding how TMs compose and reuse features in more complex real-world tasks. While jointly representing subpatterns makes the learning more efficient, it can affect interpretability: the model no longer captures each distinct subpattern explicitly.
>
> **Robustness to irrelevant bits**:
> Our analysis shows that TMs can automatically mitigate the effect of irrelevant bits during learning. This can occur via two mechanisms:
>
> - First, **when sufficient clause resources are available**: some clauses may include the irrelevant bit (a sub-pattern of the target class), while others include its negation (another sub-pattern of the target class). Both types of clauses will vote for the target class, and the presence of irrelevant bits is effectively neutralized in the voting stage.
>
> - Second, **when clause resources are limited**: irrelevant bits tend to be excluded from clauses entirely, meaning they do not contribute to the final classification decision.
>
> Understanding these mechanisms provides meaningful guidance for deploying TMs in real-world scenarios, especially when data contains substantial noise or irrelevant features, a common situation in practical machine learning tasks.
>
> To further illustrate to readers the practical usage of TM, we have included an additional appendix summarizing established TM application domains and practical advantages. Please refer to Appendix L in the revised paper (or the response to Reviewer ThEH) for discussions.
>
> ---
> **Reviewer (W2)**:
> > W2: Section 6 seems to be weaker than section 3 and 4, ... The key rationales behind these findings
> could be partially included...
>
> **Response:**
> Many thanks for your suggestion. We now have two paragraphs in this section, highlighting the proof workflow and the main insights from the proof. Please refer to the same section highlighted in blue in the revised paper.

---

### Official Review · Reviewer_aVVF · 2025-11-01

**Soundness:** 3
**Presentation:** 3
**Contribution:** 3
**Rating:** 6
**Confidence:** 2

**Summary:**

This paper studies the convergence behavior of Tsetlin Machines (TM) on basic Boolean operators. It proves almost-sure convergence for AND and OR under noise-free data, clarifies how the resource parameter T is necessary for OR (due to recurrent dynamics without it), and revisits XOR to contrast why clauses must include both literals there. The paper then analyzes learning under two noise models, mislabeled samples (non-convergence / recurrence) and irrelevant input variables (preserving almost-sure convergence with suitable T). Experiments on synthetic setups illustrate the theorems and qualitative behaviors.

**Strengths:**

1.	The paper has good theoretical novelty: Clear separation of behaviors for AND vs. OR vs. XOR, including why OR needs 𝑇 to exit recurrence, and why XOR forces inclusion of both literals (Type II feedback pressure).

2.	It has rigorous clause/TA-state analyses leading to a unique absorbing state for AND and a family of absorbing conditions for OR. The use of absorbing-state arguments vs. stationary distributions is a neat angle.

**Weaknesses:**

1.	The theory focuses on 1–2-bit settings. While justified as foundational, readers may want a clear roadmap for extending these techniques to higher-arity clauses and multi-bit operators beyond AND/OR/XOR, or to real datasets with structured features. (Some remarks suggest feasibility, but formal generalizations are not provided.)

2.	Convergence is asymptotic (“infinite time”). Finite-time/sample complexity bounds or rates (even coarse) would materially strengthen the results and their practical import. The empirical section shows good convergence in practice, but theory doesn’t quantify this.

3.	Experiments are primarily synthetic and ablation-style. Comparisons to classical concept learning or bandit baselines learning conjunctions/disjunctions (more related work) would better support the empirical importance.

**Questions:**

1.	Can you provide high-level bounds (even loose) on expected time to absorption under noise-free AND/OR as a function of m,T,s,N and input distribution?

2.	Which steps in the proofs rely critically on the 2-bit structure? Do you foresee analogous absorbing-state conditions for k-bit conjunctions/disjunctions and for multi-clause compositions? Any obstacles beyond combinatorial explosion?

3.	Can you add guidance on the qualitative effect of s (granularity) and N (TA states per action) on convergence speed/stability?

---

> ### Author Response · Authors · 2025-11-20
>
> We sincerely thank the reviewer for the thoughtful and insightful assessment of our work, and we appreciate the positive evaluation and recommendation. Regarding your questions:
>
> ---
>
> **Reviewer (Question 1):**
> > Can you provide high-level bounds (even loose) on expected time to absorption under noise-free AND/OR as a function of m,T,s,N and input distribution?
>
> **Response:**
> A possible approach to deriving finite-time bounds for the noise free case is to start from the practical initialization used in our experiments (also a common practice in real applications), where all TAs begin at the exclude-side borderline. Then the TM can be modeled as a discrete-time Markov chain. By estimating the expected number of transitions required to reach the absorbing states, one could in principle obtain high-level bounds as a function of m,T,s,N and the input distribution. This suggests that a finite-time analysis is feasible in principle.
> However, even this simplified setting remains nontrivial because AND and OR exhibit fundamentally different behaviors. AND has a single positive subpattern, and its TAs continue receiving consistent updates until they reach an edge state. In contrast, OR has three subpatterns, leading to multiple absorbing configurations in which TAs stop receiving updates entirely. These differences alter the transition dynamics and complicate the analysis, even in two-bit examples. When we consider the noise (irrelevant bits in this case), the analysis is to be more complicated. A complete and rigorous finite-time treatment would therefore require a separate, dedicated paper, and we view this as valuable future work.
>
> ---
>
> **Reviewer (Question 2):**
> > Which steps in the proofs rely critically on the 2-bit structure? Do you foresee analogous absorbing-state conditions for k-bit conjunctions/disjunctions and for multi-clause compositions? Any obstacles beyond combinatorial explosion?
>
> **Response:**
> Many thanks for this inspiring question. Your insight made us revisit the problem from a fresh perspective, and we are now able to establish convergence in the general $k$-bit case, a result that has puzzled us for more than four years.
>
> It turned out that the combinatorial explosion was the only real obstacle to a full extension of the proof. Since no deeper structural obstacles were present, the central challenge was simply to avoid being affected by this explosion. To do so, we first moved from the literal level, where the combinatorial growth occurs, to the clause level, where the number of entities is constrained (even though their internal structure is still combinatorial). Secondly, and fortunately, the operational structure of the Tsetlin Machine allows us to naturally clusters clauses into a small number of behavioral categories. By showing that exactly one of these categories is absorbing with the correct sub-pattern(s) while the others are not, we were able to demonstrate convergence.
>
> A key point is that, in the general $k$-bit case, we do not distinguish between AND, OR, or XOR operators. Instead, the proof is formulated in terms of the number of sub-patterns in the target class. We begin by analyzing the unique-single-subpattern case, where only one of the $2^k$ input combinations is labeled as $1$ (analogous to the 2-bit AND setting). We then extend the argument to the multiple-subpattern case (covering, for example, the 2-bit OR and XOR settings, where several input combinations yield output $1$). The structure of the proof follows the same logical progression as the move from the 2-bit AND to the 2-bit OR case, but now in a general form. The complete proof is included in the revised version of the paper.
>
> The paper has undergone major revision, including a new title, abstract, conclusion, and the full convergence proof (marked in blue). Empirical results will be added shortly. We expect empirical results  to be completed before December 3rd, and we will thoroughly polish the manuscript before December 3rd.
>
> Once again, thank you for your inspiring questions. They made the work significantly more robust and complete.
>
> ---

---

> ### Author Response · Authors · 2025-11-20
>
> **Reviewer (Question 3):**
> > Can you add guidance on the qualitative effect of s (granularity) and N (TA states per action) on convergence speed/stability?
>
> **Response:**
> Thanks for this question. We provide here some qualitative guidance on how the parameters s and N affect convergence speed and stability. The parameter s controls the granularity of feedback. Larger values of s tend to include more literals within each clause, which also accelerates convergence in consistent absorbing configurations and improves stability, but at the cost of an increased risk of overfitting in practice. Smaller s generally leads to shorter clauses and slower convergence in practice. The parameter N determines the number of TA states per action and thus governs the “inertia” of each TA. Larger N allows the automaton to move deeper into its action state space, which improves robustness against mislabeled or noisy samples, but requires more time to reach the innermost states and also increases memory usage in practice.
>
> ---
>
> **Reviewer (Weakness):**
> > The theory focuses on 1–2-bit settings. While justified as foundational, readers may want a clear roadmap for extending these techniques to higher-arity clauses and multi-bit operators beyond AND/OR/XOR, or to real datasets with structured features. (Some remarks suggest feasibility, but formal generalizations are not provided.)
>
> > Convergence is asymptotic (“infinite time”). Finite-time/sample complexity bounds or rates (even coarse) would materially strengthen the results and their practical import. The empirical section shows good convergence in practice, but theory doesn’t quantify this.
>
> > Experiments are primarily synthetic and ablation-style. Comparisons to classical concept learning or bandit baselines learning conjunctions/disjunctions (more related work) would better support the empirical importance.
>
> **Response:**
> We agree with the weaknesses identified and appreciate the reviewer’s careful assessment. We thank the reviewer for the suggestion to include comparisons in the numerical results. While it is not feasible to incorporate such comparisons within the current rebuttal period, for reference, tables 7 and 8 in [1] provides empirical results on convergence comparisons for more complex problems than 2-bit cases on concept learning, and additional relevant insights can also be found in [2].
>
> To highlight the diverse applications of the TM, we have added a new appendix (Appendix L) that summarizes several TM applications.
>
> [1] M. B. Belaid et al. “Generalized Convergence Analysis of Tsetlin Automaton–Based Algorithms: A Probabilistic Approach to Concept Learning.” AAAI, 2025.
>
> [2] Raihan Seraj et al. “Tsetlin Machine for Solving Contextual Bandit Problems.” NeurIPS, 2022.
>
> As a remark: we noticed that you assessed your confidence level rather modestly. Based on our experience with your comments and insights, you certainly deserve to rate yourself higher.

---

> > ### Author Response · Authors · 2025-11-30
> > **Empirical results added in the revised version**
> >
> > A new version of the paper is uploaded.
> >
> > We have added a qualitative discussion of the effects of N and s in Appendix A.
> >
> > The empirical results for the k-bit case are provided in Appendix K.
> >
> > Thank you again for your professional evaluation.

---

### Author Response · Authors · 2025-11-30
**Summary of the Rebuttal**

**Regarding the score changes** The scores before reverting were **6, 6, and 6**. Note that Reviewer ThEH, the only reviewer who initially submitted a negative rating, explicitly agreed with our rebuttal and **raised the score from 4 to 6**. This is clearly documented in the official response: “*Increased my rating to 6 in light of the improvements.*” All of this was completed **on the 26th**, well before the 27th. Please refer to the rebuttal exchange with this reviewer for details.

**Summary for rebuttal exchange with Reviewer ThEH** The reviewer's concerns involved the practical impact of the proof, the results on real datasets, and the demonstration that Tsetlin Machines can serve as practical alternatives. These points were clearly addressed in our rebuttal, and **the reviewer acknowledged this by raising the score**.

**Summary for rebuttal exchange with Reviewer PyEi** The reviewer's concerns included the limitations of the 2-bit proof, the proof’s implications for the practical use of Tsetlin Machines, and the explanation of the hyperparameter
T. These points were clearly addressed in our rebuttal, and the reviewer acknowledged this, explicitly stating: “*I will keep my positive rating*.”

**Summary for rebuttal exchange with Reviewer aVVF** The reviewer’s concerns included the comparison with related work, the possibly finite-time analysis, the possibly extension of the proof from 2-bit to the k-bit case, and the qualitative effects of the hyperparameters
s and N. We have resolved the k-bit proof and clarified the qualitative effects of the hyperparameters. The remaining points are addressed by referencing existing articles or by marking them as future work.

Notably,  **the k-bit proof is a significant contribution**, as it provides a general foundation, compared with the 2-bit proof, to explain the convergence behavior of TM. This result had puzzled the authors for years and was ultimately resolved thanks to the reviewer’s insightful comments. Although the reviewer did not respond before being muted, we sincerely appreciate this inspiration and anticipate that he/she would be satisfied with our response, given the original positive assessment of the paper and the significant improvement in the proof.


In short, we have addressed all reviewers’ comments with great care, and all conversations are fully documented online with clear timestamps. We hope the above summary helps reduce the AC’s workload and supports an informed and fair decision.

---

### Meta-Review · Area_Chair_pxGr · 2026-01-06

**Summary:**

This paper provides a convergence analysis of Tsetlin Machines on basic Boolean operators, extending prior results beyond the 1-bit and 2-bit XOR settings to cover 2-bit AND/OR, noisy training regimes (mislabels and irrelevant variables), and a more general k-bit formulation. The reviews are overall positive on the technical direction and rigor, with the main decision-relevant concerns being (1) whether the contribution is “too 2-bit/synthetic” versus broadly useful, and (2) whether the paper sufficiently connects the theory to practical TM usage and existing empirical evidence.

**Reviewer Concerns:**

Concerns addressed by the rebuttal:
1) Limited scope beyond 2-bit: the authors state they extended the analysis to a general k-bit case and revised the paper structure accordingly, which directly targets the biggest “scope” concern.
2) Practical relevance and guidance: the authors added a section on practical implications (especially around the role of T, interpretability trade-offs, and robustness to irrelevant bits) and an appendix summarizing established application domains and empirical results from prior work.
3) Clarity around hyperparameter T and presentation: the authors clarified the definition and role of T, and added qualitative discussion of s and N.

Concerns still outstanding:
1) New contextual experiments in the paper itself remain limited. One reviewer explicitly remained unconvinced without real-dataset experiments in the submission, and the response mainly points to prior literature rather than adding substantial new end-to-end evaluations.
2) Finite-time/rate guarantees are still not provided; the authors frame this as future work.

**Reviewer Scores:**

1) Reviewer aVVF (score 6): likely unchanged at 6; their main requests were about extending beyond 2-bit, clarifying hyperparameter effects, and improving positioning, which the authors addressed.
2) Reviewer PyEi (score 6): likely unchanged at 6; they were already positive and explicitly indicated their concerns were handled.
3) Reviewer ThEH (score 4): likely increase to 5. The k-bit extension and the added practical discussion reduce the “limited contextual significance” concern, but the lack of new real-dataset experiments inside the submission still makes it hard to fully align with a clear accept from this reviewer.

---

### Decision · Program_Chairs · 2026-01-26

Accept (Poster)